# Dual pathways via CENP-C and Mis18C recruit HJURP for CENP-A deposition into vertebrate centromeres

Tetsuya Hori ⬡ ✉, Yutaka Mahana ⬡, Mariko Ariyoshi ⬡ & Tatsuo Fukagawa ⬡ ✉

## Abstract

Centromere position is specified and maintained by sequence-independent epigenetic mechanisms in vertebrate cells, with the incorporation of the centromere-specific histone H3 variant CENP-A into chromatin being a key event for centromere specification. Although many models for CENP-A incorporation have been proposed, much remains unknown. In this study, we reveal that the CENP-A chaperone HJURP directly binds to the C-terminal domain of chicken CENP-C in vitro and that this interaction is essential for new CENP-A incorporation in chicken DT40 cells. While existing models have suggested that HJURP is recruited by the Mis18 complex (Mis18C), here, we propose that CENP-C and Mis18C provide dual recruitment pathways for HJURP localization to centromeres in DT40 cells. We demonstrate that both HJURP localization and new CENP-A incorporation are completely abolished in Mis18C knockout cells expressing an HJURP mutant lacking CENP-C binding ability. Furthermore, co-immunoprecipitation experiments reveal that CENP-C, HJURP and Mis18C form a tight association in the chromatin fraction. These two pathways are critical for robust CENP-A incorporation to maintain centromere position in vertebrate cells.

**Keywords** Centromere; CENP-A; CENP-C; Mis18 Complex; HJURP
**Subject Categories** Cell Cycle; Chromatin, Transcription & Genomics

## Introduction

The centromere is an essential genomic region that ensures accurate chromosome segregation. The kinetochore, which is formed at the centromere, binds to spindle microtubules to facilitate this process during mitosis (Fukagawa and Earnshaw, 2014). In most organisms, the position of the centromere is specified and maintained by sequence-independent epigenetic mechanisms at a specific locus on each chromosome (Black and Cleveland, 2011; Fukagawa and Earnshaw, 2014). The centromere-specific histone H3 variant CENP-A plays a key role in this process

as an epigenetic mark because CENP-A deposition into chromatin triggers the formation of active centromeres (Allshire and Karpen, 2008; Barnhart et al, 2011; Black and Cleveland, 2011; Fukagawa and Earnshaw, 2014; Guse et al, 2011; Mellone and Fachinetti, 2021; Mendiburo et al, 2011; Westhorpe and Straight, 2013). Therefore, many studies have investigated how CENP-A is specifically incorporated into centromeric chromatin (Conti et al, 2024; Fujita et al, 2007; Hayashi et al, 2004; Maddox et al, 2007; Nardi et al, 2016; Pan et al, 2017; Pan et al, 2019; Parashara et al, 2024; Spiller et al, 2017; Subramanian et al, 2016; Thamkachy et al, 2024). These studies suggest that the current model is as follows: prior to G1 phase, Mis18α, Mis18β and KNL2 (M18BP1) form a complex on centromeric chromatin. Then, in G1 phase, the CENP-A chaperone HJURP recognizes and binds to the Mis18 complex (Mis18C), which ensures the correct deposition of new CENP-A on centromeric chromatin from the HJURP-CENP-A-H4 complex in vertebrate cells. This cycle ensures that the centromere remains in the same location across generations. In this model, Mis18C and HJURP play important roles in the deposition of new CENP-A (Fig. 1A, Mis18C pathway).

In parallel with these studies, the protein tethering approach has been useful in understanding the CENP-A deposition mechanism. Using this approach, we created an artificial centromere by tethering centromere components to a non-centromeric locus in chicken DT40 cells with a LacO-LacI system (Cao et al, 2024; Gascoigne et al, 2011; Hori et al, 2013). Similar tethering approaches have been used in various cells, including human and Drosophila cells (Barnhart et al, 2011; Gascoigne et al, 2011; Lacefield et al, 2009; Mendiburo et al, 2011; Palladino et al, 2020; Shono et al, 2015). Using the tethering approach to characterize different artificial centromeres, we categorized them into two types: those that contain CENP-A and those that do not. In CENP-A-less artificial centromeres, tethering proteins directly recruit the microtubule-binding KMN network (KNL1-Mis12-Ndc80 complexes) to a non-centromeric locus. This occurs without the recruitment of the Constitutive Centromere Associated Network (CCAN) proteins, which assemble on CENP-A-containing chromatin (Hori et al, 2013). This type of artificial centromere can be formed by tethering the N-terminal domains of either CENP-T or CENP-C to a non-centromeric locus in DT40 cells. Another type of artificial centromere incorporates CENP-A near a non-centromeric LacO locus. After CENP-A is incorporated into

Graduate School of Frontier Biosciences, The University of Osaka, Suita, Osaka 565-0871, Japan. ✉E-mail: t.hori.fbs@osaka-u.ac.jp; fukagawa.tatsuo.fbs@osaka-u.ac.jp

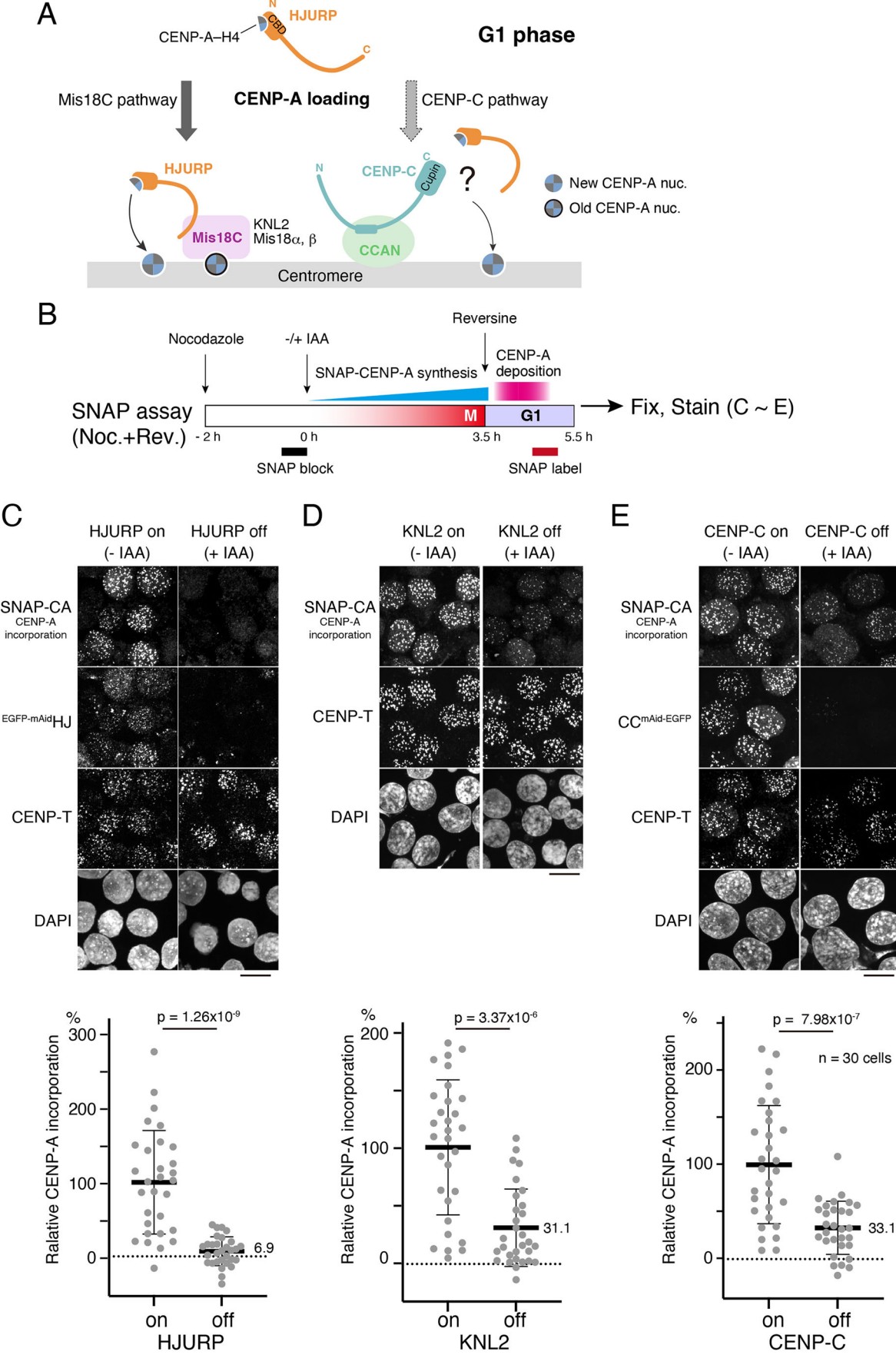

**Figure 1.  New CENP-A incorporation assays in each knockout cell line.**

(A) Schematic models for CENP-A loading into centromeres. Pre-deposited CENP-A forms a complex with histone H4 and the CENP-A chaperone HJURP. The CENP-A-H4-HJURP complex recognizes the Mis18 complex, which associates with centromeric chromatin through CENP-A nucleosome binding, and CENP-A is incorporated into centromeres during G1 phase (Mis18C pathway). CENP-C may be recognized by the CENP-A-H4-HJURP complex and may support CENP-A incorporation into centromeres (CENP-C pathway); however, detailed mechanisms of the CENP-C pathway for new CENP-A loading remain unclear. (B) Schematic representation of the new CENP-A incorporation assay (SNAP assay). AID-based knockout cell lines were cultured with Nocodazole, and existing SNAP-tagged CENP-A was quenched with SNAP-Cell Block followed by the addition of IAA to degrade the target protein. After allowing new CENP-A synthesis for 3.5 h, Reversine was added to permit mitotic exit and G1 progression. Newly incorporated SNAP-CENP-A at the G1 phase was pulse-labeled with TMR-Star and evaluated in each KO cell line. (C) Representative images of the new CENP-A incorporation (SNAP-CENP-A) in AID-based HJURP knockout cell lines in either the absence (HJURP On) or presence (HJURP Off) of IAA. EGFP-mAid-tagged HJURP localization is also shown. CENP-T was used as a centromere marker. DNA was stained with DAPI. Scale bar, 10 μm. SNAP-CENP-A intensities were measured to evaluate CENP-A incorporation into centromeres. Each assay was conducted three times ($n = 30$ cells per condition). The average of SNAP-CENP-A signal intensities of M-phase cells (without Reversine; Fig. EV1E) was subtracted from the corresponding average intensity in each G1 cell to determine CENP-A incorporation. Results are presented as mean relative intensity ± standard deviation (SD). A two-tailed Student's t-test was performed to compare HJURP On and HJURP Off conditions. The p-value is $1.26 \times 10^{-9}$. (D) The new CENP-A incorporation in AID-based KNL2 knockout cell lines in either the absence (KNL2 On) or presence (KNL2 Off) of IAA as shown in (C). CENP-T was used as a centromere marker. DNA was stained with DAPI. Scale bar, 10 μm. Each assay was conducted three times ($n = 30$ cells per condition), and results are presented as mean relative intensity ±SD as described in (C). A two-tailed Student's t-test was performed to compare KNL2 On and KNL2 Off conditions. The p-value is $3.37 \times 10^{-6}$. (E) The new CENP-A incorporation in AID-based CENP-C knockout cell lines in either the absence (CENP-C On) or presence (CENP-C Off) of IAA as shown in (C). mAid-EGFP-tagged CENP-C localization is also shown. CENP-T was used as a centromere marker. DNA was stained with DAPI. Scale bar, 10 μm. Each assay was conducted three times ($n = 30$ cells per condition), and results are presented as mean relative intensity ±SD as described in (C). A two-tailed Student's t-test was performed to compare CENP-C On and CENP-C Off conditions. The p-value is $7.98 \times 10^{-7}$. Source data are available online for this figure.

chromatin, most of the kinetochore components, including the CCAN proteins, assemble there to form a functional kinetochore. This type of artificial centromere forms when HJURP, KNL2, CENP-I, or a C-terminal fragment of CENP-C is tethered into the LacO locus in DT40 cells (Cao et al, 2024; Hori et al, 2013; Perpelescu et al, 2015). The same tethering assay using HJURP, KNL2, CENP-I, or CENP-C in human cells demonstrated CENP-A recruitment to the tethering locus (Shono et al, 2015).

Since HJURP is a CENP-A chaperone and KNL2 is a Mis18C component, and both are Mis18C pathway key factors (Fig. 1A), it can be explained that ectopic tethering of these proteins could induce CENP-A recruitment at a non-centromeric locus. However, explaining why CENP-C or CENP-I recruits CENP-A at non-centromeric loci is not trivial. We previously proposed that CENP-C associates with HJURP to induce CENP-A incorporation at a non-centromeric locus, independently of KNL2 (Cao et al, 2024). This was an interesting result, but it was still unclear if KNL2-independent CENP-A incorporation via CENP-C occurs in native chicken centromeres. Because CENP-C plays multiple roles in centromeres by binding to different proteins, evaluating its role in CENP-A incorporation into native centromeres in vertebrate cells is difficult. Additionally, KNL2 knockout (KO) cells exhibit reduced new CENP-A incorporation and subsequently die due to mitotic defects. This suggests that KNL2 (Mis18C)-mediated CENP-A incorporation is crucial for centromere formation (Fig. 1A, Mis18C pathway). However, it is possible that CENP-C also contributes to the CENP-A incorporation pathway (CENP-C pathway) in addition to the Mis18C pathway (Fig. 1A). In fact, CENP-C-dependent CENP-A incorporation occurs in Xenopus cells (Flores Servin et al, 2023; French and Straight, 2019; French et al, 2017). These previous studies using Xenopus egg extracts investigated CENP-A incorporation into native centromeres and showed that CENP-C directly binds to and recruits HJURP to centromeres with Mis18C. However, some questions remain: (1) How does CENP-C participate in the CENP-A incorporation pathway in native centromeres in cultured cells? (2) If CENP-C recruited HJURP for CENP-A incorporation, what is the binding basis? (3) How do the Mis18C pathway and CENP-C pathways coordinate to deposit CENP-A?

To address these questions, in this study, we performed quantitative ChIP-seq analyses of the centromere region in combination with various knockout cell lines, including CENP-C and KNL2 double knockout DT40 cell lines. We demonstrated that HJURP can be recruited to centromeres to induce new CENP-A incorporation in either CENP-C or KNL2 knockout DT40 cells; however, this recruitment was completely abolished in CENP-C- and KNL2-double-knockout cells. This suggests that CENP-C contributes to CENP-A incorporation. We also identified HJURP-specific residues that bind directly to CENP-C and demonstrated that cells expressing mutant HJURP with altered residues exhibited defects in CENP-A incorporation and growth delays. Together with co-immunoprecipitation (IP), cellular analysis, and in vitro bio-chemistry, our results suggest that the Mis18C and CENP-C pathways operate independently. Based on these findings, we propose the existence of dual CENP-A incorporation pathways via CENP-C and Mis18C in native chicken DT40 centromeres.

## Results

### New CENP-A incorporation is reduced but not completely abolished in either KNL2 or CENP-C knockout cell line

To investigate the role of Mis18C or CENP-C in the incorporation of new CENP-A (Fig. 1A), we performed the SNAP assay (Jansen et al, 2007) (Fig. EV1A–C) using each knockout cell line. As in human cells, new CENP-A incorporation into centromeres occurs during the G1 phase in chicken DT40 cells (Hori et al, 2017; Jansen et al, 2007). However, cells lacking centromere components are typically arrested in mitosis due to the loss of important mitotic functions and the activation of the spindle assembly checkpoint. Therefore, it is difficult to apply the SNAP assay directly to knockout cells lacking centromere proteins in order to evaluate new CENP-A incorporation in the G1 phase. We used Reversine, an inhibitor of the checkpoint protein Mps1, to overcome this problem (Santaguida et al, 2010). Adding Reversine to mitotically arrested cells allows them to enter the next G1 phase without completing mitosis. This approach enables us to examine CENP-A incorporation in the G1 state of each knockout cell

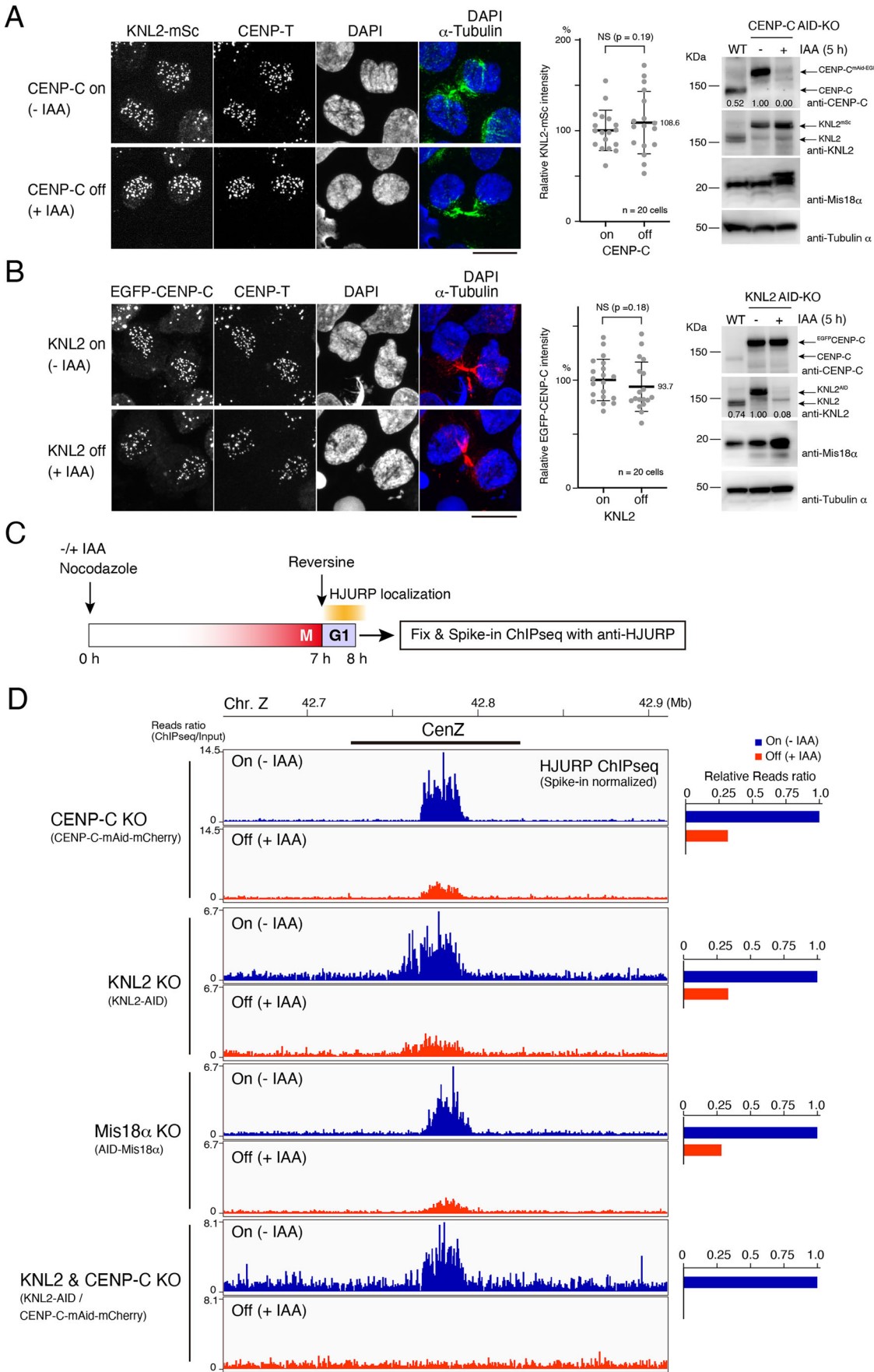

**Figure 2.   Quantification of HJURP levels on centromeres in each knockout cell line.**

(A) KNL2 localization in G1 phase cells in AID-based CENP-C knockout cell lines in either the absence (CENP-C On) or presence (CENP-C Off) of IAA. CENP-T was used as a centromere marker. DNA was stained with DAPI. α-tubulin was used to identify G1 cells. Scale bar, 10 µm. Signal intensities of mScarlet-tagged KNL2 were measured in CENP-C On or CENP-C Off cells. Each assay was conducted twice ($n = 20$ cells per condition), and results are presented as the mean of relative intensities ± standard deviation (SD). A two-tailed Student's t-test was performed to compare CENP-C On and CENP-C Off conditions. The p-value is shown. Immunoblot analyses were performed to detect CENP-C, KNL2, and Mis18α in AID-based CENP-C knockout cell lines in either the absence (−) or presence (+) of IAA (5 h). Signal intensities of CENP-C or CENP-C$^{mAid-EGFP}$ protein on the blot were measured with normalization to α-tubulin intensity in each blot. Wild-type cells (WT) were also analyzed. α-tubulin was probed as a loading control. Molecular weight markers (kDa) are shown on the left. (B) CENP-C localization in G1 phase cells in AID-based KNL2 knockout cell lines in either the absence (KNL2 On) or presence (KNL2 Off) of IAA. CENP-T was used as a centromere marker. DNA was stained with DAPI. α-tubulin was used to identify G1 cells. Scale bar, 10 µm. Signal intensity of EGFP-tagged CENP-C was measured in KNL2 On or KNL2 Off cells. Each assay was conducted twice ($n = 20$ cells per condition), and results are presented as the mean of relative intensities ± SD. A two-tailed Student's t-test was performed to compare KNL2 On and KNL2 Off conditions. The p-value is shown. Immunoblot analyses were performed to detect CENP-C, KNL2, and Mis18α in AID-based KNL2 knockout cell lines in either the absence (−) or presence (+) of IAA (5 h). Signal intensities of KNL2 or KNL2$^{AID}$ protein on the blot were measured with normalization to α-tubulin intensity in each blot. WT cells were also analyzed. α-tubulin was probed as a loading control. Molecular weight markers (kDa) are shown on the left. (C) Experimental scheme for quantitative ChIP-seq with an anti-HJURP antibody to evaluate HJURP levels at G1 centromeres. AID-based conditional knockout cell lines were cultured with Nocodazole in the absence (On) or presence (Off) of IAA for 8 h, with Reversine addition 1 h before fixation. The fixed G1-accumulated cells were subjected to Spike-in ChIP-seq. (D) Spike-in normalized ChIP-seq profiles with an anti-HJURP antibody around the centromere of the chicken chromosome Z (CenZ) in AID-based CENP-C, KNL2, Mis18α, and KNL2/CENP-C double knockout cell lines in either the absence (On) or presence (Off) of IAA. Quantification of the normalized mapped-read ratios on the CenZ region is shown on the right. Source data are available online for this figure.

line after Reversine is added, even if the cells are arrested in mitosis (Figs. 1B and EV1D,E).

First, we generated auxin-inducible degron (AID)-based conditional knockout (cKO) cell lines for HJURP, KNL2, and CENP-C (Fig. EV1A–C). The target proteins in these cell lines are degraded immediately upon the addition of auxin (indole-3-acetic acid, IAA) (Fig. EV1A–C). We observed that the degradation efficiency of target proteins varies in these knockout cells. While residual protein may play a role, we demonstrate that it is almost undetectable in these knockouts (Fig. EV1A–C). We also confirmed that G1 synchronization is effective with our protocol (Fig. EV1D). In our protocol, we used Nocodazole to synchronize mitotic cells and quench old SNAP-CENP-A just before adding IAA. Then, we cultured the cells for 3.5 h after adding IAA. Then, Reversine was added. One hour later, we pulse-labeled SNAP-CENP-A and fixed the cells to evaluate the incorporation of new CENP-A two hours after the addition of Reversine (Fig. 1B). At this time, more than 70% of the cells were in the late G1 phase, during which CENP-A incorporation occurs (Figs. 1B and EV1D). Because new CENP-A was not detected in mitotic cells (Fig. EV1E), we evaluated the incorporation of new CENP-A as SNAP-CENP-A signals in G1 cells, subtracted from the signals in mitotic cells.

We added IAA to cKO-HJURP cells and found that SNAP-CENP-A signals were significantly reduced (by 6.9%) compared to the signals in the cells before IAA addition (Fig. 1C). This indicates that new CENP-A incorporation is mostly impaired in HJURP KO cells. We also tested CENP-A incorporation in the presence of IAA in cKO-KNL2 or cKO-CENP-C cells and found that SNAP-CENP-A signal intensities were significantly reduced (Fig. 1D,E). However, SNAP-CENP-A intensities in KNL2 KO cells (31.1%) and CENP-C KO cells (33.1%) were higher than in HJURP KO cells (6.9%). Since KNL2 is a Mis18C member, we tested new CENP-A incorporation in Mis18α KO cells (another Mis18C member) (Fig. EV1F–J) and found that SNAP-CENP-A intensities were higher in Mis18α KO cells (28.8%) than in HJURP KO cells (6.9%). Based on these results, we propose that both Mis18C and CENP-C contribute to new CENP-A incorporation. However, new CENP-A incorporation occurs in individual knockout cell lines, but is almost completely abolished in HJURP KO cells.

## Centromeric HJURP localization is completely abolished in a KNL2 and CENP-C double knockout cell line

HJURP is the most important factor for the incorporation of new CENP-A because it is a CENP-A-specific chaperone. Since HJURP localizes to centromeres by recognizing Mis18C, we examined its centromeric localization by quantitative ChIP-seq analysis in G1 cells after knockout of KNL2 or CENP-C. Prior to this analysis, we examined the localization of KNL2 at centromeres in cKO-CENP-C cells and the localization of CENP-C at centromeres in cKO-KNL2 cells. As shown in Fig. 2A,B, the localization of KNL2 and CENP-C at centromeres was not significantly altered in cKO-CENP-C and cKO-KNL2 G1 cells, respectively, after the addition of IAA. Additionally, we found that Mis18α and CENP-C localization at centromeres was not dramatically altered in cKO-CENP-C and cKO-Mis18α G1 cells after IAA addition, though CENP-C levels decreased by 82.0% in Mis18α KO cells (Fig. EV2A,B). Based on these results, we conclude that the centromeric localization of CENP-C and Mis18C is largely independent of each other.

Next, we examined HJURP levels at centromeres during the G1 phase in each KO cell line using quantitative ChIP-seq analysis (Fig. 2C,D) to corroborate the findings of the cytogenetic methods. Since the chicken genome contains non-repetitive centromeres on chromosomes, such as chromosomes Z (CenZ) and 5 (Cen5) (Shang et al, 2010), the quantification of sequence reads on these regions is an accurate method for evaluating HJURP levels. In cKO-CENP-C cells, the spike-in normalized ChIP-seq read ratio using an anti-HJURP antibody on CenZ decreased to ~30% after IAA addition compared to pre-IAA addition levels (Fig. 2D). A similar reduction profile was observed for the ChIP-seq read ratio using an anti-HJURP antibody on Cen5 (Fig. EV2G). In cKO-KNL2 and cKO-Mis18α cells, the ChIP-seq read ratio using an anti-HJURP antibody on CenZ or Cen5 decreased to 25–30% after IAA addition compared to pre-IAA addition levels (Figs. 2D and EV2G). These ChIP-seq data are consistent with data on new CENP-A incorporation (Fig. 1). Residual HJURP on centromeres may support weak CENP-A incorporation in CENP-C KO, KNL2 KO, and Mis18α KO cells. Finally, we generated a conditional double KO for KNL2 and CENP-C and performed quantitative ChIP-seq analysis using an anti-HJURP antibody. We found that HJURP centromere

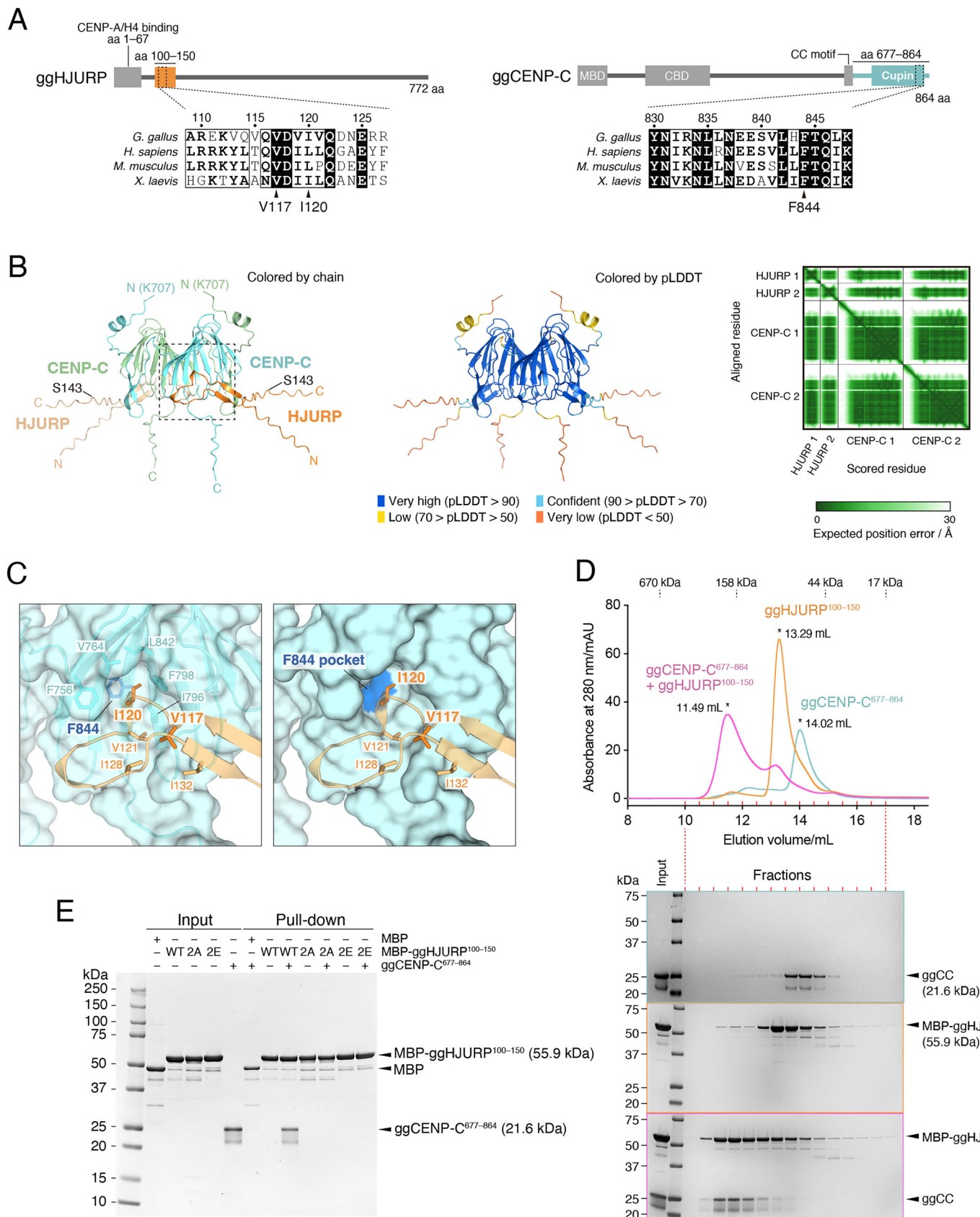

**Figure 3.   Chicken HJURP binds to chicken CENP-C C-terminal region.**

(A) Schematic representation of domains in chicken HJURP (ggHJURP) and CENP-C (ggCENP-C). The HJURP and CENP-C regions (HJURP, aa 100–150; CENP-C, 677–864) used for AlphaFold3 (AF3) prediction and in vitro analysis are highlighted as orange and light blue boxes, respectively. Sequence alignments of the predicted interaction regions from vertebrates (chicken: *G. gallus*, human: *H. sapiens*, mouse: *M. musculus*, and frog: *X. laevis*) are shown with key residues indicated by arrowheads. (B) Predicted structure generated by AF3 is colored by chain (left) or by the pLDDT values (middle) showing the interaction between ggHJURP (aa 100–150) and a dimer of ggCENP-C C-terminal region (aa 677–864 region). The N-terminal region of the CENP-C model with low pLDDT scores <50 (aa 677–706) is therefore omitted for clarity. The potential phosphorylation site, S143 of ggHJURP, is indicated in the structure model (left). The dashed box indicates the interaction surface enlarged in (C). The predicted alignment error (PAE) value plot is shown on the right. (C) Detailed view of the predicted interface between ggHJURP and ggCENP-C. V117 and I120 of ggHJURP interact with a hydrophobic pocket containing F844 of ggCENP-C (highlighted in blue). (D) Validation of the predicted interaction between ggHJURP[100-150] and ggCENP-C[677-864] using size exclusion chromatography. Elution profiles of the indicated proteins were monitored by 280 nm absorbance (upper). SEC fractions from each chromatogram were analyzed by SDS-PAGE (lower). Molecular weight markers (kDa) and elution volume (ml) are shown on the top and bottom, respectively (upper). Molecular weight markers (kDa) are shown on the left (lower). (E) MBP pull-down assay demonstrating the interaction between recombinant MBP-fused ggHJURP[100-150] and ggCENP-C[677-864]. Wild-type MBP-ggHJURP[100-150] pulled down ggCENP-C[677-864], while MBP-ggHJURP[100-150] variants carrying either V117E/I120E (2E) or V117A/I120A (2A) mutations failed to interact with ggCENP-C[677-864], confirming the importance of these hydrophobic residues for the interaction. Molecular weight markers (kDa) are shown on the left. Source data are available online for this figure.

localization was completely abolished in KNL2/CENP-C double KO cells (Figs. 2D and EV2G), indicating that HJURP centromeric localization is supported by both CENP-C and the Mis18 complex.

## Chicken HJURP directly binds to chicken CENP-C

Human Mis18C binds directly to human HJURP (Nardi et al, 2016; Pan et al, 2019). Although HJURP co-precipitates with CENP-C in several species including human, chicken, and Xenopus as demonstrated by pull-down experiments (Cao et al, 2024; Flores Servin et al, 2023; Tachiwana et al, 2015), it is unclear which chicken HJURP residues are involved in direct binding to chicken CENP-C. Previous co-precipitation experiments in chicken cells indicated that the C-terminal region of chicken CENP-C may interact with the N-terminal region of HJURP (Cao et al, 2024). We generated an AlphaFold3 (AF3) model (Abramson et al, 2024) of ggHJURP bound to ggCENP-C using these regions: chicken HJURP amino acids (aa) 100–150 (ggHJURP[100-150]) and chicken CENP-C aa 677–864 (ggCENP-C[677-864]). Figure 3A,B shows the model. The CENP-C binding region of ggHJURP (aa 111–138) was predicted to adopt a two-stranded β-sheet with a high confidence level (pLDDT > 70) (Fig. 3B). In this model, the Cupin domain of the ggCENP-C protein accommodates a protruding loop (aa 116–131) of ggHJURP between the β-strands in an evolutionarily conserved hydrophobic pocket. This allows two HJURP fragments to bind to the Cupin dimer of ggCENP-C (Fig. 3B,C). Specifically, the V117 and I120 residues of ggHJURP in the loop are thought to be involved in interactions with the hydrophobic pocket containing F844 of CENP-C (Fig. 3A,C). To verify the predicted interaction, we expressed MBP-fused ggHJURP[100-150] and ggCENP-C[677-864] in *E. coli* cells and purified these recombinant proteins. Size exclusion chromatography analysis revealed that the peaks of each protein were significantly shifted when the two proteins were mixed (Fig. 3D), indicating that ggHJURP[100-150] interacts with ggCENP-C[677-864] in vitro. MBP pull-down experiments with MBP-ggHJURP[100-150] and ggCENP-C[677-864] confirmed the interaction of these proteins, but ggCENP-C[677-864] was not pulled down with MBP-ggHJURP[100-150] having V117A/I120A or V117E/I120E mutations (referred to as 2 A or 2E mutation, respectively) (Fig. 3E). Amino acid sequence comparison suggests that the CENP-C binding key residues of HJURP are conserved (Fig. 3A). However, no confident HJURP-CENP-C interface was identified in the AF3 model of the human HJURP-CENP-C complex containing hsHJURP[137-192] and hsCENP-C[760-943] corresponding

to ggHJURP[100-150] and ggCENP-C[677-864], respectively (Fig. EV3A). hsHJURP[137-192] is unlikely to adopt the β-sheet predicted in ggHJURP[100-150], which contributes to the formation of the protruding loop containing CENP-C binding key residues. To test this, we purified recombinant hsHJURP[137-192] and hsCENP-C[760-943], and performed a pull-down assay. Unlike ggHJURP[100-150], hsHJURP[137-192] did not efficiently pull-down hsCENP-C[760-943] (Fig. EV3B), indicating that hsHJURP does not bind to hsCENP-C via the HJURP aa 137–192 region. A previous study demonstrated that hsHJURP interacts with hsCENP-C by in vitro pull-down experiments (Tachiwana et al, 2015). Therefore, it is possible that other regions of hsHJURP interact with hsCENP-C. However, our AF3 prediction detected no reliable interface when analyzing the full-length hsHJURP and hsCENP-C C-terminal regions (Fig. EV3C).

A previous experiment using Xenopus proteins (Flores Servin et al, 2023) suggested that HJURP phosphorylation at S220 by CDK inhibits the HJURP-CENP-C interaction, thereby supporting the G1-specific HJURP-CENP-C interaction. The AF3 prediction of the Xenopus HJURP-CENP-C (xlHJURP-xlCENP-C) complex indicates that xlHJURP interacts with xlCENP-C similarly to how ggHJURP interacts with ggCENP-C. Additionally, S220 of xlHJURP is positioned near the CENP-C binding interface (Fig. EV3D). In contrast, the potential CDK phosphorylation site S143 of ggHJURP in the AF3 model of the ggHJURP-ggCENP-C complex is located in an extended C-terminal region outside the CENP-C binding site (Fig. 3B). In support of this prediction, the ggHJURP[100-150] protein with a phosphomimetic mutation at the corresponding residue (S143D) bound to ggCENP-C[677-864] (Fig. EV3E). Based on our biochemical data and AF3 prediction, we conclude that ggHJURP[100-150] binds directly to ggCENP-C[677-864] and its binding regulation is different from that of Xenopus proteins.

## DT40 cells expressing an HJURP mutant lacking CENP-C interaction show growth delay

Next, we investigated the contribution of the HJURP-CENP-C interaction to new CENP-A incorporation in DT40 cells. We introduced constructs containing EGFP-fused wild-type HJURP (EGFP-HJURP[WT]) or HJURP with the 2E mutation (EGFP-HJURP[2E]) into the PGK1 locus in tetracycline (Tet)-based cKO-HJURP cells (Fig. EV4A). In the presence of Tet, cKO-HJURP cells failed to grow (HJURP-cKO cells, Fig. 4A). However, expression of EGFP-HJURP[WT] rescued the growth defects in cKO-HJURP cells,

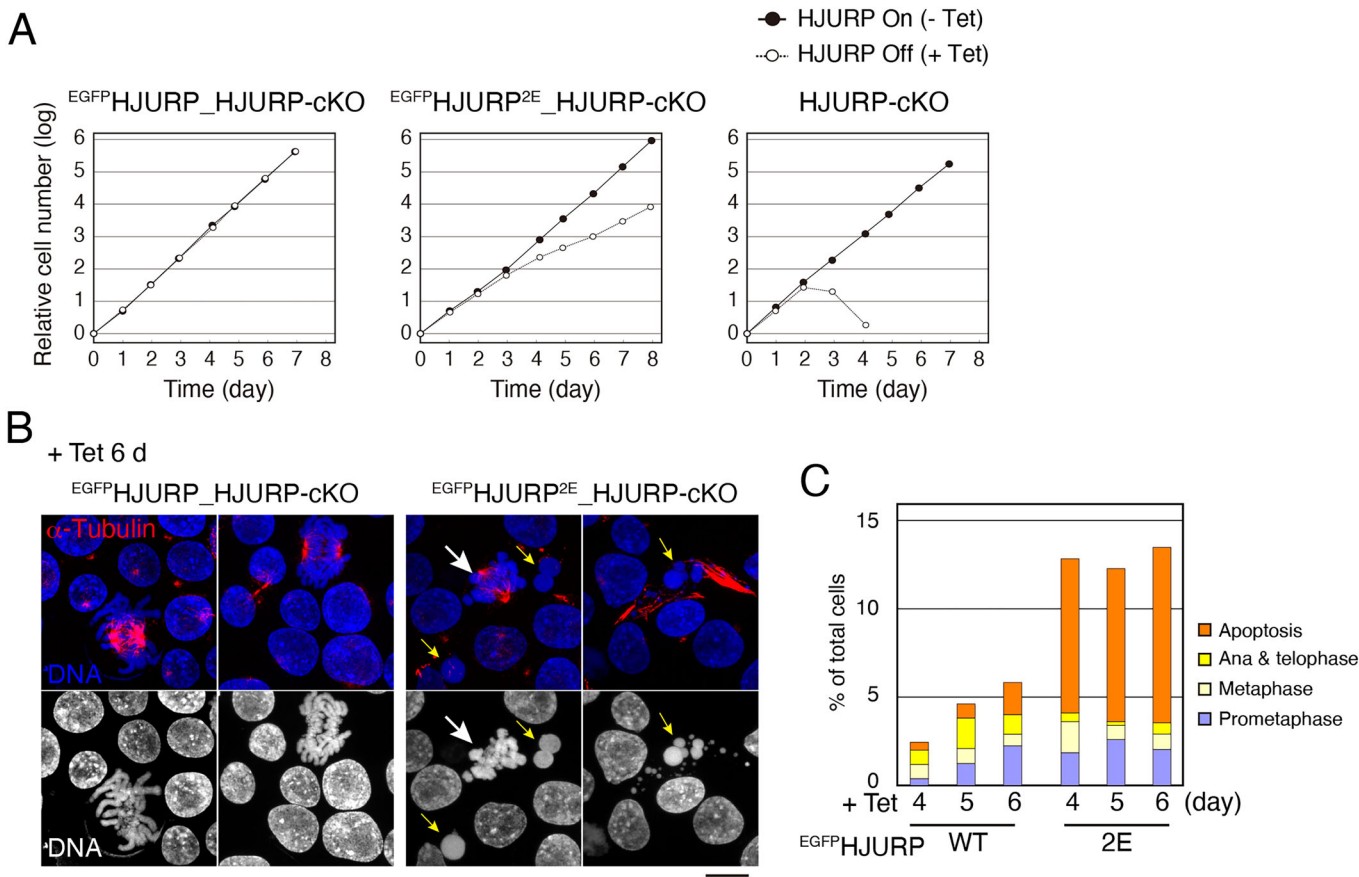

**Figure 4. HJURP-cKO DT40 cells expressing HJURP²ᴱ shows growth delay.**

(A) Growth curve of Tet-based HJURP conditional knockout cells (HJURP-cKO), and cells either expressing EGFP-fused HJURP or HJURP²ᴱ in the absence or presence of Tet. Log-scaled relative cell numbers were measured every 24 h for each cell line. (B) Immunofluorescence images showing α-tubulin staining of HJURP-cKO cells expressing EGFP-fused HJURP or HJURP²ᴱ at 6 days after Tet addition. Yellow arrows indicate apoptotic cells and a white arrow indicates an abnormal mitotic cell with condensed and misaligned chromosomes. DNA was stained with DAPI. Scale bar, 10 μm. (C) Cell cycle distribution (prometaphase, metaphase, anaphase and telophase, apoptotic cells) of HJURP-cKO cells expressing EGFP-fused HJURP (WT) or HJURP²ᴱ (2E) at 4, 5, and 6 days after Tet addition. The proportion of cells was scored based on DAPI and α-tubulin staining profiles. Source data are available online for this figure.

even after the addition of Tet (ᴱᴳᶠᵖHJURP_HJURP-cKO cells, Fig. 4A). Conversely, expression of EGFP-HJURP²ᴱ did not fully restore growth in cKO-HJURP cells in the presence of Tet (ᴱᴳᶠᵖHJURP²ᴱ_HJURP-cKO cells, Fig. 4A). This suggests that the HJURP-CENP-C interaction is necessary for proper cell proliferation in DT40 cells. We examined the cell cycle and mitotic progression of ᴱᴳᶠᵖHJURP_HJURP-cKO and ᴱᴳᶠᵖHJURP²ᴱ_HJURP-cKO cells after adding Tet (Figs. 4B,C and EV4B,C). Flow cytometry analysis revealed an increased G2/M and sub-G1 fraction in ᴱᴳᶠᵖHJURP²ᴱ_HJURP-cKO cells compared to ᴱᴳᶠᵖHJURP_HJURP-cKO cells (Fig. EV4B,C). In addition, we examined the mitotic behavior of ᴱᴳᶠᵖHJURP²ᴱ_HJURP-cKO cells using a microscope (Fig. 4B,C). We observed an increase in abnormal mitotic and apoptotic cells in the ᴱᴳᶠᵖHJURP²ᴱ_HJURP-cKO cells. This suggests that the abnormal mitotic cells caused apoptosis in the ᴱᴳᶠᵖHJURP²ᴱ_HJURP-cKO cells.

The N-terminal domain of HJURP binds to CENP-A. It was possible that HJURP²ᴱ could reduce HJURP's ability to bind to CENP-A. However, since HJURP²ᴱ efficiently co-precipitated CENP-A like HJURPᵂᵀ (Fig. EV4D), this was not the case.

Next, we examined HJURP levels at centromeres in CENP-C or KNL2 KO cells expressing EGFP-HJURP²ᴱ (Fig. 5A). For this experiment, we generated cKO-AID-CENP-C/cKO-Tet-HJURP or cKO-AID-KNL2/cKO-Tet-HJURP cells expressing either EGFP-HJURPᵂᵀ or EGFP-HJURP²ᴱ (Fig. EV4E). First, we evaluated the localization of EGFP-HJURP by microscopic observation. Since HJURP localization is transient, we used cyclin B2 to identify early G1 cells (Fig. EV4F). We found that HJURP signals were very strong in the cyclin B2-negative cells during the G1 phase (Fig. EV4F). Therefore, we evaluated EGFP-HJURPᵂᵀ or EGFP-HJURP²ᴱ levels in the presence of Tet with or without IAA in cKO-AID-CENP-C or cKO-AID-KNL2 cells in the early G1 phase (cyclin B2-negative cells). We observed a clear centromeric localization of EGFP-HJURPᵂᵀ in early G1 cells expressing both CENP-C and KNL2 (Fig. 5B,C, KNL2 On or CENP-C On). EGFP-HJURPᵂᵀ levels were reduced in CENP-C or KNL2 knockout cells (18.8% and 51.9% of the median relative intensity, respectively), but centromeric signals were still visible (Fig. 5B,C, KNL2 Off or CENP-C Off), consistent with quantitative ChIP-seq data in these knockout cells (Fig. 2D). We also observed EGFP-HJURP²ᴱ

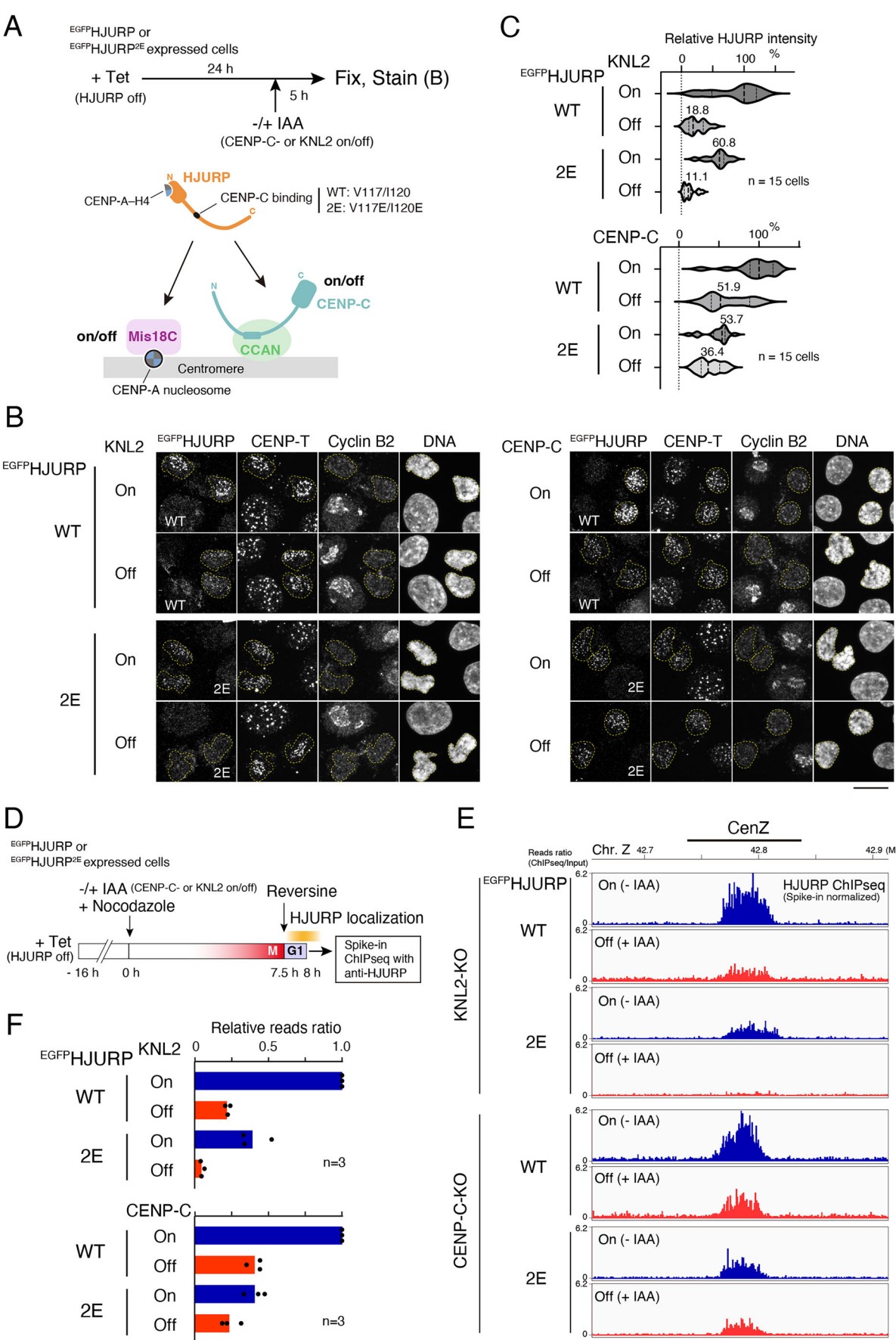

**Figure 5. Expression of HJURP²ᴱ lacking its CENP-C interaction reduces centromere localization of HJURP in DT40 cells.**

(A) Experimental scheme to analyze the significance of the interaction between CENP-C and HJURP in cells. EGFP-HJURP (WT) or EGFP-HJURP²ᴱ (2E) expressing AID-based CENP-C or KNL2 knockout cells with HJURP-cKO background were generated. Tetracycline was added to the culture medium 24 h before fixation to replace wild-type HJURP expression with EGFP-fused HJURP variants. At 5 h after IAA addition, CENP-C or KNL2 were undetectable in the respective knockout cells as shown in Fig. EV4E. In CENP-C On or Off cells expressing EGFP-HJURP²ᴱ, the Mis18C pathway is expected to be still active, while in KNL2 Off cells expressing EGFP-HJURP²ᴱ, both Mis18C and CENP-C pathways are expected to be inactive. (B) Localization of EGFP-tagged HJURP (WT) or HJURP²ᴱ (2E) in AID-based CENP-C or KNL2 knockout cells with HJURP-cKO background as described in (A). Cyclin B2-negative cells were identified as early G1 cells. Cyclin B2 staining profiles with HJURP localization are shown in Fig. EV4F. CENP-T was used as a centromere marker. DNA was stained with DAPI. Scale bar, 10 μm. (C) Relative intensities of EGFP-HJURP (WT) or EGFP-HJURP²ᴱ (2E) in AID-based CENP-C or KNL2 knockout cell lines in the absence or presence of IAA. Each assay was conducted twice (n = 15 cells per condition), and results are presented as median relative signal intensities (bold dotted-line) with first and third quartiles (dotted-lines) using violin plots. (D) Experimental scheme for quantitative ChIP-seq with an anti-HJURP antibody to evaluate HJURP levels at G1 centromeres in AID-based CENP-C or KNL2 knockout cell lines expressing EGFP-HJURP (WT) or EGFP-HJURP²ᴱ. Cells were treated with tetracycline for 24 h before fixation to replace endogenous HJURP expression with the EGFP-fused HJURP variants. Cells were then cultured with Nocodazole in the absence or presence of IAA for 8 h, with Reversine added 0.5 h before fixation. Fixed G1-accumulated cells were subjected to Spike-in ChIP-seq. (E) Spike-in normalized ChIP-seq profiles around the centromere of the chicken chromosome Z (CenZ) with an anti-HJURP antibody in AID-based CENP-C or KNL2 knockout cells expressing EGFP-HJURP (WT) or EGFP-HJURP²ᴱ (2E) in either the absence or the presence of IAA as described in (D). (F) Quantification of normalized mapped-read ratios on the CenZ region by spike-in ChIP-seq experiments shown in (E). Each assay was performed three times. Results are presented as the mean with individual experimental values shown as dots. For the ChIP-seq profiles, a representative data set is shown in (E). Source data are available online for this figure.

centromeric localization in KNL2 On or CENP-C On cells, but these levels were significantly reduced (60.8% and 53.7% of the median relative intensity, respectively) compared to those of HJURPᵂᵀ (KNL2 On 2E or CENP-C On 2E, Fig. 5B,C). EGFP-HJURP²ᴱ signals were further reduced in KNL2 knockout cells (11.1%); however, EGFP-HJURP²ᴱ levels at centromeres were only slightly reduced in CENP-C knockout cells (36.4%, CENP-C Off 2E) (Fig. 5B,C). These data suggest that HJURP²ᴱ disrupts the CENP-C pathway and that HJURP²ᴱ expression in KNL2 knockout cells affects both pathways. Accurate measurement of HJURP intensities in G1 cells by microscopic imaging is challenging because HJURP levels fluctuate during G1 progression, resulting in variable signal intensities. To analyze a larger number of cells, we employed a quantitative ChIP-seq approach, as described below.

To more accurately assess HJURP levels at G1 centromeres, we performed quantitative ChIP-seq analysis for cells treated with Nocodazole and Reversine (Fig. 5D–F). Using this method, we were able to quantitatively compare HJURP levels at centromeres across different experimental conditions. Consistent with the data in Fig. 2D, HJURPᵂᵀ levels were reduced to 25–40% on centromeres in cKO-KNL2 or cKO-CENP-C cells after IAA addition (KNL2 Off WT, CENP-C Off WT, Figs. 5E,F and EV4G). HJURP²ᴱ levels were approximately 30–40% of HJURPᵂᵀ levels at centromeres in KNL2 and CENP-C-expressing cells (KNL2 On 2E, CENP-C On 2E, Figs. 5E,F and EV4G). Although HJURP²ᴱ levels on centromeres were 30–40% in KNL2 On cells, they were close to 0% in KNL2 Off cells (KNL2 Off 2E, Figs. 5E,F and EV4G). In contrast, HJURP²ᴱ levels remained at ~25% in HJURP²ᴱ expressing CENP-C KO cells (CENP-C Off 2E, Figs. 5E,F and EV4G). Based on these results, we conclude that HJURP binds to CENP-C via the V117 and I120 residues of HJURP. Additionally, HJURP²ᴱ did not show a further reduction in CENP-C KO cells. However, HJURP²ᴱ levels decreased further in KNL2 KO cells. This finding suggests that there are two independent pathways—the Mis18C pathway and the CENP-C pathway—to recruit HJURP to centromeres.

## Stable CENP-C-HJURP interaction occurs in the absence of the Mis18 complex

The chicken HJURP aa 100-150 region binds to CENP-C, while the C-terminal region of chicken HJURP has been suggested to interact

with Mis18C (Perpelescu et al, 2015). Human Mis18C also interacts with the HJURP C-terminal region (Pan et al, 2019). Although the C-terminal region of HJURP is poorly conserved, careful sequence comparisons of chicken HJURP with human and mouse proteins, as well as structure predictions using AF3, identified a fragment of chicken HJURP containing aa 480-580 as a potential Mis18C binding module (Fig. EV5A–C). This region was found to directly bind the chicken Mis18 α/β complex in vitro; CENP-C did not compete with Mis18 α/β for binding to this HJURP fragment (Fig. EV5D,E). Furthermore, we confirmed that the HJURP domain containing aa 480-580 is necessary for HJURP to localize to centromeres independently of CENP-C (Fig. EV5F–I).

These data suggest that the chicken HJURP protein can simultaneously bind to CENP-C and Mis18C using its N-terminal and C-terminal regions, respectively (Fig. 6A, bottom). This model proposes that HJURP forms a ternary complex with Mis18C and CENP-C. Additionally, previous reports have indicated that Mis18C binds to CENP-C in various species. Furthermore, the levels of Mis18C at centromeres were reduced in CENP-C KO in these species (Dambacher et al, 2012; McKinley and Cheeseman, 2014; Moree et al, 2011; Stellfox et al, 2016). However, our previous studies and the current results shown in Figs. 2A and EV2A indicate that the localization of Mis18C does not depend on CENP-C in chicken cells during the G1 phase (Hori et al, 2017; Perpelescu et al, 2015) and the level of CENP-C at centromeres does not change in KNL2 KO DT40 cells (Fig. 2B). This suggests that the localization of Mis18C and CENP-C at centromeres is independent. This finding supports the top model in Fig. 6A.

To further understand the association and dependency among Mis18C, HJURP and CENP-C in DT40 cells at G1 phase, we performed co-immunoprecipitation (IP) experiments using different KO cell lines synchronized in G1 phase. In KNL2 On G1 cells, EGFP-CENP-C co-precipitated with both HJURP and Mis18C. When KNL2 was knocked out (KNL2 Off), EGFP-CENP-C co-precipitated with HJURP with the same efficiency as in KNL2 On G1 cells (Fig. 6B), indicating that the stable CENP-C-HJURP interaction occurs without Mis18C. Next, we tested the CENP-C-Mis18C association. In HJURP On cells, CENP-C-mAid-EGFP co-precipitated with KNL2 and Mis18α (Fig. 6C). However, in HJURP Off cells, the intensities of KNL2 and Mis18α co-precipitated with CENP-C were weaker than in HJURP On cells (Fig. 6C), indicating

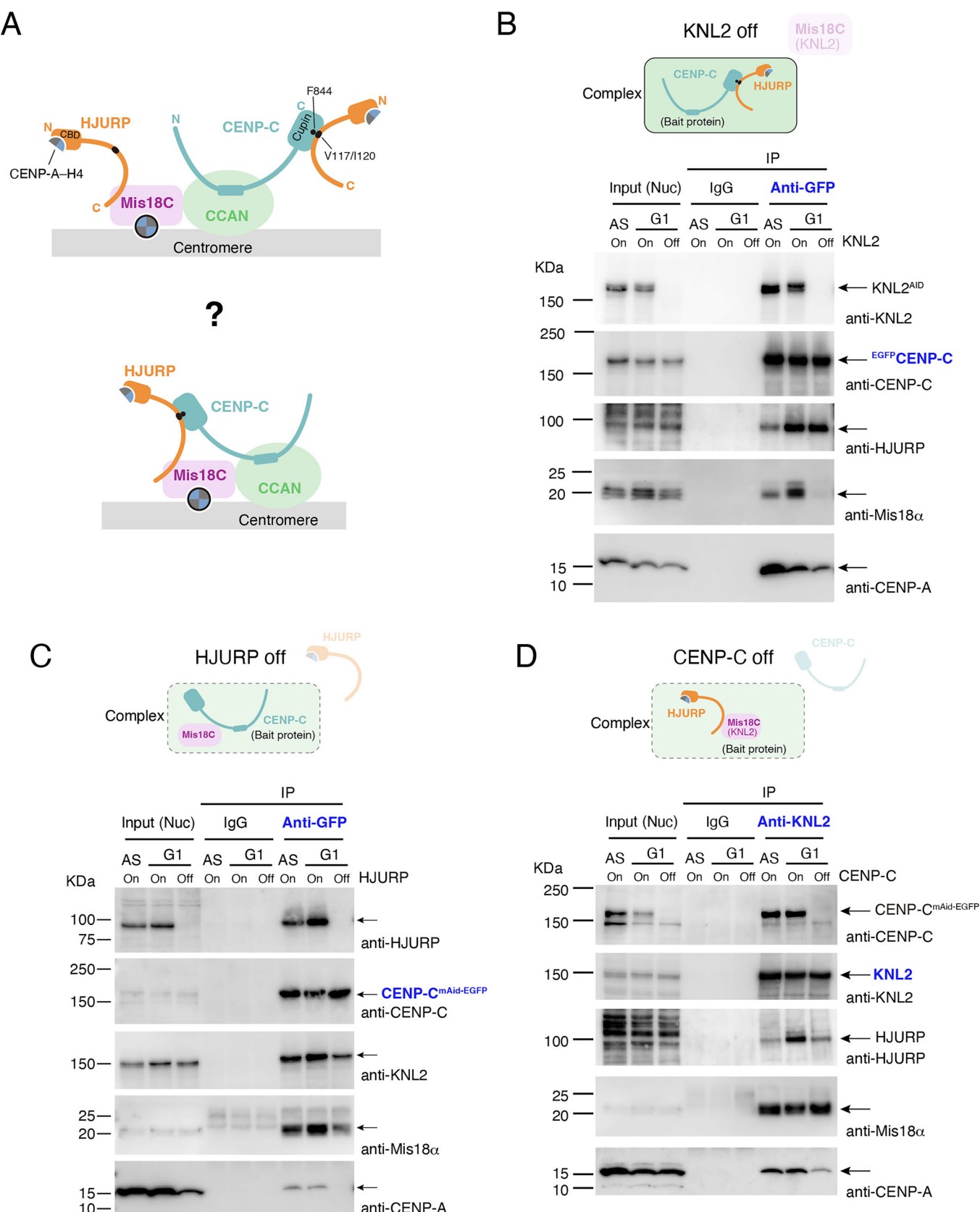

**Figure 6. Co-IP experiments in combined with HJURP, KNL2, or CENP-C knockout cell lines.**

(**A**) Proposed models for the organization of the Mis18 complex (Mis18C), HJURP, and CENP-C. Upper panel: A possible model in which Mis18C-HJURP and CENP-C-HJURP exist as distinct complexes. Lower panel: Alternative model in which Mis18C, HJURP, and CENP-C form a single complex, as the binding sites for Mis18C and CENP-C on HJURP are distinct. (**B**) Co-IP experiments using solubilized chromatin extracts with an anti-GFP antibody in KNL2 On or Off cells expressing EGFP-CENP-C as a bait. Immunoblot analyses were performed to detect KNL2, CENP-C, HJURP, Mis18α, and CENP-A in asynchronous (AS) KNL2 On cells and G1 phase (G1) KNL2 On or Off cells. Each assay was performed twice. Molecular weight markers (kDa) are shown on the left. The co-immunoprecipitated HJURP amount increased in G1 cells compared to AS cells, indicating that the CENP-C-HJURP interaction occurs in G1. HJURP levels remained comparable before and after KNL2 knockout in G1 cells, suggesting that the CENP-C-HJURP interaction in G1 is stable without Mis18C. (**C**) Co-IP experiments using solubilized chromatin extracts with an anti-GFP antibody in HJURP On or Off cells expressing CENP-C-mAid-EGFP as a bait. Immunoblot analyses were performed to detect HJURP, CENP-C, KNL2, Mis18α, and CENP-A in AS HJURP On cells and HJURP On or Off G1 cells. Each assay was performed twice. Molecular weight markers (kDa) are shown on the left. The co-immunoprecipitated KNL2 and Mis18α amounts increased in G1 cells compared to AS cells, indicating that the CENP-C-Mis18C interaction stabilizes in G1. This interaction was reduced upon HJURP knockout in G1, suggesting that the CENP-C-Mis18C interaction is unstable without HJURP. (**D**) Co-IP experiments using solubilized chromatin extracts with an anti-KNL2 antibody in CENP-C On or Off cells. Immunoblot analyses were performed to detect CENP-C, KNL2, HJURP, Mis18α, and CENP-A in asynchronous (AS) CENP-C On cells or CENP-C On or Off G1 cells. Each assay was performed twice. Molecular weight markers (kDa) are shown on the left. The co-immunoprecipitated HJURP amount increased in G1 cells compared to AS cells, indicating that the Mis18C-HJURP interaction occurs in G1. This interaction was reduced upon CENP-C knockout in G1, suggesting that the Mis18C-HJURP interaction is unstable without CENP-C. Source data are available online for this figure.

that the CENP-C-Mis18C association is reduced in HJURP Off cells. We also tested HJURP-Mis18C association in CENP-C Off cells (Fig. 6D). The amount of HJURP co-precipitated with KNL2 was reduced in CENP-C Off cells compared to CENP-C On cells (Fig. 6D), suggesting that CENP-C facilitates a stable interaction of HJURP with Mis18C.

Although the reduced recovery of HJURP by the KNL2-IP (Fig. 6D) appears to support the bottom model in Fig. 6A, the levels of KNL2 at the centromeres did not change in CENP-C KO cells (Fig. 2A). Therefore, both models (top and bottom) in Fig. 6A are possible. Since HJURP and CENP-C facilitate the associations of CENP-C-Mis18C and HJURP-Mis18C, respectively, it appears that all three components (CENP-C, HJURP, and Mis18C) bind stably to centromeres in G1 when they are all associated together (Fig. 6C,D).

### New CENP-A incorporation is completely abolished in KNL2 knockout cells expressing HJURP$^{2E}$ mutant

Finally, we tested whether the deposition of new CENP-A was impaired in cells lacking both the CENP-C and Mis18C pathways. We stably expressed SNAP-CENP-A in cKO-AID-KNL2/cKO-Tet-HJURP cells expressing either EGFP-HJURP$^{WT}$ or EGFP-HJURP$^{2E}$ and in cKO-AID-CENP-C/cKO-Tet-HJURP cells expressing either EGFP-HJURP$^{WT}$ or EGFP-HJURP$^{2E}$ (Fig. EV6A,B). We confirmed that new CENP-A incorporation does not occur in mitosis in these cell lines (Fig. EV6C) and used SNAP-CENP-A intensities in mitosis as a background. As shown in Fig. 7A, the intensity of new CENP-A (SNAP-CENP-A) was significantly reduced in the G1 phase of KNL2 Off cells expressing HJURP$^{WT}$ (cKO-KNL2 cells in the presence of IAA) or KNL2 On cells expressing HJURP$^{2E}$ (cKO-KNL2 cells in the absence of IAA). However, some new CENP-A was detected in these cells (32.2% and 30.1%, respectively). These results are consistent with the reduction of HJURP at centromeres in these cells (Figs. 5B,C,E,F, and EV4G), which suggests that reduced HJURP plays a role in the deposition of new CENP-A. However, SNAP-CENP-A intensities decreased further to 3.8% in KNL2 Off cells expressing HJURP$^{2E}$ (Fig. 7A), indicating that CENP-A incorporation is almost completely abolished in these cells. In contrast, we found that the deposition of new CENP-A in CENP-C Off cells (cKO-CENP-C cells in the presence of IAA) expressing HJURP$^{2E}$ (20.0%) was not reduced further compared to CENP-C On cells (cKO-CENP-C cells in the absence of IAA)

expressing HJURP$^{2E}$ (26.1%) (Fig. 7B). This indicates that the HJURP-CENP-C interaction is not involved in the Mis18C pathway for the deposition of new CENP-A. Based on these findings, we propose that two independent pathways recruit HJURP to centromeres: the CENP-C pathway and the Mis18C pathway. These pathways play a role in the incorporation of new CENP-A into centromeres in chicken DT40 cells.

## Discussion

Analyzing how new CENP-A is incorporated into centromeres is important because CENP-A acts as an epigenetic marker for centromere specification in most organisms. According to the current model, the pre-deposited HJURP-CENP-A-H4 complex recognizes Mis18C on centromeric chromatin via the HJURP-Mis18C interaction. Then, new CENP-A is incorporated into centromeres in vertebrates. However, new CENP-A incorporation was completely abolished in HJURP KO DT40 cells but still occurred in KO cells lacking a Mis18C component. This suggests the existence of another pathway for new CENP-A incorporation. In this study, we demonstrated that CENP-C is involved in a new CENP-A incorporation pathway via its direct binding to HJURP in chicken DT40 cells (Fig. 7C). According to our model, two independent pathways recruit HJURP to centromeres: the CENP-C pathway and the Mis18C pathway. These pathways play a role in the incorporation of new CENP-A into centromeres in chicken DT40 cells.

The HJURP-CENP-C association has been reported in other organisms such as Xenopus and human cells (Flores Servin et al, 2023; Tachiwana et al, 2015). Our data show that the region of chicken HJURP (aa 100-150) clearly binds to the C-terminal region (Cupin dimer region) of chicken CENP-C. While the corresponding region of Xenopus HJURP appears to bind to the Xenopus CENP-C C-terminal region, our in vitro experiments suggest that the human HJURP counterpart does not bind distinctly to the human CENP-C C-terminal region. It is possible that there is a CENP-C pathway for new CENP-A deposition in human cells; however, the CENP-A incorporation mechanisms in human cells may differ from those in chicken and Xenopus cells, either by utilizing different HJURP-CENP-C interfaces or by employing an entirely different mechanism. In fact, the AF3 prediction did not reveal a clear binding region of human HJURP to CENP-C

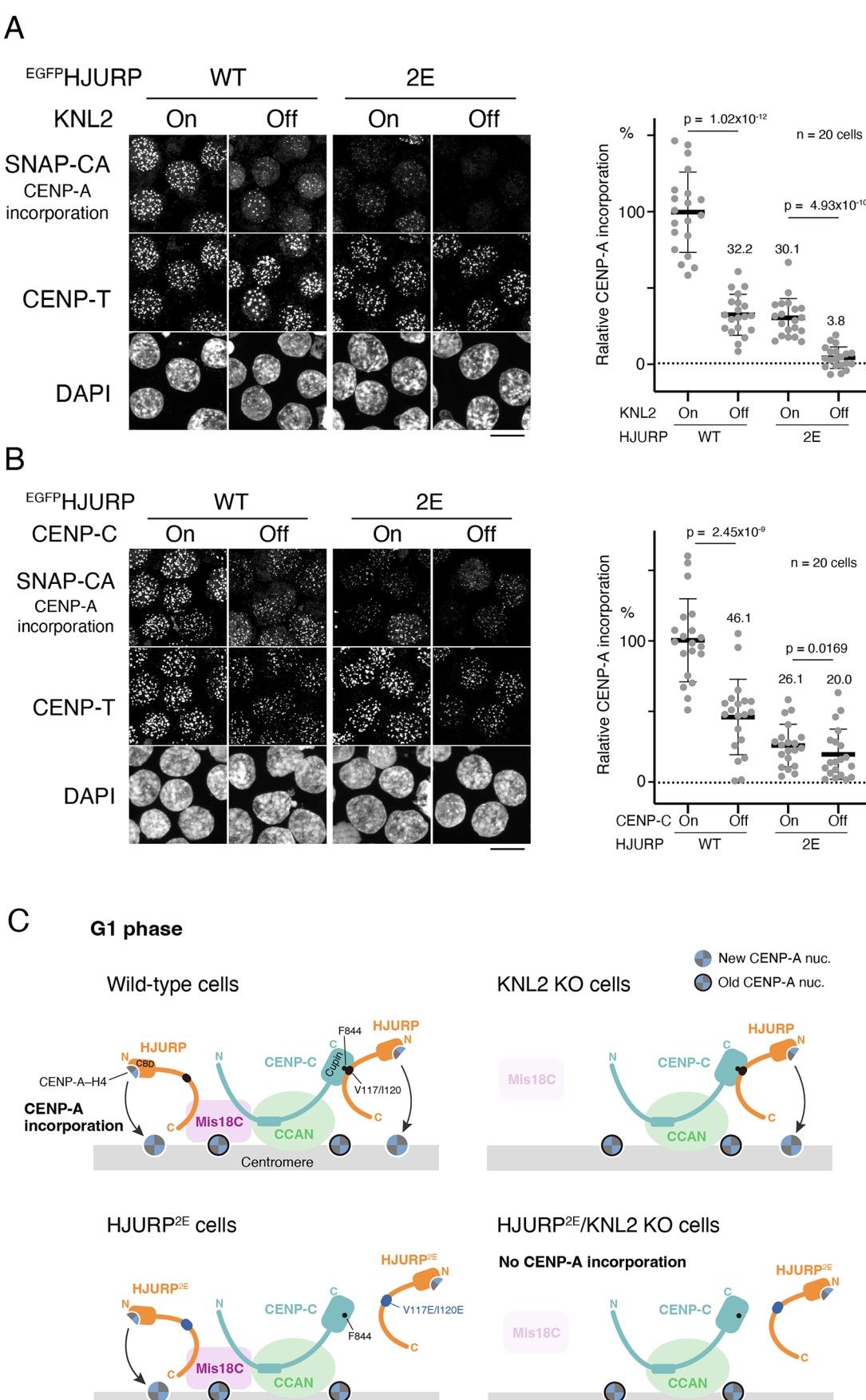

**Figure 7. New CENP-A incorporation was completely abolished in KNL2 KO DT40 cells expressing HJURP[2E].**

(A) Representative images of new CENP-A incorporation (SNAP-CENP-A) in AID-based KNL2 KO cells with Tet-based HJURP-cKO background expressing either EGFP-HJURP (WT) or EGFP-HJURP[2E] (2E) in either the absence (KNL2 On) or the presence (KNL2 Off) of IAA. Tetracycline was added to the culture medium 24 h before SNAP-labeling to replace wild-type HJURP expression with EGFP-fused HJURP variants. CENP-T was used as a centromere marker. DNA was stained with DAPI. Scale bar, 10 μm. The graph shows quantification of CENP-A incorporation into centromeres. Each assay was conducted twice ($n = 20$ cells per condition). The average of SNAP-CENP-A signal intensities in M-phase cells (without Reversine; Fig. EV6C) was subtracted from the corresponding average of intensities in each G1 cell to determine CENP-A incorporation. Results are presented as mean relative intensity ± standard deviation (SD). A two-tailed Student's t-test was performed to compare KNL2 On and Off conditions for both EGFP-HJURP (WT) and EGFP-HJURP[2E] (2E) expressing cells. The p-values are $1.02 \times 10^{-12}$ and $4.93 \times 10^{-10}$ in EGFP-HJURP (WT) or EGFP-HJURP[2E] (2E) expression conditions, respectively. (B) Representative images of SNAP-CENP-A in AID-based CENP-C KO cells with Tet-based HJURP-cKO background expressing either EGFP-HJURP (WT) or EGFP-HJURP[2E] (2E) in either the absence (CENP-C On) or the presence (CENP-C Off) of IAA as described in (A). CENP-T was used as a centromere marker. DNA was stained with DAPI. Scale bar, 10 μm. Each assay was conducted twice ($n = 20$ cells per condition). Results are presented as mean relative intensity ±SD as described in (A). A two-tailed Student's t-test was performed to compare CENP-C On and Off conditions for both EGFP-HJURP (WT) and EGFP-HJURP[2E] (2E) expressing cells. The p-values are $2.45 \times 10^{-9}$ and 0.0169 in EGFP-HJURP (WT) or EGFP-HJURP[2E] (2E) expression conditions, respectively. (C) Summary of CENP-A incorporation pathways in chicken DT40 cells. HJURP mediates new CENP-A incorporation at centromeres through two distinct pathways: the Mis18C and CENP-C pathways. In wild-type cells (upper-left), both pathways function during G1 phase, ensuring robust CENP-A deposition. The HJURP-Mis18C interaction occurs via the HJURP C-terminal portion, while the HJURP-CENP-C interaction involves the HJURP N-terminal region (V117/I120) and CENP-C F844 in the Cupin domain. In KNL2 knockout (KO) cells (upper-right), the Mis18 pathway is compromised, but CENP-A incorporation still occurs through the CENP-C pathway. In cells expressing HJURP[2E] (V117E/I120E), which disrupts the CENP-C-HJURP interaction (HJURP[2E] cells, lower-left), CENP-A incorporation occurs via the Mis18C pathway. However, in cells where both pathways are disrupted (HJURP[2E]/KNL2 KO cells, lower-right), new CENP-A incorporation is completely abolished, demonstrating that these dual pathways function together for new CENP-A deposition in chicken DT40 cells. Source data are available online for this figure.

(Fig. EV3A). The localization of Mis18C in human cells largely depends on CENP-C (McKinley and Cheeseman, 2014; Stellfox et al, 2016). However, this is not the case in chicken and Xenopus cells. In these cells, Mis18C binds directly to the CENP-A nucleosome via a CENP-C-like motif in the Mis18 complex protein KNL2 (French and Straight, 2019; French et al, 2017; Hori et al, 2017; Jiang et al, 2023). In human cells, CENP-C may play a role in the Mis18C pathway for new CENP-A deposition, because human Mis18C cannot localize to centromeres in the absence of CENP-C. Conversely, KNL2 binds directly to the CENP-A nucleosome in chicken and Xenopus cells (French and Straight, 2019; French et al, 2017; Hori et al, 2017; Jiang et al, 2023). Additionally, Mis18C localization remains unchanged in chicken CENP-C KO G1 cells. In these cells, CENP-C may lose its Mis18C-recruiting function in G1 phase and acquire an HJURP-binding function instead. Thus, dual CENP-C and Mis18C pathways for new CENP-A deposition can function in these cells. We propose that these dual pathways exist in organisms where CENP-C and Mis18C localize independently to centromeres. This would ensure robust CENP-A incorporation even if one pathway were compromised. Alternatively, human has CENP-B, an α-satellite binding protein (Masumoto et al, 1989), whereas there are no clear homologs of CENP-B in chicken and Xenopus (Perpelescu and Fukagawa, 2011). Since CENP-B may contribute to the specification of the centromere, the difference in CENP-A incorporation mechanisms between human and chicken may be due to the existence of CENP-B.

HJURP transiently localizes to centromeres during the early G1 phase to facilitate the deposition of new CENP-A. This regulation is important for preventing multiple CENP-A depositions in a cell cycle. CDK phosphorylation inhibits interaction between the HJURP-Mis18 complex during mitosis (Flores Servin et al, 2023; Silva et al, 2012; Stankovic et al, 2017). Our co-IP experiments in G1 cells indicate that the CENP-C-HJURP interaction is enriched in DT40 G1 cells (Fig. 6B). This suggests that there is a regulatory mechanism for CENP-C binding to HJURP that is specific to chicken G1 cells. In Xenopus cells, CDK phosphorylation of HJURP at S220 inhibits the HJURP-CENP-C interaction (Flores Servin et al, 2023). We tested a phosphomimetic mutation of the

corresponding residue of chicken HJURP (S143D), but we did not observe a reduction of the HJURP-CENP-C interaction with the mutant HJURP (Fig. EV3E). While we cannot rule out the existence of other regulatory mechanisms, it is reasonable to hypothesize that phosphorylation of other HJURP regions could inhibit the CENP-C-HJURP interaction. Alternatively, CENP-C could be phosphorylated to regulate this interaction. Either way, an important next step would be to determine how the CENP-C-HJURP interaction is activated specifically in the G1 phase of chicken cells.

We have demonstrated the existence of two pathways, CENP-C and Mis18C, that recruit new CENP-A to chicken centromeres. It appears that Xenopus has similar pathways. Why do these organisms have dual pathways for CENP-A deposition? The binding sites of HJURP to CENP-C and Mis18C are different (N-terminal and C-terminal regions, respectively), so it is possible that the dual pathways function independently. However, our co-IP experiments revealed that the CENP-C-Mis18C association in HJURP KO cells and the Mis18C-HJURP association in CENP-C KO cells decreased compared to control cells (Fig. 6C,D), suggesting that Mis18C and CENP-C associate with each other. However, since no direct binding of Mis18C to CENP-C was observed in the G1 phase of chicken cells, and since the localization of CENP-C at centromeres did not change in KNL2 KO DT40 cells, this association may be mediated by other factors, such as the CCAN complex or HJURP (top model in Fig. 6A). Our current data does not rule out the possibility of the bottom model in Fig. 6A. It is also possible that the two models are present in native centromeres. This close association suggests that, although they appear to be independent in terms of their centromere localization, these pathways function closely within native centromeres. To maintain centromere position, each pathway helps the other achieve stable CENP-A incorporation into centromeres (Figs. 7C and EV7). HJURP-CENP-A-H4 recognizes both CENP-C and Mis18C. Due to their close proximity, CENP-A can be incorporated into the same centromere region to maintain centromere identity. CENP-C has multiple domains including an N-terminal KMN-binding region, a middle region with a CCAN-binding region (Klare et al, 2015; Nagpal et al, 2015), a C-terminal CENP-A-binding region (Petrovic et al, 2016), and oligomerization domains (Hara et al, 2023). These domains allow

CENP-C to function as a hub for centromere formation (Klare et al, 2015). Here, we describe a new CENP-C function. Since CENP-C binds directly to the CENP-A nucleosome (Ali-Ahmad et al, 2019; Allu et al, 2019; Ariyoshi et al, 2021; Kato et al, 2013), it is reasonable that it marks the centromere position. Although CENP-C appears to associate with Mis18C via CCAN in DT40 cells, direct binding to HJURP, independent of Mis18C, may be important for robust CENP-A incorporation. This finding provides new insights into the mechanisms of CENP-A incorporation.

# Methods

**Reagents and tools table**

| Reagent/Resource | Reference or Source | Identifier or Catalog Number |
|---|---|---|
| **Experimental models** | | |
| **Chicken DT40 cell lines** | | |
| DT40_CL18 | Fukagawa Lab | RRID: CVCL_J437 |
| HJURP-cKO | (Perpelescu et al, 2015) (Figs. 4A and EV4A) | N/A |
| EGFP-mAid-HJURP | (Hori et al, 2020) | N/A |
| KNL2-cKO | (Hori et al, 2017) | N/A |
| KNL2-AID (KNL2-cKO, +Tet) | (Hori et al, 2017) (Figs. 2D, EV2D, and EV2G) | N/A |
| HA-SNAP-CENP-A (β-actin-KI)_EGFP-mAid-HJURP | This paper (Figs. 1B,C, EV1A, and EV1E) | N/A |
| HA-SNAP-CENP-A (β-actin-KI)_KNL2-AID (KNL2-cKO, +Tet) | This paper (Figs. 1B,D, EV1B, and EV1E) | N/A |
| HA-SNAP-CENP-A (β-actin-KI)_CENP-C-mAid-EGFP | This paper (Figs. 1B,E, EV1C, and EV1E) | N/A |
| HA-SNAP-CENP-A (β-actin-KI)_Mis18α-cKO | This paper (Fig. EV1F–H) | N/A |
| KNL2-mScarlet (β-actin-KI)_CENP-C-mAid-EGFP_KNL2-cKO | This paper (Fig. 2A) | N/A |
| EGFP-CENP-C (PGK1-KI) [CENP-C (-/-)]_KNL2-AID | This paper (Figs. 2B and 5B) | N/A |
| CENP-C-mAid-mCherry | This paper (Figs. 2D, EV2C, and EV2G) | N/A |
| AID-Mis18α | This paper (Figs. 2D, EV2E, and EV2G) | N/A |
| KNL2-AID_CENP-C-mAid-mCherry | This paper (Figs. 2D, EV2F, and EV2G) | N/A |
| EGFP-HJURP (PGK1-KI)_HJURP-cKO | This paper (Figs. 4A, EV3E, and EV4A) | N/A |
| EGFP-HJURP(2E: V117E/I120E) (PGK1-KI)_HJURP-cKO | This paper (Figs. 4A, EV3E, and EV4A) | N/A |
| EGFP-HJURP (PGK1-KI)_KNL2-AID_HJURP-cKO | This paper (Figs. 5A–F, EV4D, and EV4F) | N/A |
| EGFP-HJURP(2E) (PGK1-KI)_KNL2-AID_HJURP-cKO | This paper (Figs. 5A–F, EV4D, and EV4F) | N/A |
| EGFP-HJURP (PGK1-KI)_CENP-C-AID_HJURP-cKO | This paper (Figs. 5A–F, EV4D, and EV4F) | N/A |
| EGFP-HJURP(2E) (PGK1-KI)_CENP-C-AID_HJURP-cKO | This paper (Figs. 5A–F, EV4D, and EV4F) | N/A |
| CENP-C-mAid-EGFP_HJURP-cKO | This paper (Fig. 6C) | N/A |
| CENP-C-mAid-EGFP | This paper (Figs. 6D and EV2A) | N/A |
| HA-SNAP-CENP-A (β-actin-KI)_EGFP-HJURP (PGK1-KI)_KNL2-AID_HJURP-cKO | This paper (Figs. 7A, EV6A, and EV6C) | N/A |
| HA-SNAP-CENP-A (β-actin-KI)_EGFP-HJURP(2E) (PGK1-KI)_KNL2-AID_HJURP-cKO | This paper (Figs. 7A, EV6A, and EV6C) | N/A |
| HA-SNAP-CENP-A (β-actin-KI)_EGFP-HJURP (PGK1-KI)_CENP-C-AID_HJURP-cKO | This paper (Figs. 7B, EV6B, and EV6C) | N/A |
| HA-SNAP-CENP-A (β-actin-KI)_EGFP-HJURP(2E) (PGK1-KI)_CENP-C-AID_HJURP-cKO | This paper (Figs. 7B, EV6B, and EV6C) | N/A |
| EGFP-CENP-C (PGK1-KI) [CENP-C (-/-)]_AID-Mis18α | This paper (Fig. EV2B) | N/A |
| Halo-HJURP (PGK1-KI)_CENP-C-AID_HJURP-cKO | This paper (Fig. EV5F–I) | N/A |
| Halo-HJURP(8 K) (PGK1-KI)_CENP-C-AID_HJURP-cKO | This paper (Fig. EV5F–I) | N/A |

| Reagent/Resource | Reference or Source | Identifier or Catalog Number |
|---|---|---|
| **Bacterial strains** | | |
| *E.coli.* DH5α | TOYOBO | Cat#: DNA-903 |
| *E. coli* BL21(DE3) | Merck | Cat#: 69450 |
| *E.coli.* Rosetta 2 (DE3) | Merck | Cat#: 71397 |
| **Other cells** | | |
| Drosophila S2 | Gibco | Cat#: R69007 |
| **Recombinant DNA** | | |
| pBluescript II KS (+) | Stratagene | |
| pEGFP-N1 | Clontech | |
| pmScarlet-C1 | Addgene | #85042 |
| pX330 | Addgene | #42230 |
| pX335 | Addgene | #42335 |
| pAID1.2N-mAid-T2A-BS[R] | (Nishimura and Fukagawa, 2017) | N/A |
| pBSKS_ACTB 2k | (Hara et al, 2018) | N/A |
| pX335_ggACTB | (Hara et al, 2018) | N/A |
| pX335_PGK1 | (Cao et al, 2024) | N/A |
| pX330_CENP-C | (Cao et al, 2024) | N/A |
| pX330_KNL2 | (Cao et al, 2024) | N/A |
| pX330_Mis18α | (Cao et al, 2024) | N/A |
| pX330_Ori | This paper | N/A |
| pAID1.2_CENP-C-mAid-EGFP_BS[R] | This paper | N/A |
| pAID1.2_CENP-C-mAid-mcherry_BS[R] | This paper | N/A |
| pAID1.1_CENPC-AID_Puro[R] | (Cao et al, 2024) | N/A |
| pAID1.1_AID-Mis18α_HisD | (Cao et al, 2024) | N/A |
| pAID1.1_KNL2-AID_HisD | (Cao et al, 2024) | N/A |
| pAID1.1_KNL2-AID_Puro[R] | This paper | N/A |
| pBS_Mis18α-KO_HisD | This paper | N/A |
| pBS_Mis18α-KO_Puro[R] | This paper | N/A |
| pUHD_Mis18α | This paper | N/A |
| ptTA_Bleo[R] | (Fukagawa et al, 1999; Fukagawa et al, 2001) | N/A |
| pBS_HA-SNAP-CENP-A_ACTB-KI_EcoGPT | (Hori et al, 2020) | N/A |
| pBS_HA-SNAP-CENP-A_ACTB-KI_BS[R] | This paper | N/A |
| pBS_KNL2-mScarlet_ACTB-KI_EcoGPT | This paper | N/A |
| pBS_EGFP-CENP-C_PGK1-KI_Hyg[R] | This paper | N/A |
| pBS_EGFP-HJURP_PGK1-KI_Hyg[R] | This paper | N/A |
| pBS_EGFP-HJURP[(V117E, I120E)]_PGK1-KI_Hyg[R] | This paper | N/A |
| pBS_Halo-HJURP_PGK1-KI_Hyg[R] | This paper | N/A |
| pBS_Halo-HJURP[8K (F506K, E507K, L509K, L513K, F559K, E560K, I562K, L566K)]_PGK1-KI_Hyg[R] | This paper | N/A |
| p3XFLAG-CMV-10_ggHJURP | (Hori et al, 2020) | N/A |
| pMal-T-Avi-His/BirA | Addgene | #102962 |
| pMal/BirA_ggHJURP[100-150] | This paper | N/A |
| pMal/BirA_ggHJURP[100-150(V117E, I120E)] | This paper | N/A |
| pMal/BirA_ggHJURP[100-150(V117A, I120A)] | This paper | N/A |
| pMal/BirA_ggHJURP[100-150(S143D)] | This paper | N/A |

| Reagent/Resource | Reference or Source | Identifier or Catalog Number |
|---|---|---|
| pMal/BirA_hsHJURP[137-192] | This paper | N/A |
| pGEX-6P-1 | Cytiva | Cat#: 28954648 |
| pGEX-6P-1_ggCENP-C[677-864] | (Hara et al, 2023) | N/A |
| pGEX-6P-1_ hsCENP-C[760-943] | (Hara et al, 2023) | N/A |
| pGEX-6P-1_ggHJURP[480-580] | This paper | N/A |
| pETDuet-1 | Novagen | Cat#: 71146 |
| pETDuet-1_His$_6$-ggMis18α:ggMis18β | This paper | N/A |
| pETDuet-1_His$_6$-ggMis18α[38-204]:MBP-ggMis18β[45-213] | This paper | N/A |
| **Antibodies** | | |
| Rabbit polyclonal anti-chicken CENP-A | Fukagawa Lab (Hori et al, 2008) | RRID: AB_2665547 |
| Rabbit polyclonal anti-chicken CENP-C | Fukagawa Lab (Fukagawa et al, 1999) | RRID: AB_2665548 |
| Rabbit polyclonal anti-chicken CENP-T | Fukagawa Lab (Hori et al, 2008) | RRID: AB_2665551 |
| Rabbit polyclonal anti-chicken HJURP | Fukagawa Lab (Perpelescu et al, 2015) | RRID: AB_ 3716533 |
| Rabbit polyclonal anti-chicken KNL2 | Fukagawa Lab (Hori et al, 2017) | RRID: AB_ 3716534 |
| Rabbit polyclonal anti-chicken Mis18α | Fukagawa Lab (Hori et al, 2017) | RRID: AB_ 3716535 |
| Rabbit anti-H2Av (Spike-in antibody) | Active Motif | Cat#: 61686 |
| Mouse monoclonal anti-alpha tubulin | SIGMA | Cat#: T9026 |
| Cyclin B2 Rabbit polyclonal antibody | Proteintech | Cat#: 21644-1-AP; RRID: AB_10755304 |
| Rabbit polyclonal anti-GFP | MBL | Cat#: 598 |
| Rabbit polyclonal anti-HA | SIGMA | Cat#: H6908 |
| HRP-conjugated Goat anti-Rabbit IgG | Jackson ImmunoResearch | Cat#: 111-035-003; RRID: AB_2313567 |
| HRP-conjugated Rabbit anti-mouse IgG | Jackson ImmunoResearch | Cat#: 315-035-003; RRID: AB_2340061 |
| Rabbit TrueBlot: HRP-conjugated Mouse anti-Rabbit IgG | ROCKLAND | Cat#: 18-8816-31 |
| Cy3-conjugated Mouse anti-rabbit IgG | Jackson ImmunoResearch | Cat#: 211-165-109; RRID: AB_2339158 |
| Alexa 647-conjugated Goat anti-rabbit IgG, Fab | Jackson ImmunoResearch | Cat#: 111-607-003; RRID: AB_2338084 |
| FITC-conjugated Goat anti-Mouse IgG | Jackson ImmunoResearch | Cat#: 115-095-003; RRID: AB_2338589 |
| Cy3-conjugated Goat anti-Mouse IgG | Jackson ImmunoResearch | Cat#: 115-165-003; RRID: AB_2338680 |
| Rabbit IgG | SIGMA | Cat#: 15006 |
| anti-BrdU (B44) | BD | Cat#: 347580 |
| **Oligonucleotides and other sequence-based reagents** | | |
| Ori-t sgRNA Top: CACCGtggaacg aaaactcacgtta | This paper | N/A |
| Ori-b sgRNA Bottom: AAACtaacgtga gttttcgttccaC | This paper | N/A |
| CENP-C-F cDNA: ATGGCGGAGC GCTTGGATCA | This paper | N/A |
| CENP-C-R cDNA: TGGGGAAGA GGATCCTGACAA | This paper | N/A |
| Mis18α-F cDNA: atggcgggc ttgttgtatccgttgt | This paper | N/A |
| Mis18α-R cDNA: TCAAGCGTTA CTGTGAAGAGA | This paper | N/A |
| Mis18α-3probe-F Genome DNA: GTCCCGCT CATTTGGGACGCGGAT | This paper | N/A |
| Mis18α-3probe -R Genome DNA: GTGCTGCCCATGCATTCCTAGA | This paper | N/A |

| Reagent/Resource | Reference or Source | Identifier or Catalog Number |
|---|---|---|
| HJURP-F<br>cDNA: ATGGCATT<br>GGATCTGGACGAA | This paper | N/A |
| HJURP-R<br>cDNA: TCTCGAGCTAT<br>AAAACAATGCCAT | This paper | N/A |
| HJURP$^{2E}$-F<br>Mutant cDNA: AGGTTCA<br>AgaaGATGTGgaaGTACAA<br>GATAATGAAAGAAGA<br>ATTCCCAAATGGA | This paper | N/A |
| HJURP$^{2E}$-R<br>Mutant cDNA: ttcCACATCttcT<br>TGAACCTGTACTTT<br>TTCTCGCGCTTTCACAT | This paper | N/A |
| HJURP$^{100-150}$-F<br>cDNA: tcacatatgtccatg<br>CAGAATCTGGAT<br>TTAGATGATGTGA | This paper | N/A |
| HJURP$^{100-150}$-R<br>cDNA: acttccaggcccatg<br>GCCTATCATTTCT<br>CGCACTGGTGAA | This paper | N/A |
| HJURP$^{100-150(2E)}$-F<br>Mutant cDNA: AAAGTACAGG<br>TTCAAGAAGATGTGGAA<br>GTACAAGATAATGAA | This paper | N/A |
| HJURP$^{100-150(2E)}$-R<br>Mutant cDNA: TTCATTATCTT<br>GTACTTCCACATCTT<br>CTTGAACCTGTACTTT | This paper | N/A |
| HJURP$^{100-150(2A)}$-F<br>Mutant cDNA: AAAGTACAGGTT<br>CAAGCAGATGTGGCA<br>GTACAAGATAATGAA | This paper | N/A |
| HJURP$^{100-150(2A)}$-R<br>Mutant cDNA: TTCATTATCTT<br>GTACTGCCACATCTGC<br>TTGAACCTGTACTTT | This paper | N/A |
| HJURP$^{100-150(SD)}$-F<br>Mutant cDNA: TCAAGTCATTT<br>CACTGACCCAGT<br>GCGAGAAATG | This paper | N/A |
| HJURP$^{100-150(SD)}$-R<br>Mutant cDNA: CATTTCTCG<br>CACTGGGTCAGTG<br>AAATGACTTGA | This paper | N/A |
| hsHJURP$^{137-192}$-F<br>cDNA: tgtcCATGGTGC<br>CTCAAAGCCCT<br>TTGAAAAATG | This paper | N/A |
| hsHJURP$^{137-192}$-R<br>cDNA: ggcccatgTCC<br>GGGGGCAGGCA<br>CGGCAGGTGA | This paper | N/A |
| HJURP$^{4K01}$-F<br>Mutant cDNA: aagaagAA<br>AaagTACTACAAAaagT<br>GTTCCAGAGGAAGCCA<br>AAAGCCTTTGACA | This paper | N/A |
| HJURP$^{4K01}$-R<br>Mutant cDNA: TTGTAGTActtTT<br>TcttcttCGCATCTTCGTGTT<br>TCTGGGGTGTCTTTGA | This paper | N/A |
| HJURP$^{4K02}$-F<br>Mutant cDNA: agTATCAGCAGaagT<br>CTAGTGAGGTGGT<br>TCCAAAAATTCC<br>TGGTTTCCA | This paper | N/A |
| HJURP$^{4K02}$-R<br>Mutant cDNA: ACTAGAActtC<br>TGCTGATActtCTTcttcttTG<br>CAACGTTATACT<br>GTATACTGGA | This paper | N/A |

| Reagent/Resource | Reference or Source | Identifier or Catalog Number |
|---|---|---|
| HJURP<sup>480-580</sup>-F<br>cDNA: TCCAGGGGCCC<br>CTGGGATCCTTTCA<br>ACGCAGACACTCATTTTCTTCA | This paper | N/A |
| HJURP<sup>480-580</sup>-R<br>cDNA: CTCGAGTCGACCCGGGAATT<br>TTAGTGGTGATGGTGAT<br>GATGAGCTCTTTGGAAACCAGGAA | This paper | N/A |
| ggMis18α cDNA (optimized for E.coli expression):<br>GGATCCGTCCGCTGCTGGTCG<br>TCTGGAAGTTCTGTTTCAAG<br>GTCCGGCTGGTCTGCTGTATC<br>CGCTGTCGGGTGGTGCCGTCG<br>GTCTGGACTCTAGCCTGAGCAT<br>GCTGGAAGCGACCATTCCGCGT<br>GAAGCACTGCCGGTGGAACGT<br>CCGCGCGCGAACGGCAAAGA<br>AGAAGAAGAAGAAAACGATG<br>AAAATCTGCCGATGGTTTTT<br>CTGTGCGGCGGTTGTAAACGC<br>CCGGTGGGTGATACCCTGAGC<br>TGGGTTGCGAACGATGAAGA<br>AACCGAATGCATTCTGCTGC<br>GTAGCGCCAGCAGCAATGT<br>GAGCGTTGATAAAGAACA<br>GAAACTGAGCAAACGTCC<br>GGGCGAATGCGGTTGTAT<br>GGTGGAAACCTTTTTCT<br>GCAGCGGCTGTAGCATG<br>ACCCTGGGTAGCATCTATC<br>GTTGCACCCCGAAACATC<br>TGGATTATAAACGTGACCTG<br>TTTTGCTTTAGCGTGGGCG<br>CGATTGAAAGCTATATCCTG<br>GGTAGCAGCGAAAAACAG<br>GCCGTTACCGATGAAGAA<br>CCGCTGACCCTGGAAAG<br>CCGTGCGGCACTGGAAG<br>AAGCGCTGAAACGCGCC<br>AACACCATCCTGAATGCC<br>CTGGAAGCACGCCTGGC<br>TACGGTTGAATCTCGCA<br>TTGATAGTCTGCACAGT<br>AACGCATAAGTCGAC | This paper;<br>Gene synthesis (GenScript) | N/A |
| ggMis18β cDNA (optimized for E.coli expression):<br>CATATGGACTATAAAGATG<br>ATGATGATAAATCGCCGGG<br>CACGGAAGACCTGTATTTTCA<br>ATCGGGCTCGGTTCGTCGTC<br>AACTGCAACGCTTCTTTGAAG<br>AACCGCAGGTTCGTGGCACCA<br>TTGTGGTTGAACGCCCGCTGA<br>GCAGCGAACCGCCGCCGGCAC<br>CGCCGCCGACCAGCGAAGTGC<br>CGCGTTGCTATGAACTGCGCC<br>CGGAAGATTGTGCGGTTTTTC<br>AGTGCCGTGGCTGTTGGGCC<br>GTGCTGGGTGATAGCCTGCA<br>TCTGTGCGCGCTGGAAGAAC<br>AGCGTCTGGGCCTGCTGGTG<br>TGCCTGCGTGTTACCAGCA<br>ACGTGCTGTGTGAAGAAGC<br>CCTGATGGTTGGCCTGGA<br>AGGTGCGCTGATGGGTTG<br>TGCCTATAATACCCTGAG<br>CTGCCAGAGCTGTGGCA<br>GCGTGGTTGGTTTTGTGC<br>TGTATAGCGCGTTTCGTG<br>ATCTGGCCCATCTGCGTG<br>GCTTTTTCTGCTTTTTCA<br>AAAACAGCATCTTTTGTT<br>ATCTGCTGAAAAGCAAA<br>ATGATCATCGAAGCGAGC<br>AAAGTTAAATTTCCGGCCG<br>TGACCCTGAAAGATGAAA<br>TGGAAAAACTGAAAGAA<br>AGCCTGGTTATGGCCCAC<br>ATGCGCGTGGAACTGCTG<br>ATGAAGAAAGTGGAACA<br>ACTGAAACAAAATAACG<br>TCGCTGAAAATCGCGCA<br>CTGCACCACATGCAACT<br>GCACTGACCTAGG | This paper; Gene synthesis (GenScript) | N/A |

| Reagent/Resource | Reference or Source | Identifier or Catalog Number |
| --- | --- | --- |
| M18α-del$^{1-37}$-F: AAGTTCTGT TTCAAGGTCCGGCGA ACGGCAAAGAA GAAGAAGA | This paper | N/A |
| M18α-del$^{1-37}$-R: CGGACCTTG AAACAGAACTT | This paper | N/A |
| M18β-del$^{1-44}$-F: TGTATTTTC AATCGGGCTCGGAACTGCG CCCGGAAGATTGT | This paper | N/A |
| M18β-del$^{1-44}$-R: CGAGCC CGATTGAAAAT ACAGGTCT | This paper | N/A |
| **Chemicals, Enzymes and other reagents** | | |
| Quick-CBB PLUS | Wako | Cat#: 178-00551 |
| Dulbecco's modified Eagle medium (DMEM) | Nacalai Tesque | Cat#: 08459-64 |
| Schneider's Drosophila Medium | Thermo Fisher | Cat#: 21720-024 |
| Fetal bovine serum | Biosera | Cat#: FB1290/500 |
| Chicken serum | Thermo Fisher | Cat#: 16110082 |
| 2-Mercaptoethanol | SIGMA | Cat#: M3148 |
| Penicillin-Streptomycin | Thermo Fisher | Cat#: 15140-122 |
| Nocodazole | SIGMA | Cat#: M1404 |
| Reversine | SIGMA | Cat#: R3904 |
| Puromycin (Puro) | TAKARA | Cat#: Z1305N |
| L-Histidinol dihydrochloride (His) | SIGMA | Cat#: H6647 |
| Hygromycin B (Hyg) | Wako | Cat#: 087-06152 |
| Blasticidin S hydrochloride (BS) | Wako | Cat#: 029-18701 |
| Zeocin | Thermo Fisher | Cat#: R25001 |
| Xanthine | SIGMA | Cat#: 1002581797 |
| Mycophenolic acid | Wako | Cat#: 138-11003 |
| 3-Indole acetic acid (IAA) | Wako | Cat#: 090-07123 |
| SNAP-Cell Block | New England Biolabs | Cat#: S9106S |
| SNAP-Cell TMR-Star | New England Biolabs | Cat#: S9105S |
| Dynabeads protein G | Thermo Fisher | Cat#: 10009D |
| NP-40 alternative | ChemCruz | Cat#: sc-281108 |
| BSA | Equitech-Bio Inc | Cat#: BAC62 |
| 16% Paraformaldehyde | Electron Microscopy Sciences | Cat#: 15710 |
| Triton X-100 | SIGMA | Cat#: T8787 |
| Tween 20 | Nacalai Tesque | Cat#: 28353-85 |
| DL-dithiothreitol (DTT) | SIGMA | Cat#: D0632 |
| TCEP | Nacalai Tesque | Cat#: 07277-16 |
| cOmplete protease inhibitor (EDTA-free) | Roche | Cat#: 05056489001 |
| Trichostatin A | Wako | Cat#: 209-17563 |
| Purvalanol A | Cayman Chemical | Cat#: 14579 |
| MinElute PCR purification kit | Qiagen | Cat#: 28006 |
| Signal Enhancer HIKARI for Western Blotting and ELISA | Nacalai Tesque | Cat#: 02270-81 |
| Can Get Signal immun ostain Solution A | TOYOBO | Cat#: NKB-501 |
| Skim-milk | BD | Cat#: 232100 |
| ECL Prime | GE Healthcare | Cat#: RPN2236 |
| In-Fusion® HD Cloning Kit | Takara | Cat#: 639649 |
| FlexAble 2.0 CoraLite® Plus 555 Antibody Labeling Kit for Rabbit IgG | Proteintech | Cat#: KFA502 |

| Reagent/Resource | Reference or Source | Identifier or Catalog Number |
|---|---|---|
| FlexAble CoraLite® Plus 647 Antibody Labeling Kit for Rabbit IgG | Proteintech | Cat#: KFA003 |
| FlexAble CoraLite® Plus 488 Antibody Labeling Kit for Rabbit IgG | Proteintech | Cat#: KFA001 |
| 4′,6-diamidino-2-phenylindole (DAPI) | Roche | Cat#: 10236276001 |
| VECTASHIELD PLUS Antifade Mounting Medium | Vector Laboratories | Cat#: H-1900 |
| FluoroTrans PVDF Membrane | PALL Corporation | Cat#: PVM020C-099 |
| BiodyneRB 0.45 µm | PALL Corporation | Cat#: 60207 |
| Amylose resin columns | New England Biolabs | Cat#: E8021S |
| HiLoad 16/600 Superdex 75 pg column | Cytiva | Cat#: 28989333 |
| HiLoad 16/600 Superdex 200 pg column | Cytiva | Cat#: 28989335 |
| Glutathione Sepharose 4 Fast Flow columns | Cytiva | Cat#: 17513202 |
| 5-ml Hi-Trap SP HP column | Cytiva | Cat#: 17115201 |
| 5-ml Hi-Trap Q HP column | Cytiva | Cat#: 17115401 |
| Superdex 200 Increase 10/300 GL column | Cytiva | Cat#: 28990944 |
| Bromophenol blue | MP biomedicals | Cat#: 101123 |
| RNase A | Thermo Fisher | Cat#: EN0531 |
| Proteinase K | SIGMA | Cat#: P2308 |
| KAPA Hyper Prep Kit for Illumina | KAPA BIOSYSTEMS | Cat#: KK8504 |
| Janelia Fluor 646 HaloTag Ligand | Promega | Cat#: GA1120 |
| BrdU | SIGMA | Cat#: B5002 |
| Propidium iodide | SIGMA | Cat#: P4170 |
| **Recombinant proteins** | | |
| Recombinant MBP-AviTag-His$_6$ | Tonia Rex Lab | N/A |
| Recombinant MBP-ggHJURP$^{100-150}$ | This paper | N/A |
| Recombinant MBP-ggHJURP$^{100-150 (V117E, I120E)}$ | This paper | N/A |
| Recombinant MBP-ggHJURP$^{100-150 (V117A, I120A)}$ | This paper | N/A |
| Recombinant MBP-ggHJURP$^{100-150 (S143D)}$ | This paper | N/A |
| Recombinant MBP-hsHJURP$^{137-192}$ | This paper | N/A |
| Recombinant GST | Cytiva | N/A |
| Recombinant GST-ggCENP-C$^{677-864}$ | (Hara et al, 2023) | N/A |
| Recombinant GST-hsCENP-C$^{760-943}$ | (Hara et al, 2023) | N/A |
| Recombinant GST-ggHJURP$^{480-580}$ | This paper | N/A |
| Recombinant His$_6$-ggMis18α$^{38-204}$: MBP-ggMis18β$^{45-213}$ | This paper | N/A |
| Recombinant human rhinovirus 3C protease | In-house | N/A |
| **Software** | | |
| NIS-elements AR v5.42.01 | Nikon | RRID: SCR_014329 |
| Imaris (9.1.2 Build 45902 for x64) | Bitplane | RRID: SCR_007370 |
| Fiji (ImageJ2 v2.16.0/1.54p) | (Schindelin et al, 2012) | RRID: SCR_002285 |
| GraphPad Prism v9.5.1 | GraphPad Software | RRID: SCR_002798 |
| Image Lab (6.1.0) | Bio-Rad | RRID: SCR_014210 |
| PyMOL v3.0 | Schrödinger | RRID: SCR_000305 |
| AlphaFold3 Server | (Abramson et al, 2024) | RRID: SCR_025885 |
| Fastp v0.20.0 | https://github.com/OpenGene/fastp | RRID: SCR_016962 |
| Bowtie 2 v2.4.2 | http://bowtie-bio.sourceforge.net/bowtie2/index.shtml | RRID: SCR_016368 |
| deepTools v3.5.0 | https://deeptools.readthedocs.io/en/develop/ | RRID: SCR_016366 |

| Reagent/Resource | Reference or Source | Identifier or Catalog Number |
|---|---|---|
| Integrative Genomics Viewer v2.19.2 | http://www.broadinstitute.org/igv/; (Robinson et al, 2011) | RRID: SCR_011793 |
| FACSDiva v9.0 | BD | RRID: SCR_001456 |
| Other | | |
| Deposited Data | | |
| Fastq data for ChIP-seq | This paper https://www.ncbi.nlm.nih.gov/sra/?term=PRJNA1235856 | NCBI BioProject accession number: PRJNA1235856 |

## Cell culture

Chicken DT40 cells were cultured in Dulbecco's modified Eagle medium (DMEM; Nacalai Tesque) supplemented with 10% fetal bovine serum (FBS; Biosera), 1% chicken serum (Thermo Fisher), 10 μM 2-mercaptoethanol (Sigma), and penicillin/streptomycin (final: 100 unit/ml and 100 μg/ml, respectively) (Thermo Fisher) at 38.5 °C with 5% $CO_2$. For the auxin-induced degron (AID or mini-AID; mAid) based conditional knockout (cKO) cell lines, degradation of each AID- or mAid-tagged protein was induced by the addition of 3-indole acetic acid (IAA; Wako) (final concentration 500 μM) to the culture medium. For the tetracycline-induced promoter shut off (Tet-off) conditional knockout (cKO) HJURP or Mis18α cell lines, each gene under control of the Tet-off promoter was repressed by the addition of tetracycline (Tet, Wako) (final concentration 2 μg/ml) to the culture medium. To examine cell growth for each HJURP-cKO cell line expressing EGFP-tagged Wild-type HJURP or HJURP[2E (V117E/I120E)], Tet was added to the culture medium at time 0, and cell concentrations were measured twice per time point every 24 h using a Countess II Automated Cell Counter (Thermo Fisher) for 8 days. Drosophila S2 cells (Gibco) were cultured in Schneider's Drosophila medium (Thermo Fisher) supplemented with 10% FBS and penicillin/streptomycin (final: 50 unit/ml, 50 μg/ml, respectively) at 28 °C.

## Plasmid constructions

For CRISPR-Cas9 genome editing in DT40 cells, we utilized the previously described sgRNA and SpCas9 expression vectors targeting CENP-C (pX330_CENP-C), Mis18α (pX330_Mis18α), and KNL2 (pX330_KNL2) (Cao et al, 2024). We also created pX330_Ori to linearize AID expression vectors. To express AID-tagged proteins, we used pAID1.2N-mAid-T2A-BSR (Nishimura and Fukagawa, 2017) as a backbone. CENP-C cDNA was amplified by PCR and cloned with EGFP or mCherry into this vector using the In-Fusion HD Cloning Kit (Takara) to generate pAID1.2_-CENP-C-mAid-EGFP_BS[R] or pAID1.2_CENP-C-mAid-mCherry_BS[R]. For other AID constructs, we utilized pAID1.1_AID-Mis18α_HisD and pAID1.1_KNL2-AID_HisD (Cao et al, 2024) or created pAID1.1_KNL2-AID_Puro[R] and pAID1.1_-CENP-C-AID_Puro[R] by replacing the histidinol dehydrogenase gene (HisD) marker with the Puromycin resistant gene (Puro[R]) using PCR and the In-Fusion HD Cloning Kit.

For the tetracycline-responsive Mis18α conditional knockout (cKO), targeting vectors for the first and second alleles of the Mis18α gene were constructed. A genomic DNA fragment of the Mis18α gene was digested with *Afl*II or *Bam*HI to isolate the 4.2 kb or 4.3 kb fragment for the 5′- or 3′-arm of the targeting vector,

respectively. The 5′- and 3′-arm fragments and selection marker (HisD or Puro[R]) with β-actin promoter were prepared with appropriate restriction-enzyme digestion and were sequentially cloned into the pBluescript II KS (+) vector (Stratagene) to create the targeting vectors, pBS_Mis18α-KO_HisD and pBS_Mis18a-KO_Puro[R], respectively. The Mis18α coding sequence was cloned into pUHD10-3 (Gossen and Bujard, 1992) to generate pUHD_-Mis18α. For tTA expression, ptTA_Bleo[R] (Fukagawa et al, 1999; 2001) was used.

For gene expression to maintain equal levels among each cell line, we performed knock-ins at the β-actin or PGK1 loci using CRISPR-Cas9 nickase (D10A) with pX335_ggACTB (Hara et al, 2018) or pX335_PGK1 (Cao et al, 2024), respectively. We constructed pBS_HA-SNAP-CENP-A_ACTB-KI_BS[R] as a knock-in vector by replacing the *E. coli* xanthine-guanine phosphoribosyltransferase gene (EcoGPT) in pBS_HA-SNAP-CENP-A_ACTB-KI_EcoGPT (Hori et al, 2020) with the Blasticidin resistant gene (BS[R]). For KNL2 visualization, we created pBS_KNL2-mScarlet_ACTB-KI_EcoGPT using pBSKS_ACTB 2k (Hara et al, 2018) as a backbone. For EGFP-CENP-C expression, we used pBS_EGFP-CENP-C_PGK1-KI_Hyg[R] (Cao et al, 2024). For EGFP-HJURP or -HJURP[(2E: V117E/I120E)] expression, we generated pBS_EGFP-HJURP_PGK1-KI_Hyg[R] or pBS_EGFP-HJURP[2E]_PGK1-KI_Hyg[R] by replacing CENP-C in the above PGK1 knock-in vector with wild-type HJURP or HJURP[2E] cDNA. For Halo-HJURP or -HJURP[(8K: F506K, E507K, L509K, L513K, F559K, E560K, I562K, L566K)] expression, we generated pBS_Halo-HJURP_PGK1-KI_Hyg[R] by replacing EGFP of the EGFP-HJURP_PGK1 knock-in vector with Halo and pBS_Halo-HJURP[8K]_PGK1-KI_Hyg[R] by further replacing wild-type HJURP in the Halo-HJURP_PGK1 knock-in vector with HJURP[8K] cDNA.

For recombinant protein expression, we created MBP-fused ggHJURP fragments by amplifying HJURP[100-150] or mutant variants [HJURP[100-150(2E)], HJURP[100-150(2A: V117A/I120A)], HJURP[100-150(SD: S143D)]] from p3XFLAG-CMV-10_ggHJURP (Hori et al, 2020) by PCR and inserting them into pMal-T-Avi-His/BirA (Addgene #102962) using the In-Fusion HD Cloning Kit. Human HJURP[137-192] was similarly cloned from HJURP cDNA derived from HeLa cells. For GST-fused CENP-C fragments, we used the previously described pGEX-6P-1_ggCENP-C[677-864] and pGEX-6P-1_hsCENP-C[760-943] (Hara et al, 2023). For GST-fused HJURP fragments, we amplified HJURP[480-580] from p3XFLAG-CMV-10_ggHJURP by PCR and inserted into pGEX-6P-1 (Cytiva) using the In-Fusion HD Cloning Kit. For expression of Mis18α-Mis18β complex, synthesized ggMis18α and ggMis18β gene fragments were inserted into pETDuet-1 (Novagen) to yield pETDuet-1_His₆-ggMis18α:ggMis18β. To increase the solubility and complex stability, the N-terminal regions (residues 1–37 of Mis18α and 1–44 of Mis18β) were removed from pETDuet-1_His₆-ggMis18α:ggMis18β, and a maltose-binding protein (MBP) coding fragment was further

inserted at the N-terminus of Mis18β$^{45-213}$ to yield pETDuet-1_His$_6$-ggMis18α$^{38-204}$:MBP-ggMis18β$^{45-213}$. All constructed plasmid DNAs were verified by restriction-enzyme digestion and sequencing.

## Cell lines and transfection

For AID-based conditional knockout (cKO) generation, DT40 cells were transfected with AID-tagged expression vectors (pAID1.1_AID-Mis18α_HisD, pAID1.2_CENP-C-mAid-EGFP or -mCherry_BS$^R$) along with corresponding sgRNA vectors (pX330_Mis18α or pX330_CENP-C and pX330_Ori) using the Neon Transfection System (Thermo Fisher). Transfected cells were selected with appropriate antibiotics: 1 mg/ml L-Histidinol dihydrochloride (Sigma) for HisD, 10 μg/ml Blasticidin S hydrochloride (Wako) for BS$^R$. Gene disruption was confirmed by genomic PCR, sequencing, and Immunoblotting. For double conditional knockouts (dKO), we transfected AID-tagged expression vectors (pAID1.1_CENPC-AID_Puro$^R$ or pAID1.2_-CENP-C-mAid-EGFP_BS$^R$ for CENP-C KO, pAID1.1_KNL2-AID_Puro$^R$ for KNL2 KO) along with corresponding sgRNA vectors (pX330_CENP-C or pX330_KNL2 and pX330_Ori) into previously established HJURP-cKO (Perpelescu et al, 2015), KNL2-cKO (Hori et al, 2017), or CENP-C-cKO (CENP-C-mAid-mCherry) cells and selected with 0.5 μg/ml Puromycin (TAKARA) for Puro$^R$ or 10 μg/ml Blasticidin S hydrochloride for BS$^R$ to generate CENP-C/HJURP-dKO, KNL2/HJURP-dKO, or CENP-C/KNL2-dKO cell lines.

For Tet-based Mis18α-cKO, DT40 cells were electroporated with pBS_Mis18α-KO_HisD using Gene Pulser II (Bio-Rad) for the first allele of the Mis18α gene targeting. After selection with 1 mg/ml L-Histidinol dihydrochloride, cells were transfected with pUHD_-Mis18α and ptTA_Bleo$^R$, selected with 1 mg/ml Zeocin (Thermo Fisher), and then transfected with pBS_Mis18α-KO_Puro$^R$ and selected with 0.5 μg/ml Puromycin for the second allele of the Mis18α gene targeting to generate Mis18α knockout cells. Gene disruption was confirmed by Southern blotting hybridized with a 3′ side probe against the *Apa*LI-digested genomic DNA of each targeted-clone and Immunoblotting.

For gene knock-ins at β-actin or PGK1 loci, cells were transfected with appropriate knock-in constructs and corresponding sgRNA vectors (pX335_ggACTB or pX335-PGK1). For HA-SNAP-CENP-A or KNL2-mScarlet expression from the β-actin locus, cells were selected with either 25 μg/ml Mycophenolic acid (Wako) plus 125 μg/ml Xanthine (Sigma) for the EcoGPT marker or Blasticidin S hydrochloride for the BS$^R$ marker. For EGFP-CENP-C, EGFP-HJURP, or EGFP-HJURP$^{2E}$, Halo-HJURP, or Halo-HJURP$^{8K}$ expression from the PGK1 locus, cells were selected with 2.5 mg/ml Hygromycin B (Wako). In the case of EGFP-CENP-C expressing cell lines, the endogenous CENP-C gene was disrupted with the sgRNA for CENP-C KO (pX330_CENP-C) to replace the endogenous CENP-C with an EGFP-tagged one. All generated cell lines were verified by genomic PCR, sequencing, and/or Immunoblotting with appropriate antibodies. All cell lines used in figures are listed in Reagents_Tools_Table.

## Immunoblot analysis

Whole cell extracts were prepared from DT40 wild-type Cl18 cells and AID- or Tet-based cKO cells cultured in the absence or presence of IAA or Tet, respectively. Extracts were separated by SDS-PAGE using 5–20% or 10–20% SuperSep Ace gels (Wako)

and transferred onto PVDF membranes (FluoroTrans, PALL). Immunoprecipitated samples were also blotted onto PVDF membranes. Membranes were blocked with 5% skim milk (BD) at room temperature for 10 min, then probed with the following primary antibodies: rabbit anti-GFP (1:20,000, MBL), rabbit anti-CENP-A (1:5000), mouse anti-α-tubulin (1:10,000, SIGMA), rabbit anti-HJURP (1:20,000), rabbit anti-HA (1:5000, SIGMA), rabbit anti-CENP-C (1:20,000), rabbit anti-KNL2 (1:20,000), and rabbit anti-Mis18α (1:5000). HRP-conjugated secondary antibodies included anti-rabbit IgG (1:20,000, Jackson ImmunoResearch), anti-mouse IgG (1:20,000, Jackson ImmunoResearch), or TrueBlot anti-rabbit IgG (1:20,000, ROCKLAND) for immunoprecipitated samples. Signal Enhancer Hikari (Nacalai Tesque) was used to increase sensitivity and specificity. Signals were developed using ECL Prime (GE Healthcare) and detected with the ChemiDoc Touch imaging system (Bio-Rad). Image processing was performed using Image Lab v6.0.2 (Bio-Rad) and Photoshop v26.4.1 (Adobe).

## Immunofluorescence and image acquisition

DT40 cells were centrifuged at 800 rpm at 22 °C for 5 min onto slide glasses using Cytospin4 (Epredia) and fixed with 4% paraformaldehyde (PFA) (Electron Microscopy Sciences) in PBS at room temperature for 5 min. Fixed cells were permeabilized in 0.5% NP-40 alternative (ChemCruz) in PBS at room temperature for 5 min and incubated with primary antibodies in 0.5% BSA (Equitech-Bio Inc) in PBS or Can Get Signal immunostain Solution A (TOYOBO) in the case of rabbit anti-Mis18α for 1 h at 37 °C. Primary antibodies included rabbit anti-CENP-T (1:1000), rabbit anti-Mis18α (1:1000), rabbit anti-Cyclin B2 (1:1000; Proteintech), and mouse anti-α-tubulin (1:10,000; SIGMA). After washing three times with 0.5% BSA in PBS, cells were incubated with secondary antibodies in 0.5% BSA in PBS for 30 min at 37 °C. Secondary antibodies included Alexa 647- or Cy3-conjugated goat anti-rabbit Fab (1:1000; Jackson ImmunoResearch) and FITC- or Cy3-conjugated goat anti-mouse IgG (1:1000; Jackson ImmunoResearch). For Fig. 5B, anti-CENP-T and anti-Cyclin B2 were labeled with CoraLite Plus 555 and 647, respectively. For Fig. EV5H, anti-CENP-T and anti-Cyclin B2 were labeled with CoraLite Plus 555 and 488, respectively, using FlexAble Antibody Labeling Kit for Rabbit (Proteintech) and used for direct immunostaining. For Fig. EV5H, Halo-tagged HJURP or HJURP$^{8K}$ $^{(F506K, E507K, L509K, L513K,}$ $^{F559K, E560K, I562K, L566K)}$ were labeled with 130 nM Janelia Fluor 646 HaloTag Ligand (Promega, #GA1120) for 15 min, washed twice with culture medium, incubated for 15 min, and washed once more. DNA was stained with 1 μg/ml 4′,6-diamidino-2-phenylindole (DAPI; Roche) in PBS for 1 min, and slides were mounted with VECTASHIELD PLUS Antifade mounting medium (Vector Laboratories). Immunofluorescence images were acquired as Z-stack images with 0.2 μm steps (total of 4–8 μm in thickness) using an ORCA-Fusion BT CMOS camera (Hamamatsu Photonics) mounted on a Nikon Eclipse Ti inverted microscope with a Plan Apo λD 100x oil/1.45 NA objective lens (Nikon) and a spinning disk confocal scanner unit (CSU-W1 SoRa; YOKO-GAWA), controlled by NIS-elements AR v5.42.01 (Nikon). Images shown in figures are maximum intensity projections generated with Fiji (ImageJ2 v2.16.0/1.54p) and processed using Photoshop v26.4.1.

## Measurement of cell cycle distribution

HJURP-cKO cell lines expressing EGFP-tagged wild-type HJURP or HJURP$^{2E\ (V117E/I120E)}$ were treated with tetracycline (Tet) for 4, 5, and 6 days. At each time point, cells were cytospun onto glass slides using a Cytospin4 centrifuge (800 rpm, 22 °C, 5 min) and fixed with 4% paraformaldehyde (PFA) in PBS for 5 min at room temperature. Following fixation, cells were permeabilized with 0.5% NP-40 alternative in PBS for 5 min at room temperature. For immunostaining, cells were incubated with anti-α-tubulin antibody (1:10,000; SIGMA) diluted in 0.5% BSA/PBS for 30 min at 37 °C, followed by Cy3-conjugated goat anti-mouse IgG secondary antibody (1:1000; Jackson ImmunoResearch) in 0.5% BSA/PBS for 30 min at 37 °C. DNA was counterstained with DAPI (1 μg/ml). Images were acquired as described in the above section. Cell cycle stages (interphase, prometaphase, metaphase, anaphase/telophase, apoptotic cells, and cells with multi-spindle abnormalities) were scored based on DAPI and α-tubulin staining profiles. A minimum of 440 cells per each condition were analyzed to evaluate cell cycle distribution (Fig. 4B,C).

## Recombinant proteins

*E. coli* Rosetta 2 (DE3) (Merck) cells transformed with MBP-fused HJURP$^{100-150}$ or -HJURP$^{100-150\ (2E),\ (2A),\ or\ (SD)}$ protein expression vectors [pMal/BirA_HJURP$^{100-150}$ or pMal/BirA_HJURP$^{100-150\ (2E),\ (2A),\ or\ (SD)}$] were grown at 37 °C in LB media containing 100 μg/ml ampicillin (Wako) and 40 μg/ml chloramphenicol (Wako) to an OD$_{600}$ of 0.6–0.8. Protein expression was induced with 0.25 mM isopropyl β-D-thiogalactopyranoside (IPTG; Wako). After a 3 h incubation, cells were harvested by centrifugation. Cell pellets were resuspended in MBP binding buffer [50 mM Tris-HCl (pH 8.0), 300 mM NaCl] and lysed by sonication using the Digital Sonifier 250D (Branson). Lysates were clarified by centrifugation at $40,000 \times g$ at 4 °C for 30 min. Supernatants were loaded onto Amylose resin columns (New England Biolabs) equilibrated with MBP binding buffer at 4 °C. After extensive washing, bound proteins were eluted with MBP binding buffer containing 20 mM maltose (Nacalai Tesque). MBP-tagged proteins were applied to a 5-ml Hi-Trap Q HP column (Cytiva) equilibrated with Q buffer [20 mM Tris (pH 8.0), 100 mM NaCl] and purified using a salt gradient from 100 mM to 550 mM NaCl in 20 mM Tris (pH 8.0). Peak fractions were concentrated, snap-frozen in liquid nitrogen, and stored at −80 °C. MBP-AviTag-His$_6$ was expressed using pMal-T-Avi-His/BirA and purified as described above. MBP-fused hsHJURP$^{137-192}$ was expressed with pMal/BirA_hsHJURP$^{137-192}$ and purified using an Amylose resin column followed by gel filtration chromatography using a HiLoad 16/600 Superdex 75pg column (Cytiva) in gel filtration buffer A [20 mM HEPES (pH 7.4), 150 mM NaCl, and 1 mM TCEP (Nacalai Tesque)]. GST-fused ggCENP-C$^{677-864}$ was expressed with pGEX-6P-1_ggCENP-C$^{677-864}$ in Rosetta 2 (DE3) cells as described previously (Hara et al, 2023). Cell pellets were resuspended in GST binding buffer [50 mM HEPES (pH 7.4), 500 mM NaCl, 5% (v/v) glycerol, and 0.5 mM TCEP]. After sonication and clarification by centrifugation, lysates were applied to Glutathione Sepharose 4 Fast Flow columns (Cytiva) equilibrated with GST binding buffer. Columns were washed with GST binding buffer containing 1 M NaCl. The N-terminal GST-tag was cleaved by on-column treatment with human rhinovirus 3C protease (in-house) in GST binding buffer at 4 °C. The eluted tag-free CENP-C fragment was loaded onto a 5-ml Hi-Trap SP HP column (Cytiva) and purified with a linear salt gradient from 100 mM to 730 mM NaCl in SP buffer [20 mM HEPES (pH 7.4), 5% (v/v) glycerol, and 0.5 mM TCEP]. Peak fractions were combined, concentrated, and further purified using a HiLoad 16/600 Superdex 200 pg column (Cytiva) in gel filtration buffer B [20 mM HEPES (pH 7.4), 500 mM NaCl, 5% (v/v) glycerol, and 1 mM TCEP]. The hsCENP-C$^{760-943}$ sample was prepared using the expression vector pGEX-6P-1_hsCENP-C$^{760-943}$ as described previously (Hara et al, 2023).

## Size exclusion chromatography

Samples for analytical size exclusion chromatography were prepared at a final volume of 220 μl containing each protein at 20 μM, except for the ggCENP-C$^{677-864}$ sample without MBP-ggHJURP$^{100-150}$, which contained 100 μM protein. Samples were clarified by centrifugation at $10,000 \times g$ at 4 °C for 5 min. Subsequently, 200 μl of each sample was analyzed using a Superdex 200 Increase 10/300 GL column (Cytiva) in gel filtration buffer C [20 mM HEPES (pH 7.4), 300 mM NaCl, 5% (v/v) glycerol, and 1 mM TCEP]. Fractions were analyzed by SDS-PAGE using 10–20% SuperSep Ace gel (Wako). Protein bands were visualized by CBB staining (Quick-CBB PLUS; Wako), and gel images were acquired using the ChemiDoc Touch imaging system (Bio-Rad).

## In vitro pull-down assays

For MBP pull-down assay, proteins were diluted to 25 μM with pull-down buffer [20 mM HEPES (pH 7.4), 150 mM NaCl, 5% (v/v) glycerol, 1 mM TCEP, and 0.05% (v/v) NP-40 alternative (ChemCruz)]. Binding reactions containing 50 pmol of MBP-AviTag-His$_6$ or MBP-fused HJURP peptide (MBP-ggHJURP$^{100-150}$, -ggHJURP$^{100-150\ (2E),\ (2A),\ or\ (SD)}$, or -hsHJURP$^{137-192}$) and 100 pmol of tag-free CENP-C fragment (ggCENP-C$^{677-864}$ or hsCENP-C$^{760-943}$) were prepared in pull-down buffer (total volume of 50 μl). Mixtures were incubated with 10 μl of Amylose resin for 1 h on ice with gentle agitation every 10 min. Beads were washed three times with 200 μl of pull-down buffer. After removing the supernatant, beads were mixed with 10 μl of SDS-PAGE sample buffer [62.5 mM Tris-HCl (pH 6.8), 2% (w/v) SDS, 10% (v/v) glycerol, 5% (v/v) 2-mercaptoethanol, 0.01% (w/v) bromophenol blue (MP Biomedicals)] and boiled at 96 °C for 5 min. Samples were separated by SDS-PAGE using 10–20% SuperSep Ace gel and visualized by CBB staining. Each pull-down experiment was performed in triplicate to confirm reproducibility.

For in vitro binding assay for chicken HJURP and Mis18α/β, GST-ggHJURP$^{480-580}$ and the His$_6$-ggMis18α$^{38-204}$:MBP-ggMis18β$^{45-213}$ complex were expressed in Rosetta 2 (DE3) and BL21(DE3) (Merck), respectively. Cells were resuspended with pull-down buffer containing 1.25× protease inhibitor (40 μl of buffer per 1 mg of wet cells) and sonicated on ice. The lysate was clarified by centrifugation at $15,000 \times g$ at 4 °C for 3 min. A reaction containing 50 μl each of the supernatant samples was incubated with 10 μl of Glutathione Sepharose 4 Fast Flow beads for 1 h on ice with gentle agitation every 10 min. After washing three times with 200 μl each of pull-down buffer, beads were incubated with 20 μl of SDS-PAGE sample buffer at 37 °C for 5 min to elute the bound proteins. The eluate was then boiled at 96 °C for 5 min and subjected to SDS-PAGE analysis. For competitive pull-down assay, the

reaction also contained 0.63 or 1.25 nmol of purified ggCENP-C[677-864] in a total volume of 200 µl.

## SNAP assay

SNAP assays were performed to visualize newly incorporated CENP-A in G1 cells accumulated with Nocodazole and Reversine. For Figs. 7A,B and EV6C, EGFP-tagged wild-type HJURP or HJURP[2E]-expressing AID-based KNL2 or CENP-C cKO cells with Tet-based HJURP-cKO background were treated with tetracycline for 24 h before SNAP labeling to replace HJURP expression with EGFP-tagged variants. For Fig. EV1F–J, Tet-based Mis18α-cKO cells were cultured in the presence (Mis18α Off) or absence (Mis18α On) of tetracycline for 24 h before SNAP labeling. The AID-based cKO or Tet-based Mis18α-cKO (On or Off) cells expressing HA-SNAP-CENP-A were cultured with 500 ng/ml Nocodazole (SIGMA) for 90 min. Cells were then quenched with 2 µM SNAP-Cell Block (New England Biolabs) for 30 min, washed twice with culture medium, incubated for 30 min, and washed once more. SNAP-blocked cells were cultured again with Nocodazole for 3 h, except for the AID-based KO (Off), to which IAA was also added at this point. Reversine (1 µM; SIGMA) was then added for 1 h to allow mitotic exit (G1 cells) or omitted for M-phase cells (negative control). Newly incorporated HA-SNAP-CENP-A was pulse-labeled with 3 µM TMR-Star (New England Biolabs) for 15 min, washed twice with culture medium, incubated for 15 min, and washed once more. Labeled cells were fixed with 4% PFA in PBS for 5 min, then immunostained with rabbit anti-CENP-T (1:1000) and Alexa 647-conjugated goat anti-rabbit Fab (1:1000). Images were processed, and TMR-Star signal intensities (representing newly incorporated HA-SNAP-CENP-A) were measured using Imaris v9.1.2 (Bitplane).

## Immunoprecipitation (IP)

Immunoprecipitation with nuclear extracts was performed using asynchronous or G1-accumulated DT40 cells. For G1-accumulated cells, approximately $7 \times 10^7$ cells were cultured with 500 ng/ml of Nocodazole in the presence or absence of 500 µM IAA for 8 h (Fig. 6B,D). For Fig. 6C, cells were cultured in the presence or absence of tetracycline for 24 h and treated with Nocodazole for the final 8 h. Reversine (1 µM) was added 0.5 h before harvest. For asynchronous cells, approximately $7 \times 10^7$ cells were cultured without drugs for 8 h (Fig. 6) or with tetracycline for 24 h (Fig. EV4D) to replace HJURP expression with EGFP-tagged variants. Cells were washed twice with PBS and once with ice-cold TMS buffer [10 mM Tris-HCl (pH 7.5), 5 mM MgCl₂, 0.25 M sucrose]. Washed cells were resuspended in 1 ml of TMS-Triton buffer [TMS buffer with 0.5% Triton X-100 (SIGMA), 100 µg/ml of Trichostatin A (TSA; Wako), 25 µM of Purvalanol A (Cayman Chemical), 0.5 mM DTT, 1× protease inhibitor (Roche, cOmplete EDTA-free Protease Inhibitor Cocktail)] and incubated on ice for 10 min. Cell suspensions were centrifuged at $22,000 \times g$ at 4 °C for 20 min to separate nuclei (pellet) from cytoplasmic extract (supernatant). Nuclear pellets were sonicated in 1.8 ml of NaPi buffer [50 mM Na-phosphate (pH 7.5), 100 mM NaCl, 0.1% NP-40, 100 µg/ml of TSA, 25 µM of Purvalanol A, 0.5 mM DTT, phosphatase inhibitor mix (10 mM Na pyrophosphate; SIGMA, 5 mM Na azide; Wako, 10 mM NaF; Wako, 0.4 mM Na

orthovanadate; SIGMA, 20 mM beta-glycerophosphate; Wako), 1× protease inhibitor] followed by centrifugation at $22,000 \times g$ at 4 °C for 20 min. The supernatant (nuclear extract) was used for immunoprecipitation experiments in Fig. 6. The cytoplasmic extract was adjusted to 250 mM NaCl and re-clarified by centrifugation at $22,000 \times g$ at 4 °C for 20 min. This supernatant was used for immunoprecipitation in Fig. EV4D. Immunoprecipitations were performed using 5 µg of anti-GFP (MBL) (Figs. 6B,C and EV4D), anti-KNL2 (Fig. 6D), or rabbit IgG (negative control, SIGMA: 15006) pre-coated onto Dynabeads protein G (Thermo Fisher) in nuclear or cytoplasmic extracts at 4 °C for 12 h. Bound beads were washed 4 times with NaPi buffer (Fig. 6) or wash buffer [20 mM Tris-HCl (pH 8.0), 250 mM NaCl, 1.5 mM MgCl₂, 0.2 mM EDTA, 0.1% Tween 20 (Nacalai Tesque), 0.5 mM DTT, 1× protease inhibitor] (Fig. EV4D). Bound proteins were eluted with SDS-PAGE sample buffer at 37 °C for 15 min and subjected to Immunoblot analysis.

## Chromatin immunoprecipitation sequencing (ChIP-seq) analysis

HJURP ChIP-seq was performed in G1-accumulated cells treated with Nocodazole and Reversine. AID-based cKO cells (Figs. 2C,D and EV2G) were cultured with 500 ng/ml of Nocodazole in the presence or absence of IAA for 8 h, and then Reversine (1 µM) was added 0.5 h before fixation. For Figs. 5D–F and EV4G, EGFP-tagged HJURP[WT] or HJURP[2E]-expressing AID-based KNL2 or CENP-C KO cells with Tet-based HJURP cKO background were treated with tetracycline for 24 h before fixation. Approximately $7 \times 10^7$ G1-accumulated cells were mixed with Drosophila S2 cells at a 1000:1 chromatin ratio (DT40:S2) and cross-linked with 1% paraformaldehyde at 25 °C for 5 min in DMEM medium, then quenched by the addition of 350 mM glycine (Nacalai Tesque). Cells were lysed in nuclear lysis buffer [50 mM HEPES-KOH (pH 7.9), 140 mM NaCl, 1 mM EDTA, 10% glycerol, 0.5% NP-40, 0.25% Triton X-100, 1× protease inhibitor] for 10 min on ice. Nuclei were collected by centrifugation at $1000 \times g$ at 4 °C for 5 min and washed twice with wash buffer [10 mM Tris-HCl (pH 8.0), 200 mM NaCl, 1 mM EDTA, 0.5 mM EGTA, 1× protease inhibitor]. Nuclear pellets were resuspended in 0.4 ml of shearing buffer [50 mM Tris-HCl (pH 8.0), 0.5% SDS, 10 mM EDTA, 1× protease inhibitor], incubated for 10 min on ice, and then diluted with 1.6 ml of ChIP dilution buffer [50 mM Tris-HCl (pH 8.0), 1.1% Triton X-100, 0.11% sodium deoxycholate, 1× protease inhibitor]. Chromatin was sonicated using a Covaris M220 Focused-ultrasonicator (Covaris) with Peak Incident Power 75 W, 10% duty factor, and 200 cycles per burst at 7 °C for 15 min in milliTUBE 1 ml AFA Fiber (Covaris). Sonicated-chromatin was diluted with an equal volume of 2× ChIP dilution buffer [50 mM Tris-HCl (pH 8.0), 1.1% Triton X-100, 300 mM NaCl, 0.11% Na deoxycholate, 1× protease inhibitor] and cleared by centrifugation at $22,000 \times g$ at 4 °C for 10 min. ChIP experiments were performed using 3 µg of anti-HJURP and 1 µg of anti-H2Av (Active Motif) antibodies conjugated with Dynabeads protein G for 12 h. Immunoprecipitated complexes were washed once with low-salt RIPA buffer [50 mM Tris-HCl (pH 8.0), 150 mM NaCl, 1 mM EDTA, 0.05% SDS, 1% Triton X-100, 0.1% Na deoxycholate, 1× protease inhibitor], once with high-salt RIPA buffer (500 mM NaCl), and twice with TE buffer. Cross-links were reversed in

elution buffer [0.5% SDS, 10 mM Tris-HCl (pH 8.0), 5 mM EDTA, 300 mM NaCl] at 65 °C for 8 h. De-cross-linked DNA was treated with RNase A (5 µg/ml; Thermo Fisher) at 37 °C for 30 min and Proteinase K (0.4 mg/ml; SIGMA) at 55 °C for 30 min and purified using the MinElute PCR Purification Kit (Qiagen). ChIP-seq libraries were constructed using KAPA Hyper Prep Kit for Illumina (KAPA BIOSYSTEMS) and sequenced by NovaSeqX Plus (Illumina) with 101 bp paired-end reads at the NGS core facility in the Research Institute for Microbial Diseases, the University of Osaka. Experiments were performed once (Figs. 2C,D and EV2G) or in triplicate (Figs. 5D–F and EV4G).

### Data analysis of Spike-in ChIP-seq

DNA reads from ChIP-seq in DT40 cells were mapped against the chicken reference genome sequence (galGal6; NCBI) and the Drosophila reference genome (dm6; NCBI) used for spike-in normalization. Adapter trimming for raw sequencing data was performed using Fastp v0.20.0 (https://github.com/OpenGene/fastp). A concatenated reference genome (galGal6 + dm6) was prepared. Trimmed reads were aligned to the reference using Bowtie 2 v2.4.2 (http://bowtie-bio.sourceforge.net/bowtie2/index.shtml) with parameters: --no-mixed --no-discordant --no-unal. Mapped read counts onto chicken or Drosophila genomes were performed separately for each merged reference BAM file, and spike-in scale factors were calculated for normalizing to dm6 read counts. Mapped reads were normalized against each corresponding input and scaled with spike-in scale factors using bamCompare (deepTools v3.5.0; https://deeptools.readthedocs.io/en/develop/) with a bin size of 50 bp and a smooth length of 150 bp for reads-ratio profiles, or a bin size of 1 kbp and a smooth length of 3 kbp for comparing reads-ratio among samples, using parameters: --ignoreDuplicates --skipNAs --operation ratio. The reads-ratio profiles were visualized using Integrative Genomics Viewer v2.19.2 (http://www.broadinstitute.org/igv/) (Robinson et al, 2011). The sum of normalized mapped reads-ratio in the 100 kb centromere core (Cen) and the 1 Mb region around the centromere (1 M arm) for the chicken Cen5 or CenZ was calculated using bigwigtowig (deepTools v3.5.0). Relative reads-ratio values were plotted [the background-subtracted reads amount mapped onto CenZ or Cen5 divided by the corresponding factor before knockout (On)] using the formula: Relative reads ratio = $[Cen^R − arm^R]/[CenR (On) − arm^R (On)]$, where $Cen^R$ is Cen reads ratio and $arm^R$ is 1 M arm (-Cen) reads ratio.

### Flow-cytometry analysis

HJURP-cKO cell lines expressing EGFP-tagged wild-type HJURP or HJURP$^{2E}$ $^{(V117E/I120E)}$ were treated with tetracycline for 4, 5, and 6 days. At each time point, cells were pulsed with 20 µM BrdU (SIGMA) for 20 min. Approximately $5 × 10^6$ cells were harvested by centrifugation at $200 × g$ for 5 min at 4 °C, washed once with 10 ml ice-cold PBS, and fixed in 13 ml of ice-cold 70% ethanol overnight at −30 °C. Fixed cells were collected by centrifugation at $600 × g$ for 3 min at 4 °C and washed once with 1 ml of 1% BSA in PBS. DNA was denatured by incubating cells in 1 ml of 4 N HCl containing 0.5% Triton X-100 for 30 min at room temperature. After centrifugation at $600 g$ for 3 min at 4 °C, cells were washed three times with 1 ml of 1% BSA in PBS. Cells were then incubated with

30 µl of anti-BrdU antibody (BD) for 1 h at room temperature, washed twice with 1 ml of 1% BSA in PBS, and incubated with 30 µl of FITC-conjugated anti-mouse IgG (1:20 dilution in 1% BSA in PBS; Jackson ImmunoResearch) for 30 min at room temperature. Following a final wash with 1 ml of 1% BSA in PBS, cells were resuspended in 1 ml of 10 µg/ml Propidium iodide (SIGMA) in 1% BSA in PBS and incubated overnight at 4 °C before analysis. Stained cells were applied to FACS Canto II (BD) and analyzed by FACSDiva v9.0 (BD) (Fig. EV4B,C).

### Structure prediction

Structural models for the HJURP-CENP-C complex from chicken, human, and frog were produced using AlphaFold3 (Abramson et al, 2024) on the AlphaFold server (https://alphafoldserver.com/). All graphical presentations of the predicted models were generated using PyMOL v3.0 (Schrödinger).

### Statistical analysis

Fluorescence signal intensities of TMR-star-labeled SNAP-CENP-A, KNL2-mScarlet, EGFP-CENP-C, EGFP-HJURP, or EGFP-HJURP$^{2E}$ on centromeres were quantified using Imaris software (Bitplane) and Fiji (Schindelin et al, 2012). For quantification, fluorescence signals from approximately 30–70 kinetochores in each of 30 (Figs. 1C–E and EV1H), 20 (Figs. 2A,B, 7A,B, and EV2A,B), 15 (Fig. 5C) or 10 (Fig. EV5I) G1-phase cells and in an equal number of M-phase cells (negative control in SNAP assay; Figs. 1C–E, 7A,B, EV1E, EV1J, and EV6C) were analyzed. Signal intensities were calculated by subtracting average background signals in non-centromere regions within each cell. Average values of relative signal intensities are shown with standard deviation, except in Figs. 5C and EV5I, where median relative signal intensities are shown with first and third quartiles using violin plots. For SNAP assays, we subtracted the average of SNAP-CENP-A signal intensities of M-phase cells from the corresponding average intensity in each G1 cell to determine CENP-A incorporation values (Figs. 1C–E, 7A,B and EV1J). Sample collection was conducted as a no-biased blind test. Statistical differences between groups were evaluated by two-sided Student's t-test, with $p < 0.05$ defined as significant. NS indicates no significant difference. Graphs of fluorescence signal intensities were created using GraphPad Prism v9.5.1 (GraphPad Software).

## Data availability

ChIP-seq data are deposited in the NCBI database under the accession number PRJNA1235856.

The source data of this paper are collected in the following database record: biostudies:S-SCDT-10_1038-S44318-025-00674-z.

## Peer review information

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

## Acknowledgements

The authors are very grateful to members of the Fukagawa Lab for their fruitful discussions. We also thank R. Fukuoka, Y. Kubota, T. Motohashi, and K. Oshimo for technical assistance and R. Maeda in the University of Osaka for discussions and helpful suggestions on the ChIP-seq analysis. We appreciate the sequence core facility at the University of Osaka. This work was supported by CREST of JST (JPMJCR21E6), JSPS KAKENHI Grant Numbers 22H00408, 23K18113, 24H02281, and 25H00975 to TF, JSPS KAKENHI Grant Number 23K27153 to TH, JSPS KAKENHI Grant Number 24K23185 to YM, and JSPS KAKENHI Grant Number 24K09340 to MA.

## Author contributions

**Tetsuya Hori**: Conceptualization; Resources; Data curation; Formal analysis; Investigation; Writing—original draft; Writing—review and editing. **Yutaka Mahana**: Formal analysis; Investigation; Writing—review and editing. **Mariko Ariyoshi**: Formal analysis; Investigation; Writing—review and editing. **Tatsuo Fukagawa**: Conceptualization; Resources; Supervision; Writing—original draft; Project administration; Writing—review and editing.

Source data underlying figure panels in this paper may have individual authorship assigned. Where available, figure panel/source data authorship is listed in the following database record: biostudies:S-SCDT-10_1038-S44318-025-00674-z.

## Disclosure and competing interests statement

The authors declare no competing interests.

# Expanded View Figures

**Figure EV1. Generation of various AID-based HJURP, KNL2, CENP-C, and Tet-based Mis18α knockout cell lines expressing SNAP-CENP-A, related Figs. 1 and 2.** ▶

(A) Immunoblot analysis to detect HJURP with an anti-HJURP antibody in AID-based HJURP knockout cells at the indicated times after IAA addition. Signal intensities of HJURP or ^EGFP-mAid HJURP on the blot were measured with normalization to α-tubulin intensity in each blot. Wild-type DT40 cells (WT) were also analyzed. SNAP-CENP-A was detected with an anti-HA antibody. α-tubulin was probed as a loading control. Molecular weight markers (kDa) are shown on the left. Asterisk (*) indicates a non-specific band. (B) Immunoblot analysis to detect KNL2 with an anti-KNL2 antibody in AID-based KNL2 knockout cells in either the absence (−) or the presence (+) of IAA (3.5 h). Signal intensities of KNL2 or KNL2^AID on the blot were measured with normalization to α-tubulin intensity in each blot. Wild-type DT40 cells (WT) were also analyzed. SNAP-CENP-A was detected with an anti-HA antibody. α-tubulin was probed as a loading control. Molecular weight markers (kDa) are shown on the left. (C) Immunoblot analysis to detect CENP-C with an anti-CENP-C antibody in AID-based CENP-C KO cells in either the absence (−) or the presence (+) of IAA (3.5 h). Signal intensities of CENP-C or CENP-C^mAid-EGFP on the blot were measured with normalization to α-tubulin intensity in each blot. Wild-type DT40 cells (WT) were also analyzed. SNAP-CENP-A was detected with an anti-HA antibody. α-tubulin was probed as a loading control. Molecular weight markers (kDa) are shown on the left. (D) Cell synchronization using Nocodazole/Reversine treatment to accumulate G1 phase cells. Cells were synchronized according to the indicated scheme. At the indicated time points in the presence or absence of Reversine after Nocodazole treatment (4 h), cell populations were examined by microscopy observation of DAPI-stained nuclei, and cell-cycle distribution was shown as M-phase and interphase. Scale bar, 10 μm. (E) A scheme of the SNAP assay for mitotic cells as a negative control. AID-based knockout cell lines were cultured with Nocodazole, and existing SNAP-tagged CENP-A was quenched with SNAP-Cell Block followed by the addition of IAA to degrade the target protein. After allowing for new CENP-A synthesis for 4.5 h, the SNAP-CENP-A was pulse-labeled with TMR-Star and evaluated in each KO M-phase cell. Representative mitotic images of SNAP-CENP-A in AID-based HJURP, KNL2, and CENP-C knockout cell lines in either the absence or the presence of IAA. EGFP-mAid-tagged HJURP or mAid-EGFP-tagged CENP-C localization is also shown in HJURP KO or CENP-C KO, respectively. CENP-T was used as a centromere marker. DNA was stained with DAPI. Scale bar, 10 μm. (F) Genomic map of the Mis18α gene locus showing exons, targeting vectors, and a probe for Southern hybridization analysis. ApaLI restriction sites and predicted detection fragments are indicated. The strategy shows the generation of the Tet-based Mis18α knockout by sequential targeting with first and second allele targeting vectors to replace exons 1–3 of the Mis18α gene with the histidinol dehydrogenase gene (HisD) or Puromycin resistant gene (Puro^R). Mis18α cDNA with a Tet-responsive promoter was introduced before the second allele targeting. (G) Southern hybridization analysis confirming Mis18α gene disruption. ApaLI-digested genomic DNAs in wild-type (+/+), first allele-targeted (−/+), and knockout with Mis18α cDNA (−/−, +cDNA) cells were hybridized with the 3′-probe. Expected fragment sizes were detected: 10.8 kb (+/+), 10.8 and 28.3 kb (−/+), or 27.4 and 28.3 kb (−/−, +cDNA) bands. Molecular weight markers (kb) are shown on the left. (H) A scheme of the SNAP assay for G1 (Nocodazole/Reversine double treatment; Noc.+Rev.) and mitotic (Nocodazole treatment; Noc.) cells in Tet-based Mis18α knockout cells expressing SNAP-CENP-A. Cells were cultured in the presence or absence of Tet for 24 h before SNAP labeling. The cells were cultured with Nocodazole, and existing SNAP-tagged CENP-A was quenched with SNAP-Cell Block. After allowing for new CENP-A synthesis, Reversine was added to permit mitotic exit (G1) or omitted for M-phase cells. Newly incorporated SNAP-CENP-A was pulse-labeled with TMR-Star, and SNAP-CENP-A intensities were evaluated under each condition. (I) Immunoblot analysis to detect Mis18α with an anti-Mis18α antibody in Tet-based Mis18α knockout cells in either the absence (−) or the presence (+) of Tet (22 h). Signal intensities of Mis18α on the blot were measured with normalization to α-tubulin intensity in each blot. Wild-type DT40 cells (WT) were also analyzed. SNAP-CENP-A was detected with an anti-HA antibody. α-tubulin was probed as a loading control. Molecular weight markers (kDa) are shown on the left. (J) Representative images of new CENP-A incorporation (SNAP-CENP-A) in Tet-based Mis18α knockout cell lines for G1 (Nocodazole/Reversine double treatment; Noc.+Rev.) and mitotic (Nocodazole treatment; Noc.) cells in either the absence or the presence of Tet. CENP-T was used as a centromere marker. DNA was stained with DAPI. Scale bar, 10 μm. Each assay was conducted twice (n = 30 cells per condition). The average of SNAP-CENP-A signal intensities in M-phase cells (Noc.) was subtracted from the corresponding average intensity in each G1 cell (Noc.+Rev.) to determine CENP-A incorporation. Results are presented as mean relative intensity ± standard deviation (SD). A two-tailed Student's t-test was performed to compare Mis18α On and Off conditions. The p-value is $1.99 \times 10^{-6}$.

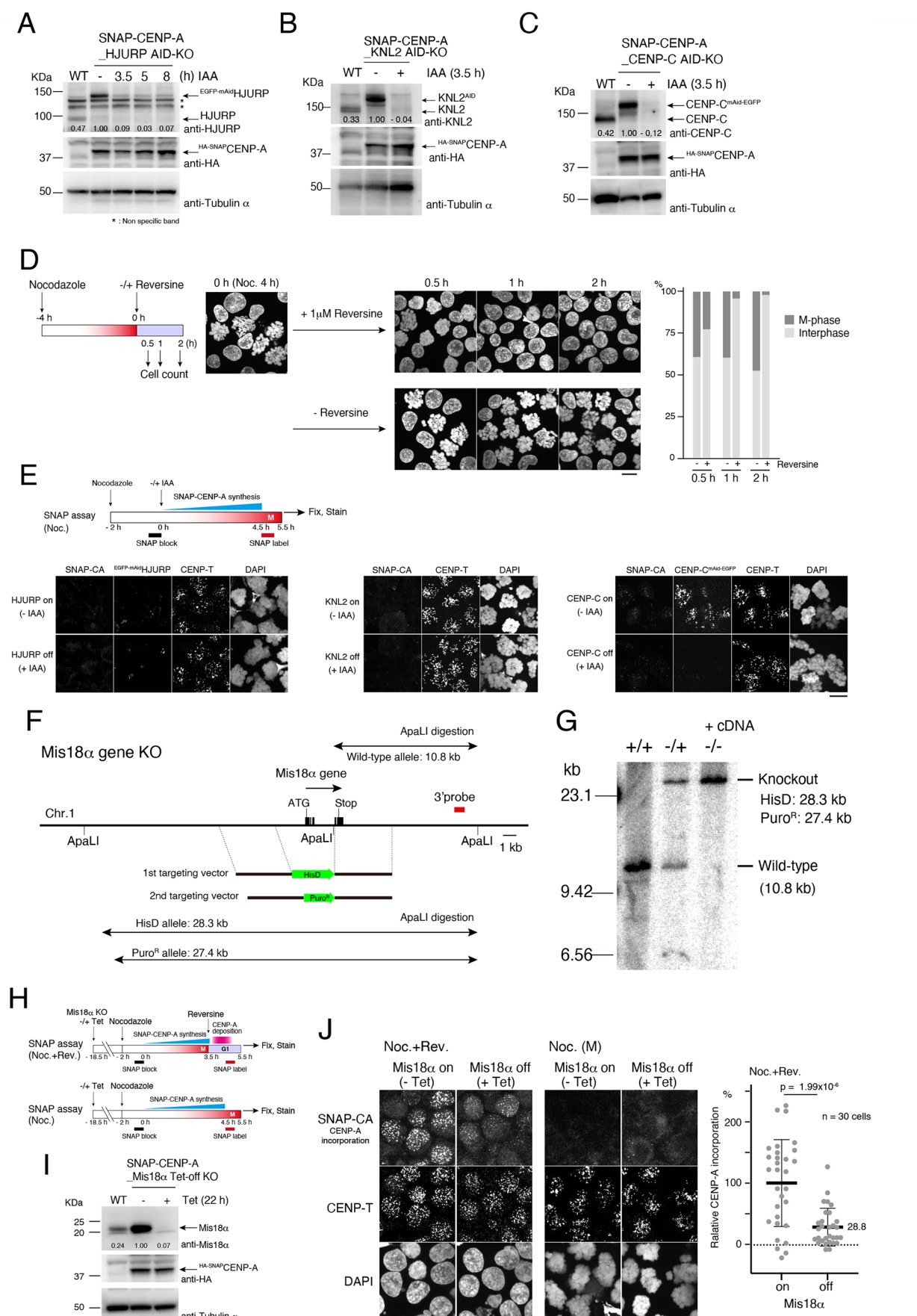

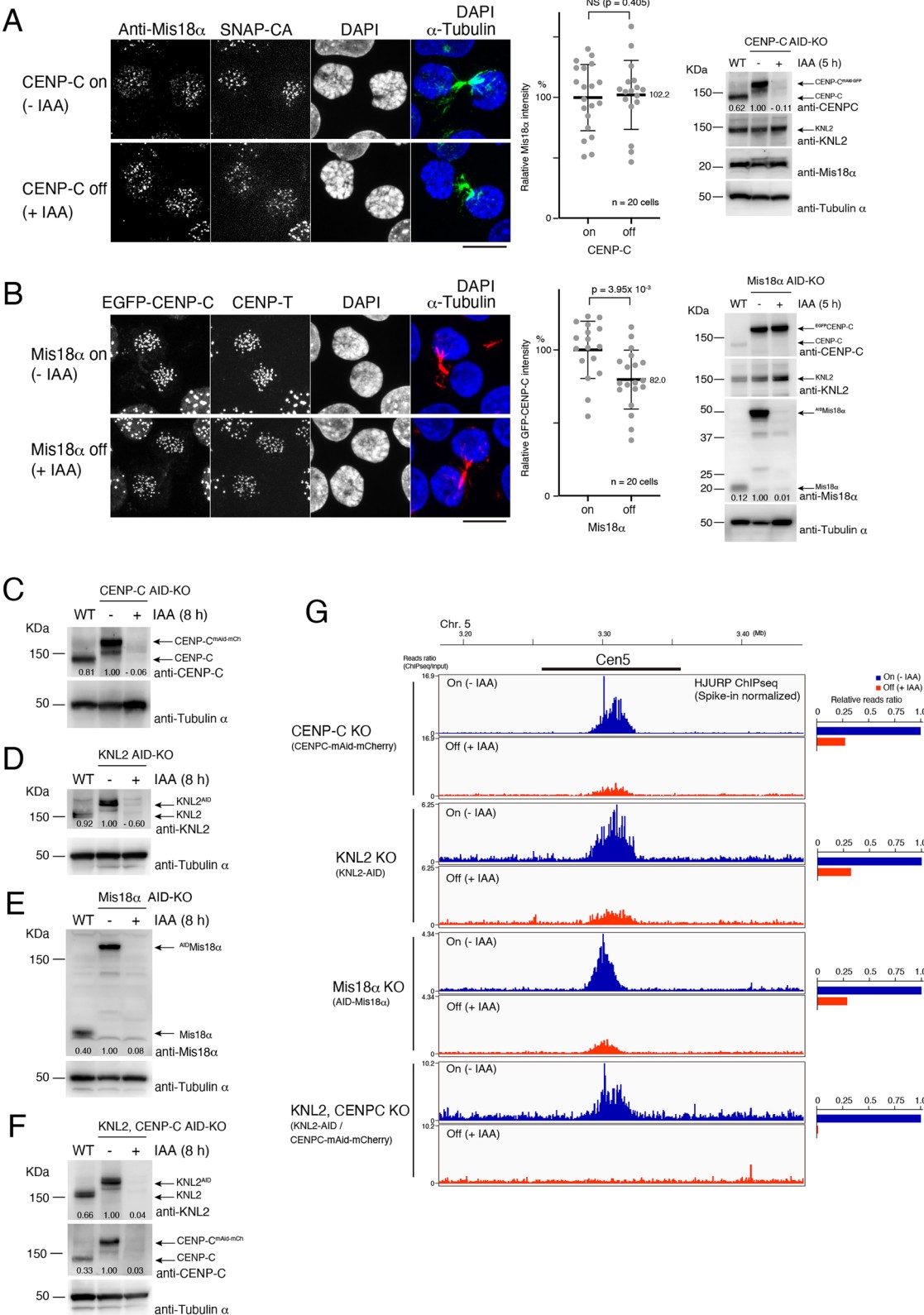

◀ **Figure EV2.   Quantification of HJURP levels on centromeres in each knockout cell line, related to Fig. 2.**

(A) Mis18α localization in G1 phase cells in AID-based CENP-C knockout cell lines in either the absence (CENP-C On) or the presence (CENP-C Off) of IAA. Mis18α was detected by indirect-immunofluorescence analysis with an anti-Mis18α antibody. α-tubulin was used to identify G1 cells. Full-labeled SNAP-CENP-A (SNAP-CA) was used as a centromere marker. DNA was stained with DAPI. Scale bar, 10 μm. Signal intensities of Mis18α were measured in CENP-C On or Off cells. Each assay was conducted twice ($n = 20$ cells per condition), and results are presented as the mean relative intensities ± standard deviation (SD). A two-tailed Student's t-test was performed to compare CENP-C On and Off conditions. The p-value is 0.405. Immunoblot analyses to detect CENP-C, KNL2, and Mis18α in AID-based CENP-C knockout cell lines in either the absence (−) or the presence (+) of IAA (5 h). Signal intensities of CENP-C or CENP-C$^{mAid-EGFP}$ on the blot were measured with normalization to α-tubulin intensity in each blot. Wild-type cells (WT) were also analyzed. α-tubulin was probed as a loading control. Molecular weight markers (kDa) are shown on the left. (B) CENP-C localization in G1 phase cells in AID-based Mis18α knockout cell lines in either the absence (Mis18α On) or the presence (Mis18α Off) of IAA. CENP-C was visualized by expression of EGFP-tagged CENP-C. CENP-T was used as a centromere marker. DNA was stained with DAPI. α-tubulin was used to identify G1 cells. Scale bar, 10 μm. Signal intensities of EGFP-tagged CENP-C were measured in Mis18α On or Off cells. Each assay was conducted twice ($n = 20$ cells per condition), and results are presented as the mean relative intensities ± standard deviation (SD). A two-tailed Student's t-test was performed to compare Mis18α On and Off conditions. The p-value is $3.95 \times 10^{-3}$. Immunoblot analyses to detect CENP-C, KNL2, and Mis18α in AID-based Mis18α knockout cell lines in either the absence (−) or the presence (+) of IAA (5 h). Signal intensities of Mis18α or $^{AID}$Mis18α on the blot were measured with normalization to α-tubulin intensity in each blot. Wild-type cells (WT) were also analyzed. α-tubulin was probed as a loading control. Molecular weight markers (kDa) are shown on the left. (C) Immunoblot analysis to detect CENP-C with an anti-CENP-C antibody in AID-based CENP-C knockout cells in either the absence (−) or the presence (+) of IAA (8 h). Signal intensities of CENP-C or CENP-C$^{mAid-mCh}$ on the blot were measured with normalization to α-tubulin intensity in each blot. Wild-type DT40 cells (WT) were also analyzed. α-tubulin was probed as a loading control. Molecular weight markers (kDa) are shown on the left. (D) Immunoblot analysis to detect KNL2 with an anti-KNL2 antibody in AID-based KNL2 knockout cells in either the absence (−) or the presence (+) of IAA (8 h). Signal intensities of KNL2 or KNL2$^{AID}$ on the blot were measured with normalization to α-tubulin intensity in each blot. Wild-type DT40 cells (WT) were also analyzed. α-tubulin was probed as a loading control. Molecular weight markers (kDa) are shown on the left. (E) Immunoblot analysis to detect Mis18α with an anti-Mis18α antibody in AID-based Mis18α knockout cells in either the absence (−) or the presence (+) of IAA (8 h). Signal intensities of Mis18α or

$^{AID}$Mis18α on the blot were measured with normalization to α-tubulin intensity in each blot. Wild-type DT40 cells (WT) were also analyzed. α-tubulin was probed as a loading control. Molecular weight markers (kDa) are shown on the left. (F) Immunoblot analysis to detect KNL2 and CENP-C with an anti-KNL2 or CENP-C antibody, respectively, in AID-based KNL2/CENP-C double knockout cells in either the absence (−) or the presence (+) of IAA (8 h). Signal intensities of KNL2 or KNL2$^{AID}$ and CENP-C or CENP-C$^{mAid-mCh}$ on the blot were measured with normalization to α-tubulin intensity in each blot. Wild-type DT40 cells (WT) were also analyzed. α-tubulin was probed as a loading control. Molecular weight markers (kDa) are shown on the left. (G) Spike-in normalized ChIP-seq profiles with an anti-HJURP antibody around the centromere of the chicken chromosome 5 (Cen5) in AID-based CENP-C, KNL2, Mis18α, and KNL2/CENP-C double knockout cell lines in either the absence or the presence of IAA. Quantification of the normalized mapped-read ratios on the Cen5 region is shown on the right.

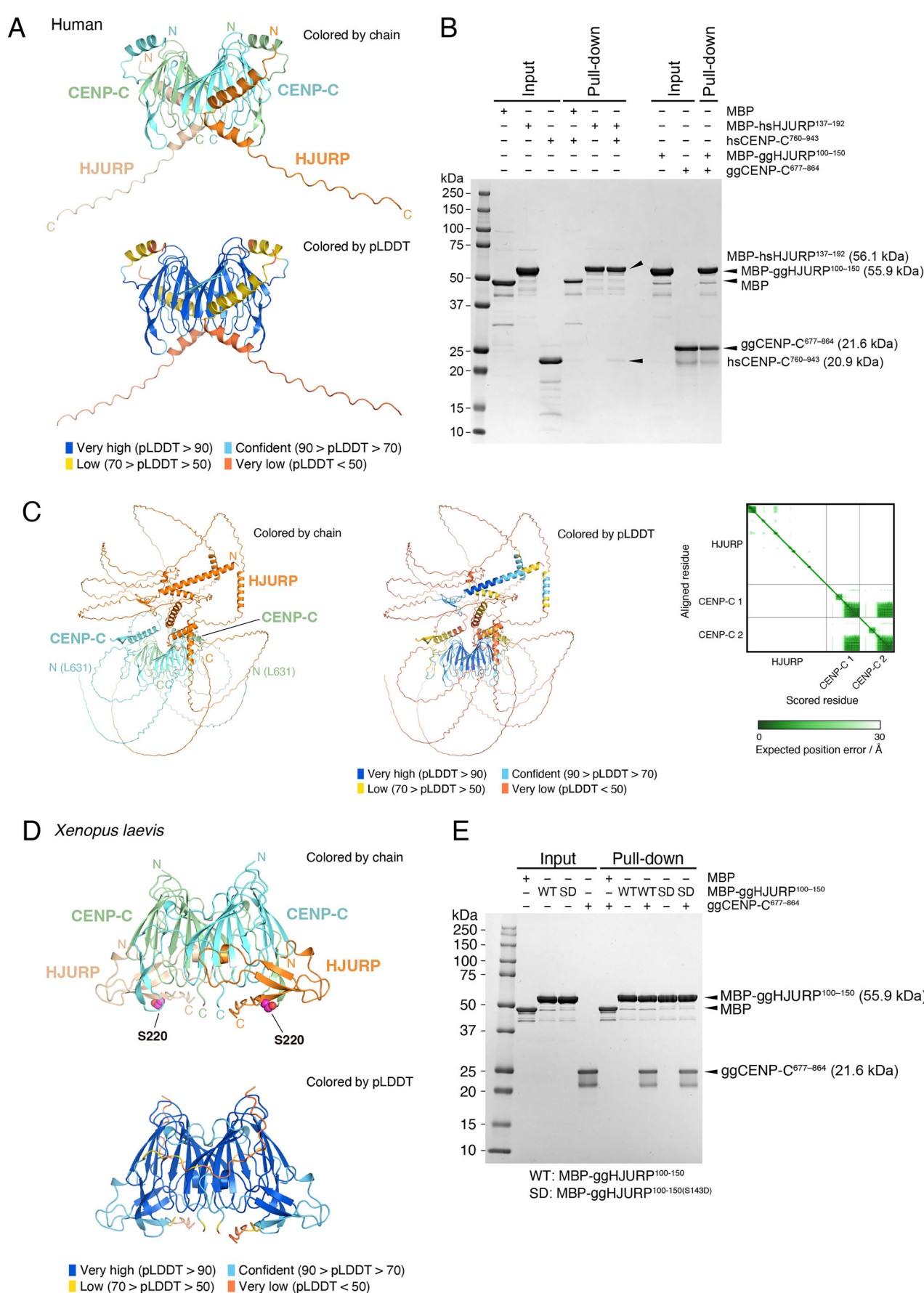

◄ **Figure EV3.   HJURP-CENP-C interaction in other species, related to Fig. 3.**

(A) AlphaFold3 (AF3) prediction for the interaction between the aa 137–192 region of human HJURP and a dimer of human CENP-C C-terminal region (aa 760–943 region). Models are colored by chain (top) or by the pLDDT values (bottom). The N-terminal region of the CENP-C model (aa 760–800) is omitted for clarity. Human HJURP[137-192] is predicted to adopt α-helices with relatively low confidence, with low or very low pLDDT values. (B) MBP pull-down assay demonstrating the interaction of MBP-fused HJURP (chicken: ggHJURP[100-150], human: hsHJURP[137-192]) with CENP-C C-terminus (chicken: ggCENP-C[677-864], human: hsCENP-C[760-943]). MBP-ggHJURP[100-150] pulled down ggCENP-C[677-864], while MBP-hsHJURP[137-192] failed to pull down hsCENP-C[760-943]. Molecular weight markers (kDa) are shown on the left. (C) AF3 prediction for the interaction between full-length human HJURP and a dimer of human CENP-C C-terminal region (aa 631–943). The models are colored by chain (left) or by the pLDDT scores (middle). The PAE plot is shown on the right. (D) AF3 prediction for the interaction between the aa 173–232 region of Xenopus laevis HJURP and a dimer of Xenopus CENP-C C-terminal region (aa 1228–1400 region). Models are colored by chain (top) or by the pLDDT values (bottom). The N-terminal region of the CENP-C model (aa 1228–1254) is not shown for clarity. Xenopus HJURP[1255-1400] is predicted to contain an additional β-strand in the CNEP-C binding region, which carries a potential CDK1 phosphorylation site, S220. The sidechain of S220 in Xenopus HJURP is shown as spheres. (E) MBP pull-down assay demonstrating the interaction of MBP-fused ggHJURP[100-150] (either wild-type (WT) or phosphomimetic mutant (S143D: SD)) with ggCENP-C[677-864]. Both wild-type and phosphomimetic mutant HJURP pulled down ggCENP-C[677-864]. Molecular weight markers (kDa) are shown on the left.

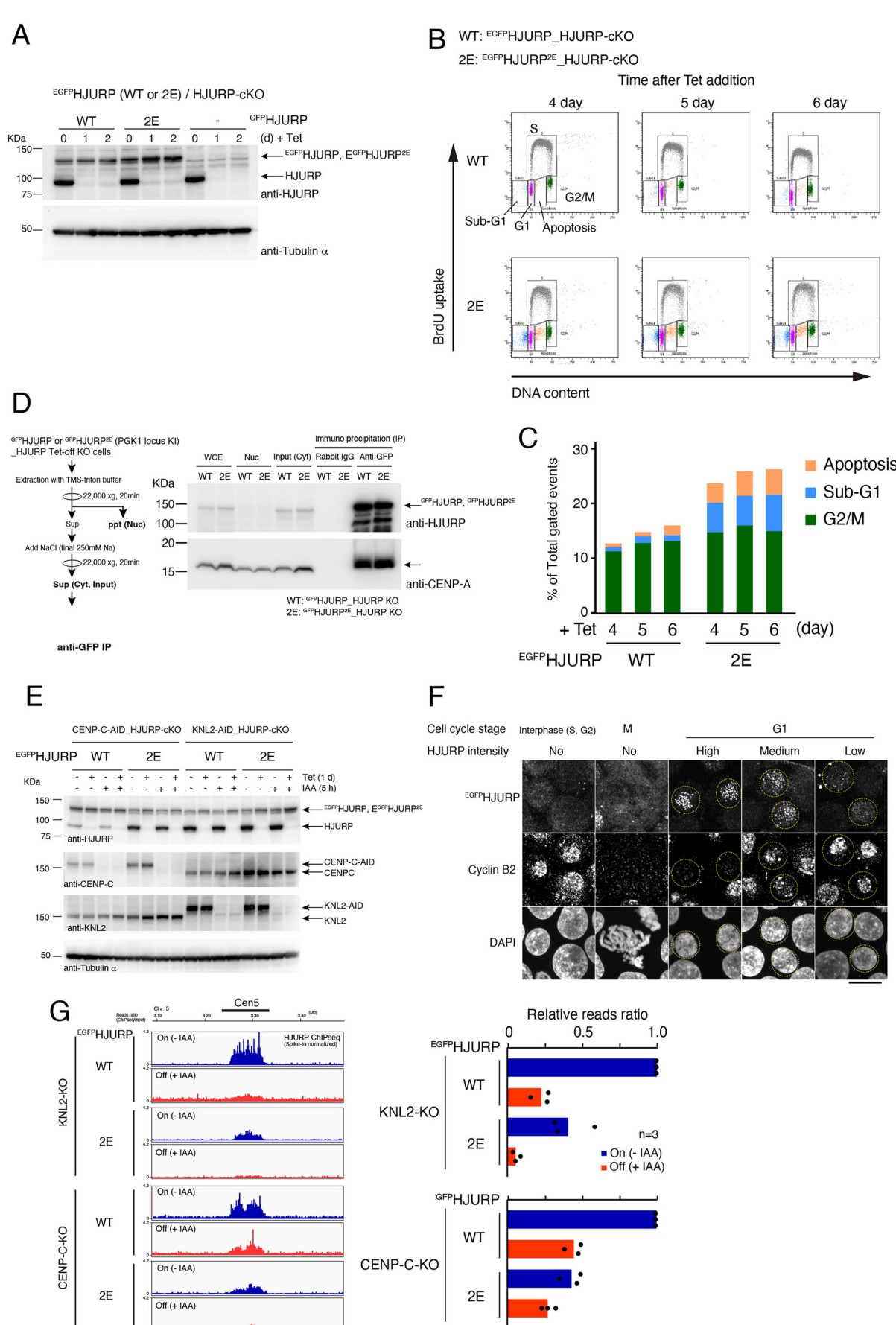

◀ **Figure EV4. Expression of HJURP²ᴱ lacking its CENP-C interaction reduced centromere localization of HJURP in DT40 cells, related to Figs. 4 and 5.**

(A) Immunoblot analysis to detect HJURP with an anti-HJURP antibody in Tet-based HJURP conditional knockout (cKO) cells expressing EGFP-HJURP (WT) or EGFP-HJURP²ᴱ (2E) at the indicated time points after Tet addition. α-tubulin was probed as a loading control. Molecular weight markers (kDa) are shown on the left. (B) Flow cytometry analysis of HJURP-cKO cells expressing EGFP-fused HJURP (WT) or HJURP²ᴱ (2E) at 4, 5, and 6 days after Tet addition. G1, S, G2/M, Sub-G1, and apoptotic phases are shown. (C) Quantification of cell populations of HJURP-cKO cells expressing EGFP-fused HJURP (WT) or HJURP²ᴱ (2E) at 4, 5, and 6 days after Tet addition as determined by flow cytometry analysis. (D) Co-IP experiments using cytoplasmic extracts with an anti-GFP antibody in Tet-based HJURP conditional knockout cells expressing either EGFP-HJURP or EGFP-HJURP²ᴱ ⁽ⱽ¹¹⁷ᴱ/ᴵ¹²⁰ᴱ⁾ as a bait. Tetracycline was added to the culture medium 24 h before extraction to replace endogenous HJURP expression with EGFP-fused HJURP variants. Immunoblot analyses to detect HJURP and CENP-A in either EGFP-HJURP (WT) or EGFP-HJURP²ᴱ (2E) expressing cells. The co-immunoprecipitated CENP-A amounts in WT cells were comparable to those in cells expressing EGFP-HJURP²ᴱ, indicating that the 2E mutation does not affect HJURP-CENP-A binding. Each assay was performed twice. Molecular weight markers (kDa) are shown on the left. The workflow of the cytoplasmic extract preparation and immunoprecipitation is shown on the left. (E) Immunoblot analysis to detect HJURP, CENP-C, and KNL2 with anti-HJURP, anti-CENP-C, and anti-KNL2 antibodies, respectively, in AID-based CENP-C or KNL2 knockout cells with Tet-based HJURP-cKO background. Cells expressing either EGFP-HJURP (WT) or EGFP-HJURP²ᴱ (2E) were analyzed in the presence (+) or absence (−) of IAA (5 h) or Tet (24 h) as indicated. α-tubulin was probed as a loading control. Molecular weight markers (kDa) are shown on the left. (F) EGFP-HJURP localization in DT40 cells at various cell-cycle stages stained with an anti-Cyclin B2 antibody. EGFP-HJURP signals are classified as high, medium, or low intensity, corresponding to different cell-cycle stages. In Cyclin B2-negative interphase cells (early G1 phase), EGFP-HJURP signals are high. In contrast, EGFP-HJURP signals were weak (medium and low) in Cyclin B2-positive G1 cells and they were undetectable in Cyclin B2-positive S and G2 phases. DNA was stained with DAPI. Scale bar, 10 μm. (G) Left: Spike-in normalized ChIP-seq profiles around the centromere of the chicken chromosome 5 (Cen5) with an anti-HJURP antibody in AID-based CENP-C or KNL2 knockout cells expressing EGFP-HJURP (WT) or EGFP-HJURP²ᴱ (2E) in either the absence or the presence of IAA. Right: Quantification of normalized mapped-read ratios on the Cen5 region by Spike-in ChIP-seq for each condition. Each assay was performed three times. Results are presented as the mean with individual experimental values shown as dots. For the ChIP-seq profiles, the representative data sets are shown.

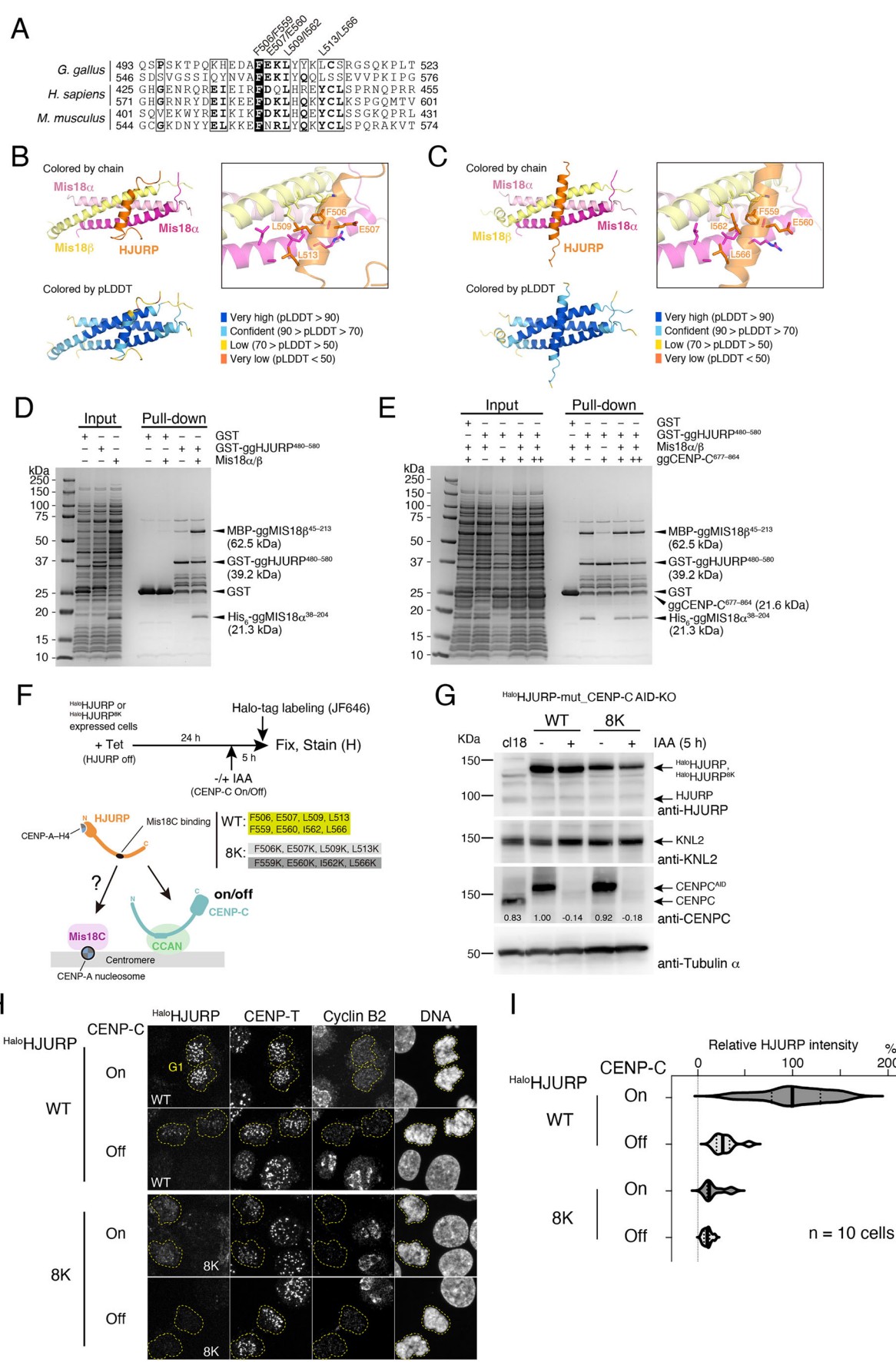

◀ **Figure EV5. CENP-C and Mis18α/β separately bind to HJURP, related to Fig. 6.**

(A) Amino acid sequence of the short repeats of HJURP from chicken, human, and mouse. Putative key residues for the Mis18α/β binding are annotated on the top. (B) An AF3 model for the interaction between chicken HJURP (aa 493–523) and the Mis18α/β complex containing two Mis18α molecules (aa 160–204) and one Mis18β molecule (aa 160–213). In the magnified view, the residues involved in the putative interface are shown as sticks. (C) An AF3 model showing the interaction between chicken HJURP (aa 546–576) and the Mis18α/β complex. (D) GST pull-down assay using lysate from bacterial cells expressing GST-ggHJURP$^{480-580}$ and His$_6$-ggMis18α$^{38-204}$:MBP-ggMis18β$^{45-213}$. Molecular weight markers (kDa) are shown on the left. (E) Competitive GST pull-down assay for the ggHJURP$^{480-580}$ binding to the Mis18α/β complex and the CENP-C C-terminal region. The concentration of the purified ggCENP-C$^{677-864}$ in the reaction was 3.1 μM (+) or 6.2 μM (++). Molecular weight markers (kDa) are shown on the left. (F) Schematic representation of the experimental scheme to analyze the significance of the interaction between Mis18C and HJURP in cells. AID-based CENP-C knockout cells with a Tet-inducible HJURP-cKO background expressing Halo-HJURP (WT) or Halo-HJURP$^{8K}$ (8K) were used. Tet was added to the culture medium 24 h before fixation to replace wild-type HJURP expression with Halo-fused HJURP variants. In CENP-C off cells, the Mis18C pathway alone could be examined in cells expressing Halo-HJURP or Halo-HJURP$^{8K}$. (G) Immunoblot analysis of HJURP, KNL2, and CENP-C using anti-HJURP, anti-KNL2, and anti-CENP-C antibodies, respectively, in AID-based CENP-C knockout cells with Tet-based HJURP-cKO background expressing Halo-HJURP (WT) or Halo-HJURP$^{8K}$ (8K) in the presence (+) or absence (−) of IAA (5 h). Signal intensities of CENP-C or CENP-C$^{AID}$ on the blot were measured with normalization to α-tubulin intensity in each blot. Wild-type cells (Cl18) were also analyzed. α-tubulin was probed as a loading control. Molecular weight markers (kDa) are shown on the left. (H) Localization of Halo-tagged HJURP (WT) or HJURP$^{8K}$ (8 K) in AID-based CENP-C knockout cells with a HJURP-cKO background as described in (F). Cyclin B2-negative cells were identified as early G1 cells and marked with yellow dotted lines. CENP-T was used as a centromere marker. DNA was stained with DAPI. Scale bar, 10 μm. (I) Relative intensities of Halo-HJURP (WT) or Halo-HJURP$^{8K}$ (8K) in AID-based CENP-C knockout cell lines in the absence or presence of IAA. Data are presented as median relative signal intensities (bold line) with first and third quartiles (dotted lines) using violin plots.

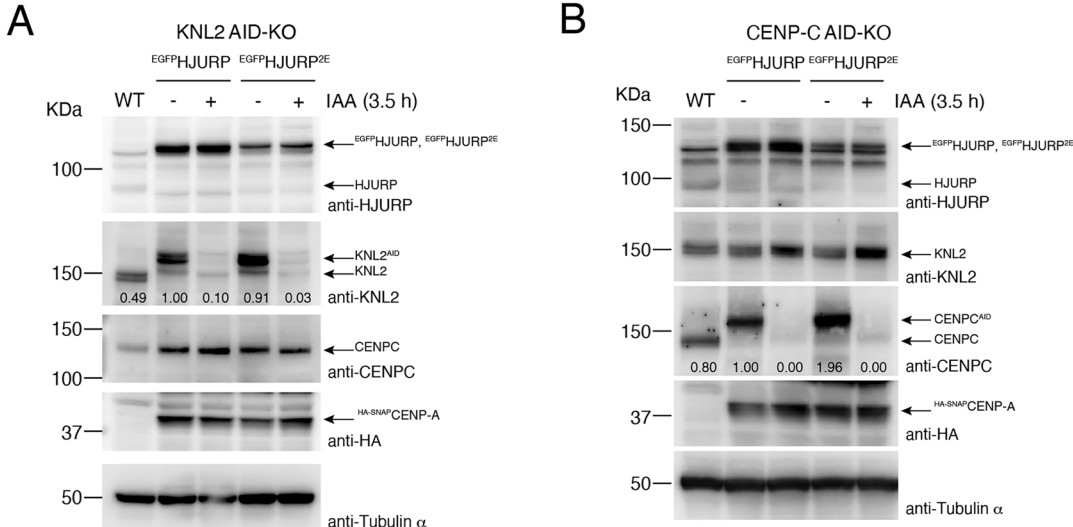

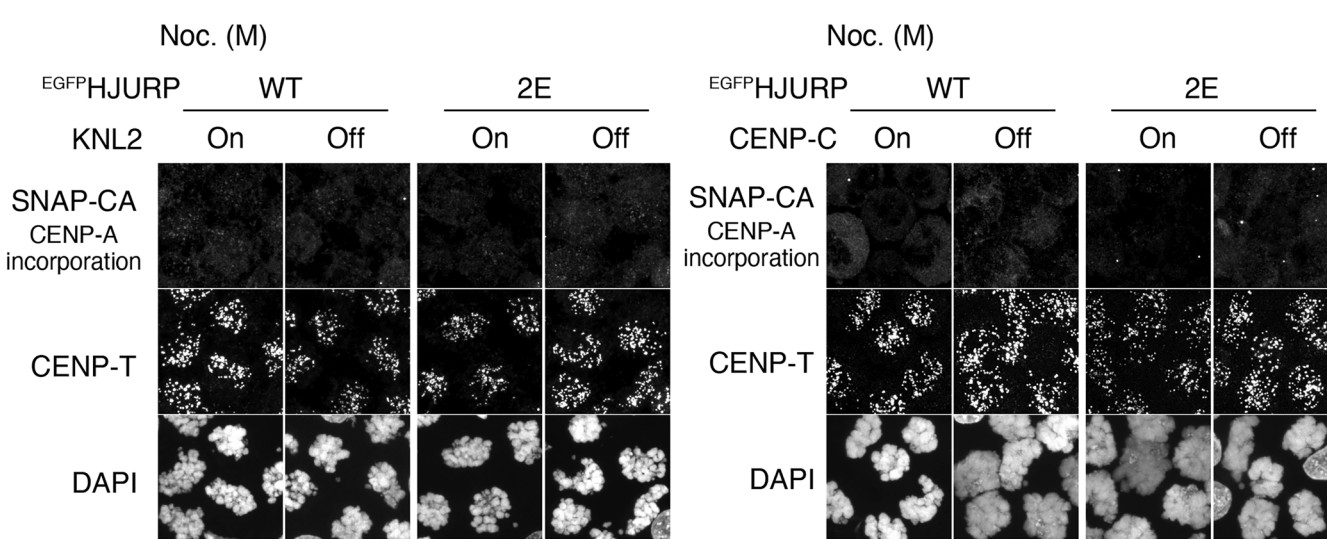

**Figure EV6.    Characterization of cell lines used for SNAP assay in Fig. 7, related to Fig. 7.**

(A) Immunoblot analysis to detect HJURP, KNL2, CENP-C, and SNAP-CENP-A with anti-HJURP, anti-KNL2, anti-CENP-C, and anti-HA antibodies, respectively, in AID-based KNL2 knockout cells with Tet-based HJURP-cKO background expressing EGFP-HJURP or EGFP-HJURP[2E] in the presence (+) or absence (−) of IAA (3.5 h). Tetracycline was added to the culture medium 24 h before extraction to replace endogenous HJURP expression with EGFP-fused HJURP variants. Signal intensities of KNL2 or KNL2[AID] on the blot were measured with normalization to α-tubulin intensity in each blot. Wild-type cells (WT) were also analyzed. α-tubulin was probed as a loading control. Molecular weight markers (kDa) are shown on the left. (B) Immunoblot analysis to detect HJURP, KNL2, CENP-C, and SNAP-CENP-A with anti-HJURP, anti-KNL2, anti-CENP-C, and anti-HA antibodies, respectively, in AID-based CENP-C knockout cells with Tet-based HJURP-cKO background expressing EGFP-HJURP or EGFP-HJURP[2E] in the presence (+) or absence (−) of IAA (3.5 h). Tetracycline was added to the culture medium 24 h before extraction to replace endogenous HJURP expression with EGFP-fused HJURP variants. Signal intensities of CENP-C or CENP-C[AID] on the blot were measured with normalization to α-tubulin intensity in each blot. Wild-type cells (WT) were also analyzed. α-tubulin was probed as a loading control. Molecular weight markers (kDa) are shown on the left. (C) Representative images of new CENP-A incorporation (SNAP-CENP-A) at mitosis in AID-based KNL2 or CENP-C knockout cells with Tet-based HJURP-cKO background expressing EGFP-HJURP (WT) or EGFP-HJURP[2E] (2E). Tetracycline was added to the culture medium 24 h before SNAP-labeling to replace endogenous HJURP expression with EGFP-fused HJURP variants. Cells were arrested in mitosis (M) with Nocodazole (Noc.) 6.5 h before SNAP-labeling in the absence (On) or presence (Off) of IAA. CENP-T was used as a centromere marker. DNA was stained with DAPI. Scale bar, 10 μm.

## G1 phase

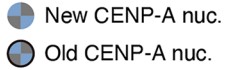

New CENP-A nuc.

Old CENP-A nuc.

### CENP-C KO cells

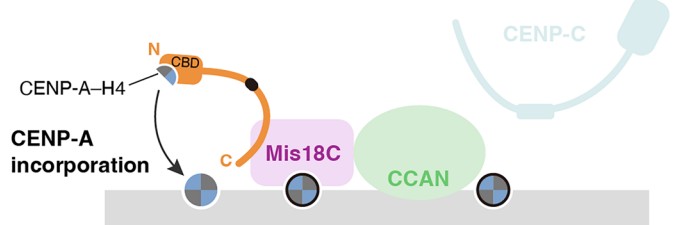

### HJURP²ᴱ/CENP-C KO cells

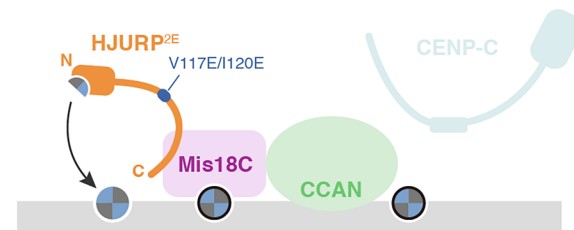

**Figure EV7. Summary of CENP-A incorporation pathways in CENP-C knockout cells, related to Fig. 7.**

Although the CENP-C pathway is compromised in CENP-C knockout (KO) cells (left panel), CENP-A incorporation still occurs through the Mis18C pathway. In CENP-C KO cells expressing HJURP²ᴱ mutant, which does not bind to CENP-C (right panel), CENP-A incorporation still occurs via the Mis18C pathway, as the HJURP²ᴱ mutant maintains its interaction with both CENP-A-H4 and Mis18C.

