## [Peer Review File · The EMBO Journal]

Dual pathways via CENP-C and Mis18C recruit HJURP for CENP-A deposition into vertebrate centromeres

Tetsuya Hori, Yutaka Mahana, Mariko Ariyoshi, and Tatsuo Fukagawa

Corresponding author(s): Tatsuo Fukagawa (fukagawa.tatsuo.fbs@osaka-u.ac.jp), Tetsuya Hori (t.hori.fbs@osaka-u.ac.jp)

Review Timeline:

Submission Date:	25th May 25
Editorial Decision:	8th Jul 25
Revision Received:	4th Oct 25
Editorial Decision:	14th Nov 25
Revision Received:	18th Nov 25
Accepted:	8th Dec 25

Editor: Hartmut Vodermaier

Transaction Report:

Dr. Tatsuo Fukagawa
Osaka University
Graduate School of Frontier Biosciences
1-3 Yamadaoka
Suita, Osaka 565-0871
Japan

8th Jul 2025

Re: EMBOJ-2025-121470
Dual pathways via CENP-C and Mis18C recruit HJURP for CENP-A deposition into vertebrate centromeres

Dear Tats,

Thank you for submitting your study on dual CENP-A deposition pathways in chicken cells for our consideration. It has now been seen by three expert referees, whose comments are copied below. As you will see, all reviewers acknowledge the interest of the subject as well as the technical quality and methodological innovations. However, they also indicate that the manuscript does not sufficiently clearly explain the key advances conveyed by the work, and would need to put the results better into the context of earlier work on epigenetic centromere specification in diverse mammalian and non-mammalian vertebrate species. In addition, they raise a number of experiment-related questions (testing assembly into a single or two separate complexes; AlphaFold use - ref 1; quantitation of degradation efficiency; following up on basis for cell lethality - ref 2; data description/presentation - esp. ref 3) that would need to be addressed prior to publication.

I conclude that we would be happy to consider this work further for EMBO Journal publication, pending satisfactory addressing of the noted points, as well as providing a more detailed discussion and comparison of diverse CENP-A recruitment pathways and their evolutionary divergence. Please be reminded that our single-major-revision-round policy makes it important to diligently respond to each referee point at the time of resubmission; therefore, please do not hesitate to contact me with any questions you may have regarding this revision. We would also be open to extension of the regular three-months revision period if needed; our 'scooping protection' (meaning that competing work appearing elsewhere in the meantime will not affect our considerations of your study) would of course remain valid also throughout such an extension.

Further information on preparing, formatting and uploading a revised manuscript can be found below and in our Guide to Authors. Thank you again for the opportunity to consider this work for The EMBO Journal, and I look forward to receiving your revised manuscript in due time.

With kind regards,

Hartmut

- 3) Revised manuscript text (including main tables, and figure legends for main and EV figures) has to be submitted as editable text file (e.g., .docx format). We encourage highlighting of changes (e.g., via text color) for the referees' reference.
- 4) Each main and each Expanded View (EV) figure should be uploaded as individual production-quality files (preferably in .eps, .tif, .jpg formats). For suggestions on figure preparation/layout, please refer to our Figure Preparation Guidelines: <http://bit.ly/EMBOPressFigurePreparationGuideline>
- 5) Point-by-point response letters should include the original referee comments in full together with your detailed responses to them (and to specific editor requests if applicable), and also be uploaded as editable (e.g., .docx) text files.
- 6) Please complete our Author Checklist, and make sure that information entered into the checklist is also reflected in the manuscript; the checklist will be available to readers as part of the Review Process File. A download link is found at the top of our Guide to Authors: embopress.org/page/journal/14602075/authorguide
- 7) All authors listed as (co-)corresponding need to deposit, in their respective author profiles in our submission system, a unique ORCID identifier linked to their name. Please see our Guide to Authors for detailed instructions.
- 8) Please note that supplementary information at EMBO Press has been superseded by the 'Expanded View' for inclusion of additional figures, tables, movies or datasets; with up to five EV Figures being typeset and directly accessible in the HTML version of the article. For details and guidance, please refer to: embopress.org/page/journal/14602075/authorguide#expandedview
- 9) To facilitate reproducibility and cross-laboratory adoption of methodologies, please structure the Materials & Methods section as outlined in our guide to authors, including a completed Reagents and Tools Table that can be downloaded from our author guidelines as well (<https://www.embopress.org/page/journal/14602075/authorguide#structuredmethods>).
- 10) Digital image enhancement is acceptable practice, as long as it accurately represents the original data and conforms to community standards. If a figure has been subjected to significant electronic manipulation, this must be clearly noted in the figure legend and/or the 'Materials and Methods' section. The editors reserve the right to request original versions of figures and the original images that were used to assemble the figure. Finally, we generally encourage uploading of numerical as well as gel/blot image source data; for details see: embopress.org/page/journal/14602075/authorguide#sourcedata

In the interest of ensuring the conceptual advance provided by the work, we recommend submitting a revision within 3 months (6th Oct 2025). Please discuss the revision progress ahead of this time with the editor if you require more time to complete the revisions. Use the link below to submit your revision:

Link Not Available

Referee #1:

Centromeres are epigenetically maintained by a self-propagating feedback loop involving the centromere-specific histone H3 variant CENP-A, together with several accessory proteins, including its dedicated chaperone HJURP, the DNA-binding protein CENP-C, and the Mis18 complex. Despite the essential role of this pathway in sustaining centromere identity and function, the molecular interactions and relative contributions of these components have diverged significantly—even among vertebrates—making it a compelling subject for comparative study. Key insights into this diversity have come from studies in human, *Xenopus*, and chicken cell lines (this paper).

More, specifically, this study follows up on a previous study published from the same lab that demonstrated a new role of the chicken CENP-C in recruiting HJURP to ectopic lacO arrays and enabling CENP-A incorporation into this locus. The previous study also demonstrated a direct protein interaction between HJURP and CENP-C and narrowed down the interacting region to the N- and C-termini of both proteins. Here, this study demonstrates that this role of CENP-C in recruiting HJURP can also be extended to endogenous centromeres in chicken DT40 cells and identified the exact interaction site between the two proteins.

The authors propose a dual recruitment pathway to ensure replenishment of CENP-A at centromeres over cell divisions.

Overall, the techniques and results of this study are high-quality of this paper and I have no concerns regarding them. However, I think it is important that the authors better highlight the (potential) novelty of their findings in the context of previously published work. More specifically, previous studies using *Xenopus* egg extracts (PMID: 28743005 and 37141119, both studies are cited by the authors) have also shown that both CENP-C and the Mis18 complex play roles in HJURP recruitment, CENP-A incorporation and demonstrated a direct protein interaction between CENP-C and KNL-2 (a Mis18 component). Therefore, the authors should clearly state these observations in the introduction and emphasize how their work advances current understanding-beyond confirming a mechanism in their experimental system that has previously been described in another vertebrate.

Based on their data, the authors propose a dual HJURP recruitment pathway where Mis18 and CENP-C act independently. This is different from the previous model that has been proposed based on the *Xenopus* data, in which all three proteins are proposed to be required to form a stable complex for efficient HJURP recruitment (PMID: 28743005). I am not convinced why the authors disfavor the previous model as this is still consistent with their findings that the Mis18-HJURP interaction is more stable in the presence of CENP-C. Mis18 and CENP-C do not compete for the same binding site in HJURP. The authors can perhaps include Mis18 in their assay in Figure 3D to test if they assemble into a single stoichiometric complex or two separate complexes as a direct test to their model. Considering these, the authors' justification to choose the bottom-model in Figure 5A is unclear from the text (last paragraph of P12). A clear and direct explanation would help.

Additional comments:

- Page 8: "Although HJURP co-precipitates with CENP-C in several species including human, chicken, and *Xenopus* as shown by pull-down experiments (Cao et al., 2024; Flores Servin et al, 2023; Tachiwana et al, 2015), it is not clear whether HJURP directly binds to CENP-C." The cited references do show a direct interaction between HJURP and CENP-C using either in vitro pull-down assay of human proteins (Tachiwana et al 2015) or in a heterologous system (rabbit reticulocyte lysates) of *Xenopus* proteins (French et al 2017, PMID: 28743005).
- The authors show that human CENP-C does not bind to HJURP via the corresponding region of chicken or *Xenopus* HJURP, nevertheless an interaction between human CENP-C and human HJURP has been demonstrated previously as the authors discussed (Tachiwana et al 2015). They can use AlphaFold using the full-length proteins to try to identify the part of human HJURP that interacts with human CENP-C. If the evidence of human CENP-C not interacting with human HJURP is stronger, this would also support their model that a CENP-C dependent HJURP recruitment occurs only in species where Mis18 interacts with CENP-A directly.
- In the discussion, the authors draw a correlation that "these dual pathways exist in organisms where CENP-C and Mis18 localize to centromeres independently." This is an interesting observation, and the authors can perhaps add to the discussion why this would necessitate dual (independent?) CENP-A deposition mechanisms.

Referee #2:

Centromere specification is crucial for accurate chromosomes segregation and cell viability. Our understanding of the epigenetic mechanisms that underlie the assembly of the centromere remains incomplete. The manuscript from Hori et al. defines the dual contributions of CENP-C and KNL2, of the Mis18 complex, in recruiting HJURP to existing centromeres, using chicken DT40 cell line. Inter-organismal comparisons such as this are important for understanding the general principles of epigenetic inheritance. The experiments in the manuscript are extremely well executed. One of the major strengths of the study is the innovative use of degron-tagged cell lines. This approach allows for precise temporal control of protein depletion and is effectively used to dissect the individual and combined roles of KNL2 and CENP-C in HJURP recruitment and CENP-A inheritance. The authors also demonstrate that while degradation of either KNL2 or CENP-C alone results in partial phenotypes, simultaneous depletion leads to more severe defects, highlighting the complementary functions of these proteins. Overall, this manuscript makes important contributions to the field and is well executed.

There are a few issues that should be addressed to strengthen the manuscript.

1. While the degron system is a powerful tool, it is known that degradation efficiency can vary between proteins. This caveat should be mentioned in the discussion. Moreover, degrons increase the rate of degradation but the overall amount of protein is still influenced by the rate of expression. This limitation should be acknowledged in the discussion, particularly when interpreting differences in phenotypes among HJURP, CENP-C, and KNL2 depletions, which is less straight forward than defining additive phenotypes using complete knockouts. Including quantitative data on the efficiency of protein degradation for each degron-tagged construct would also strengthen the conclusions.
2. The manuscript would benefit from clarification regarding the domain architecture of chicken HJURP (ggHJURP), specifically

the relationship between the CENP-C binding domain and the CENP-A interaction domain. This information is important for understanding the mechanistic basis of HJURP recruitment.

3. Given the growing evidence that HJURP may have functions beyond centromere assembly, the authors should explore the mechanism of cell death in HJURP 2E cells. It would be helpful to determine whether the observed lethality is due to chromosome missegregation or other cellular defects.

Referee #3:

The current understanding of the mechanism of CENP-A targeting to kinetochores is that the Mis18 complex associates with centromeres through an association with (among others) CENP-C and targets the CENP-A chaperone HJURP, which then deposits CENP-A to maintain centromere identity. These authors show that in chicken DT40 cells, CENP-C and the Mis18 complex target independently to centromeres. They then show that when either component alone is depleted, CENP-A incorporation is reduced, but not eliminated. When the two are removed at the same time, CENP-A incorporation is now abolished. Using AlphaFold3, the authors predict an interaction between HJURP and CENP-C. They identify key residues required for this and show that if they are mutated, then CENP-A targeting is abolished in the absence of Mis18 complex. Together, the experiments presented here reveal that in chickens (and apparently *Xenopus*, but not humans), two parallel closely interlinked pathways cooperate to target CENP-A to centromeres.

The experiments presented here are elegantly designed and executed, and the data are convincing. The paper is written clearly - with a few exceptions that I will mention. The question for the Editor is whether this represents an advance suitable for publication in the EMBO Journal. I am convinced by the story told here, but the authors (to their credit) make no attempt to hide the fact that this is a story that appears to be relevant in chicken and *Xenopus* but not in human (and presumably other mammals). Indeed, if it is true, as reported elsewhere, that CENP-C has a role in targeting the Mis18 complex to centromeres and does not exhibit the same mode of binding to HJURP shown here for chicken, then the experiments presented here could not be expected to work in human cells (hence avoiding the expected referee comment "Repeat this in human cells.").

I am "on the fence" about this MS. In view of the elegant technology and beautiful data shown here, then as an example of high-class science, the paper is worthy of publication in the EMBO J. As to whether this will broadly change thinking about centromere assembly in a field dominated by work on mammalian proteins, then I am not so sure. The authors "propose that these dual pathways exist in organisms where CENP-C and Mis18C localize to centromeres independently", but at present we have no idea whether dependent or independent localisation of CENP-C and the Mis18 complex to centromeres will turn out to be more prevalent and whether chickens are the paradigm or the exception.

Specific comments:

All of my specific comments are relatively minor and could be addressed by re-writing.

pp. 6,7 - "When we added IAA to cKO-HJURP cells, SNAP-CENP-A signals were significantly reduced (to 6.9%)"

This paragraph clearly refers to Figure 1, panels C-E. The authors should insert a call-out to the figure so that the readers do not have to figure this out for themselves.

p. 7 "HJURP localization is transient, and it is difficult to quantitatively assess HJURP levels at centromeres using cytogenetic methods. Therefore, quantitative ChIP-seq analysis is the best way to evaluate HJURP levels at centromeres in different KO cell lines. "

I apologise if it should be obvious, but it is not clear to me why ChIP-seq should be better than fluorescence microscopy at evaluating HJURP levels transiently associated at centromeres. Is this because larger numbers of cells are averaged? The data are beautiful and convincing, but the authors need to provide a convincing reason for this statement (or simply remove the statement - which appears several times).

I am uncomfortable about several statements made in reference to the pulldown experiments.

Abstract: "CENP-C, HJURP and Mis18C form a tight association on centromere chromatin." Pulldowns do not happen on chromatin. All the authors know is that when they make cell extracts, these proteins appear to retain their association.

p. 12 " the intensities of KNL2 and Mis18a co-precipitated with CENP-C were weaker than in HJURP On cells (Figure 5C), indicating that the CENP-C-Mis18C association is unstable in HJURP Off cells."

p. 15 "However, our co-IP experiments showed that the CENP-C-Mis18C association in HJURP KO cells and the Mis18C-HJURP association in CENP-C KO cells were less stable than those in control cells"

All the authors know was that the recovery of the relevant proteins was decreased in these pulldown experiments. They do not know that this was a result of complex stability. It could, for example, be due to reduced complex formation in those cells.

p. 15 "CENP-CCENP-C has multiple domains to function as a hub for centromere formation."
The authors should refer to the previous publication PMID: 26124289, entitled "CENP-C is a blueprint for constitutive centromere-associated network assembly within human kinetochores."

p. 9 "On the other hand, in the AF3 model of the ggHJURP-ggCENP-C complex, the potential CDK phosphorylation site S143 of ggHJURP is in a C-terminal extended region outside the CNEP-C binding site" should read "On the other hand, in the AF3 model of the ggHJURP-ggCENP-C complex, the potential CDK phosphorylation site S143 of ggHJURP is in a C-terminal extended region outside the CENP-C binding site"

Response to Reviewers Hori et al. (EMBOJ-2025-121470)

Reviewer #1

*Centromeres are epigenetically maintained by a self-propagating feedback loop involving the centromere-specific histone H3 variant CENP-A, together with several accessory proteins, including its dedicated chaperone HJURP, the DNA-binding protein CENP-C, and the Mis18 complex. Despite the essential role of this pathway in sustaining centromere identity and function, the molecular interactions and relative contributions of these components have diverged significantly—even among vertebrates—making it a compelling subject for comparative study. Key insights into this diversity have come from studies in human, *Xenopus*, and chicken cell lines (this paper).*

More, specifically, this study follows up on a previous study published from the same lab that demonstrated a new role of the chicken CENP-C in recruiting HJURP to ectopic lacO arrays and enabling CENP-A incorporation into this locus. The previous study also demonstrated a direct protein interaction between HJURP and CENP-C and narrowed down the interacting region to the N- and C-termini of both proteins. Here, this study demonstrates that this role of CENP-C in recruiting HJURP can also be extended to endogenous centromeres in chicken DT40 cells and identified the exact interaction site between the two proteins. The authors propose a dual recruitment pathway to ensure replenishment of CENP-A at centromeres over cell divisions.

Overall, the techniques and results of this study are high-quality of this paper and I have no concerns regarding them.

*However, I think it is important that the authors better highlight the (potential) novelty of their findings in the context of previously published work. More specifically, previous studies using *Xenopus* egg extracts (PMID: 28743005 and 37141119, both studies are cited by the authors) have also shown that both CENP-C and the Mis18 complex play roles in HJURP recruitment, CENP-A incorporation and demonstrated a direct protein interaction between CENP-C and KNL-2 (a Mis18 component). Therefore, the authors should clearly state these observations in the introduction and emphasize how their work advances current understanding—beyond confirming a mechanism in their experimental system that has previously been described in another vertebrate.*

We appreciate the reviewer recognizing the high quality of our work. Based on the comments, we emphasized the significance of our work in the introduction of the revised text.

*Based on their data, the authors propose a dual HJURP recruitment pathway where Mis18 and CENP-C act independently. This is different from the previous model that has been proposed based on the *Xenopus* data, in which all three proteins are proposed to be required to form a stable complex for efficient HJURP recruitment (PMID: 28743005). I am not convinced why the authors disfavor the previous model as this is still consistent with their findings that the Mis18-HJURP interaction is more stable in the presence of CENP-C. Mis18 and CENP-C do not compete for the same binding site in HJURP. The authors can perhaps include Mis18 in their assay in Figure 3D to test if they assemble into a single stoichiometric complex or two separate complexes as a direct test to their model. Considering these, the authors' justification to choose the bottom-model in Figure 5A is unclear from the text (last paragraph of P12). A clear and direct explanation would help.*

We appreciate this comment. Although we did not rule out the possibility of the

bottom model in Figure 5A (Figure 6A in the revised version), we prefer the upper model, in which it is not necessary to form a stable stoichiometric ternary complex, because:

1) HJURP localization remained on centromeres in Mis18C or CENP-C KO cells (Figure 2D and 5B), and 2) CENP-C localization did not change at interphase centromeres in Mis18C KO (Figure 2B). It would be difficult to explain such a phenotype if the bottom model were case. We agree with the reviewer that the weaker interaction between HJURP and Mis18C in the absence of CENP-C may support the bottom model (previous Figure 5D: now Figure 6D). However, CENP-C localization did not change in Mis18C KO, and efficiency of HJURP recovery by IP with CENP-C did not change in Mis18C KO (new Figure 6B). Knocking out CENP-C may alter the organization of centromeric chromatin, leading to a weak interaction between Mis18C and HJURP (Figure 6D), which could also explain the upper model. In addition to the co-IP data, we included the explanation why we prefer the upper model in the revised version.

Furthermore, we performed in vitro experiments. We prepared the Mis18C and its binding domain of HJURP as recombinant proteins and demonstrated that CENP-C does not compete with Mis18C-HJURP binding (see new Figure EV5). We note that we could not include Mis18 in Figure 3D, because we could not purify the full-length HJURP for technical reasons. Based on the combined results of co-IP, cell biology, and ChIP-seq experiments, we favor the upper model in Figure 6A. However, distinguishing complex formation by in vitro experiments alone is essentially difficult because our in vitro data support both the upper and bottom models. While we prefer the upper model when combining other data, we emphasize that we did not completely rule out the bottom model. It is also possible that two states exist in cells. We have carefully edited these statements in the revised version.

Additional comments:

• Page 8: *"Although HJURP co-precipitates with CENP-C in several species including human, chicken, and Xenopus as shown by pull-down experiments (Cao et al., 2024; Flores Servin et al, 2023; Tachiwana et al, 2015), it is not clear whether HJURP directly binds to CENP-C."*

The cited references do show a direct interaction between HJURP and CENP-C using either in vitro pull-down assay of human proteins (Tachiwana et al 2015) or in a heterologous system (rabbit reticulocyte lysates) of Xenopus proteins (French et al 2017, PMID: 28743005).

We agree with the reviewer's comment and have carefully revised this sentence, including the indicated references in the revised version.

• *The authors show that human CENP-C does not bind to HJURP via the corresponding region of chicken or Xenopus HJURP, nevertheless an interaction between human CENP-C and human HJURP has been demonstrated previously as the authors discussed (Tachiwana et al 2015). They can use AlphaFold using the full-length proteins to try to identify the part of human HJURP that interacts with human CENP-C. If the evidence of human CENP-C not interacting with human HJURP is stronger, this would also support their model that a CENP-C dependent HJURP recruitment occurs only in species where Mis18 interacts with CENP-A directly.*

We appreciate this comment. Using AlphaFold3, we performed a prediction with

the entire region of human HJURP and a C-terminal region of human CENP-C, which was used in the previous study (Tachiwana *et al.*, 2015), and found no clear interaction between the two proteins (see new Figure EV 3C). We did not use the full-length CENP-C because a large portion of it contains disordered regions, which leads to ambiguous predictions. To exclude the possibility that the disordered nature of full-length HJURP affects the prediction, we also performed prediction using various 200-aa fragments of human HJURP and the CENP-C C-terminal region (see the figure below for the reviewer). As you can see, the pLDDT scores were low for human HJURP residues, indicating that AlphaFold3 did not support an interaction between human HJURP and CENP-C. Although a previous pull-down experiment showed an interaction between human CENP-C and human HJURP, this interaction may be weaker than the interaction between chicken CENP-C and chicken HJURP. We have added this prediction data to the revised version and expanded the discussion.

Figure for Reviewer
 A) The pLDDT scores for the structure of chicken HJURP (aa 100-150) predicted as a complex with chicken CENP-C (aa 677-864). B) The pLDDT scores for the structure of human HJURP (aa 137-192) predicted as a complex with human CENP-C (aa 760-943). C) The pLDDT scores for the structure of various regions of human HJURP modeled with human CENP-C (aa 631-943; used in the previous study).

- *In the discussion, the authors draw a correlation that "these dual pathways exist in organisms where CENP-C and Mis18 localize to centromeres independently." This is an interesting observation, and the authors can perhaps add to the discussion why this would necessitate dual (independent?) CENP-A deposition mechanisms.*

We appreciate this comment and have expanded our discussion in the revised version.

Reviewer #2

Centromere specification is crucial for accurate chromosomes segregation and cell viability. Our understanding of the epigenetic mechanisms that underlie the assembly of the centromere remains incomplete. The manuscript from Hori et al. defines the dual contributions of CENP-C and KNL2, of the Mis18 complex, in recruiting HJURP to existing centromeres, using chicken DT40 cell line. Inter-organismal comparisons such as this are important for understanding the general principles of epigenetic inheritance. The experiments in the manuscript are extremely well executed. One of the major strengths of the study is the innovative use of degron-tagged cell lines. This approach allows for precise temporal control of protein depletion and is effectively used to dissect the individual and combined roles of KNL2 and CENP-C in HJURP recruitment and CENP-A inheritance. The authors also demonstrate that while degradation of either

KNL2 or CENP-C alone results in partial phenotypes, simultaneous depletion leads to more severe defects, highlighting the complementary functions of these proteins. Overall, this manuscript makes important contributions to the field and is well executed. There are a few issues that should be addressed to strengthen the manuscript.

We appreciate the positive comments. We addressed all concerns in an effort to strengthen the manuscript.

1. While the degron system is a powerful tool, it is known that degradation efficiency can vary between proteins. This caveat should be mentioned in the discussion. Moreover, degrons increase the rate of degradation but the overall amount of protein is still influenced by the rate of expression. This limitation should be acknowledged in the discussion, particularly when interpreting differences in phenotypes among HJURP, CENP-C, and KNL2 depletions, which is less straight forward than defining additive phenotypes using complete knockouts. Including quantitative data on the efficiency of protein degradation for each degron-tagged construct would also strengthen the conclusions.

We agree with the reviewer. In the revised text, we described degradation efficiency and the limitations of the degron system. We also included data in the revised version (New Figures 2A-B, EV1A-C, EV1I EV2A-F, EV5G, and EV6A-B) showing the efficiency of protein degradation for each degron-tagged construct.

2. The manuscript would benefit from clarification regarding the domain architecture of chicken HJURP (ggHJURP), specifically the relationship between the CENP-C binding domain and the CENP-A interaction domain. This information is important for understanding the mechanistic basis of HJURP recruitment.

We appreciate this comment. In the revised version, we included new information about the CENP-A binding domain in Figure 3A.

3. Given the growing evidence that HJURP may have functions beyond centromere assembly, the authors should explore the mechanism of cell death in HJURP 2E cells. It would be helpful to determine whether the observed lethality is due to chromosome missegregation or other cellular defects.

We appreciate this comment. We performed phenotype analyses in mitotic progression for HJURP 2E cells and included the results in the revised version as Figure 4. We also included FACS data for HJURP 2E cells (Figure EV4B-C). The HJURP 2E cells increased the mitotic fraction, indicating that it took these cells longer to complete mitosis. Additionally, we found abnormal mitotic cells and an increase in apoptotic cells in HJURP 2E cells. We interpret this as meaning that HJURP 2E reduces CENP-A incorporation, leading to abnormal mitosis and cell death.

Reviewer #3

The current understanding of the mechanism of CENP-A targeting to kinetochores is that the Mis18 complex associates with centromeres through an association with (among others) CENP-C and targets the CENP-A chaperone HJURP, which then deposits CENP-A to maintain centromere identity. These authors show that in chicken DT40 cells, CENP-C and the Mis18 complex target independently to centromeres. They then show that when either component alone is depleted, CENP-A incorporation is

reduced, but not eliminated. When the two are removed at the same time, CENP-A incorporation is now abolished. Using AlphaFold3, the authors predict an interaction between HJURP and CENP-C. They identify key residues required for this and show that if they are mutated, then CENP-A targeting is abolished in the absence of Mis18 complex. Together, the experiments presented here reveal that in chickens (and apparently Xenopus, but not humans), two parallel closely interlinked pathways cooperate to target CENP-A to centromeres.

The experiments presented here are elegantly designed and executed, and the data are convincing. The paper is written clearly - with a few exceptions that I will mention. The question for the Editor is whether this represents an advance suitable for publication in the EMBO Journal. I am convinced by the story told here, but the authors (to their credit) make no attempt to hide the fact that this is a story that appears to be relevant in chicken and Xenopus but not in human (and presumably other mammals). Indeed, if it is true, as reported elsewhere, that CENP-C has a role in targeting the Mis18 complex to centromeres and does not exhibit the same mode of binding to HJURP shown here for chicken, then the experiments presented here could not be expected to work in human cells (hence avoiding the expected referee comment "Repeat this in human cells.").

I am "on the fence" about this MS. In view of the elegant technology and beautiful data shown here, then as an example of high-class science, the paper is worthy of publication in the EMBO J. As to whether this will broadly change thinking about centromere assembly in a field dominated by work on mammalian proteins, then I am not so sure. The authors "propose that these dual pathways exist in organisms where CENP-C and Mis18C localize to centromeres independently", but at present we have no idea whether dependent or independent localisation of CENP-C and the Mis18 complex to centromeres will turn out to be more prevalent and whether chickens are the paradigm or the exception.

We appreciate the positive comments about the quality of our work. We believe that there is diversity in centromere assembly among organisms. In addition to common mechanisms, it is important to demonstrate centromere assembly mechanisms that differ from those in human cells.

Specific comments:

All of my specific comments are relatively minor and could be addressed by re-writing.

pp. 6,7 - "When we added IAA to cKO-HJURP cells, SNAP-CENP-A signals were significantly reduced (to 6.9%)"

This paragraph clearly refers to Figure 1, panels C-E. The authors should insert a call-out to the figure so that the readers do not have to figure this out for themselves.

We appreciate this comment. This sentence does indeed refer to the data in Figures 1C-E. In the revised version, we mentioned "Figure 1C-E" in this sentence.

p. 7 "HJURP localization is transient, and it is difficult to quantitatively assess HJURP levels at centromeres using cytogenetic methods. Therefore, quantitative ChIP-seq analysis is the best way to evaluate HJURP levels at centromeres in different KO cell lines. "

I apologise if it should be obvious, but it is not clear to me why ChIP-seq should be better than fluorescence microscopy at evaluating HJURP levels transiently associated at centromeres. Is this because larger numbers of cells are averaged? The data are beautiful and convincing, but the authors need to provide a convincing reason for this

statement (or simply remove the statement - which appears several times).

We agree with this comment. We removed this statement from the revised version because our reasoning might not be correct.

I am uncomfortable about several statements made in reference to the pulldown experiments.

Abstract: "CENP-C, HJURP and Mis18C form a tight association on centromere chromatin." Pulldowns do not happen on chromatin. All the authors know is that when they make cell extracts, these proteins appear to retain their association.

We appreciate the comment that pulldowns do not occur on chromatin. We have carefully revised the statement on pulldown experiments in the revised version.

p. 12 " the intensities of KNL2 and Mis18a co-precipitated with CENP-C were weaker than in HJURP On cells (Figure 5C), indicating that the CENP-C-Mis18C association is unstable in HJURP Off cells."

p. 15 "However, our co-IP experiments showed that the CENP-C-Mis18C association in HJURP KO cells and the Mis18C-HJURP association in CENP-C KO cells were less stable than those in control cells"

All the authors know was that the recovery of the relevant proteins was decreased in these pulldown experiments. They do not know that this was a result of complex stability. It could, for example, be due to reduced complex formation in those cells.

We agree with this comment. The low recovery of a protein by co-IP does not indicate protein stability. We have carefully revised such statements in the updated version.

p. 15 "CENP-CCENP-Chas multiple domains to function as a hub for centromere formation."

The authors should refer to the previous publication PMID: 26124289, entitled "CENP-C is a blueprint for constitutive centromere-associated network assembly within human kinetochores."

We appreciate this comment and have cited the indicated reference in the revised version.

p. 9 "On the other hand, in the AF3 model of the ggHJURP-ggCENP-C complex, the potential CDK phosphorylation site S143 of ggHJURP is in a C-terminal extended region outside the CNEP-C binding site" should read "On the other hand, in the AF3 model of the ggHJURP-ggCENP-C complex, the potential CDK phosphorylation site S143 of ggHJURP is in a C-terminal extended region outside the CENP-C binding site"

We appreciate this comment. Based on this reviewer's suggestion, I changed the following sentence in the revised version: "On the other hand, in the AF3 model of the ggHJURP-ggCENP-C complex, the potential CDK phosphorylation site S143 of ggHJURP is in a C-terminal extended region outside the CENP-C binding site".

Dr. Tatsuo Fukagawa
Osaka University
Graduate School of Frontier Biosciences
1-3 Yamadaoka
Suita, Osaka 565-0871
Japan

14th Nov 2025

Re: EMBOJ-2025-121470R
Dual pathways via CENP-C and Mis18C recruit HJURP for CENP-A deposition into vertebrate centromeres

Dear Tats,

Thank you for submitting your revised manuscript to The EMBO Journal. Two of the original referees have now reviewed it once more, and their comments are copied below. Both of them appreciate your revisions and improvements to the study, but referee 1 still retains several concerns regarding presentation of the relevant background, interpretation of some of the data, and discussion of the new results in the context of previous findings in various systems. Furthermore, referee 3 suggests that the study would benefit from careful language proofreading and editing, maybe ideally by a native speaker. Based on their reports, I decided to return the manuscript to you for an additional round of minor revision, in which I would invite you to address these remaining referee concerns.

When preparing a re-revised manuscript, please also take care of the following formal/editorial issues:

- Please adjust the order of the manuscript sections, and also make sure to use the correct section headers: Title page with complete author information, Abstract, Keywords, Introduction, Results, Discussion, Methods, Data Availability, Acknowledgements, Disclosure and Competing Interests Statement, References, Main Figure Legends, Tables, Expanded Figure Legends.
Also, a reference to Appendix material in the main text is not needed and should be removed.
- Please reduce the number of keywords on the abstract page to five (ideally choosing broad general terms), and make sure to place them right after the abstract.
- Please carefully go through the reference list and make sure that each reference is complete with citation year, volume, and page/eLocator numbers - this information is currently missing for several of them.
- As we are switching from a free-text author contribution statement towards a more formal statement based on Contributor Role Taxonomy (CRediT) terms, please remove the present Author Contribution section and instead specify each author's contribution(s) directly in the Author Information page of our submission system during upload of the final manuscript. See <https://casrai.org/credit/> for more information.
- Please rename the Conflict of Interest section into "Disclosure and Competing Interests Statement", in accordance with our updated Guide to Authors (<https://www.embopress.org/competing-interests>)
- Please move the Reagents and Tools table from the main article file, and upload it as a separate text file. Also, please make sure to adhere to the template table downloadable from our author guidelines: <https://www.embopress.org/page/journal/14693178/authorguide#structuredmethods>
- In the Data Availability section, please remove the referee access information, and ensure that the data are becoming publicly available at this point. Furthermore, please include a direct link to the database where data have been deposited (suggested wording: "The [structural coordinates | microarray | mass spectrometry] data from this publication have been deposited to the [name of the database] database [URL] and assigned the identifier [accession | permalink | hashtag].").
- Please note that Source data files need to be saved in a scheme one figure/folder and then uploaded as .zip files. E.g. all the Source data files for figure 1 need to be saved in a single folder and this needs to be zipped and then uploaded as "SD figure 1.zip" file.
- During routine pre-acceptance checks, our data editors noted that the exact p values are not provided in the legends of figures 1C-E; 7A, B; EV1 J, EV2 A, B - please amend them, with changes/additions highlighted via the "Track changes" option to facilitate our final checking.

- Finally, please provide suggestions for a short 'blurb' text prefacing and summing up the conceptual aspect of the study in two sentences (max. 250 characters), followed by 3-5 one-sentence 'bullet points' with brief factual statements of key results of the paper; they will form the basis of an editor-written 'Synopsis' accompanying the online version of the article. Please also upload a synopsis image, which can be used as a "visual title" for the synopsis section of your paper. The image (maybe based on a simplified/condensed panel from Figure 7C?) should be in PNG or JPG format, and please make sure that it remains in the modest dimensions of (exactly) 550 pixels wide and 300-600 pixels high.

I am therefore returning the manuscript to you for a final round of revision, with the link below for eventual resubmission. Should you have any questions regarding the referee comments or this decision, please do not hesitate to contact me directly.

With kind regards,

Hartmut

9) To facilitate reproducibility and cross-laboratory adoption of methodologies, please structure the Materials & Methods section as outlined in our guide to authors, including a completed Reagents and Tools Table that can be downloaded from our author guidelines as well (<https://www.embopress.org/page/journal/14602075/authorguide#structuredmethods>).

10) Digital image enhancement is acceptable practice, as long as it accurately represents the original data and conforms to

community standards. If a figure has been subjected to significant electronic manipulation, this must be clearly noted in the figure legend and/or the 'Materials and Methods' section. The editors reserve the right to request original versions of figures and the original images that were used to assemble the figure. Finally, we generally encourage uploading of numerical as well as gel/blot image source data; for details see: embopress.org/page/journal/14602075/authorguide#sourcedata

In the interest of ensuring the conceptual advance provided by the work, we recommend submitting a revision within 3 months (12th Feb 2026). Please discuss the revision progress ahead of this time with the editor if you require more time to complete the revisions. Use the link below to submit your revision:

Link Not Available

Referee #1:

Response to author response 1: "We appreciate the reviewer recognizing the high quality of our work. Based on the comments, we emphasized the significance of our work in the introduction of the revised text."

While the authors now cite the previous results showing that in *Xenopus* CENP-C also assists in HJURP recruitment, they collapse these previous results into one very short sentence ("In fact, there is a CENP-C dependent CENP-A incorporation in *Xenopus* cells (..."). To properly acknowledge these previous results, the authors should expand on this a bit more mentioning the existence of the dual pathway (Mis18c and CENP-C) for HJURP recruitment. In addition, previous studies have investigated CENP-A incorporation in native centromeres of sperm nuclei when incubated with *Xenopus* egg extracts (PMID: 28743005 and 37141119). I think for the reader it is important that the authors describe how their study investigating native chicken centromeres is different from these previous studies.

Response to author response 2: "We appreciate this comment. Although we did not rule out the possibility of the bottom model in Figure 5A (Figure 6A in the revised version), we prefer the upper model, in which it is not necessary to form a stable stoichiometric ternary complex, because:

1) HJURP localization remained on centromeres in Mis18C or CENP-C KO cells (Figure 2D and 5B), and 2) CENP-C localization did not change at interphase centromeres in Mis18C KO (Figure 2B). It would be difficult to explain such a phenotype if the bottom model were case. We agree with the reviewer that the weaker interaction between HJURP and Mis18C in the absence of CENP-C may support the bottom model (previous Figure 5D: now Figure 6D). However, CENP-C localization did not change in Mis18C KO, and efficiency of HJURP recovery by IP with CENP-C did not change in Mis18C KO (new Figure 6B). Knocking out CENP-C may alter the organization of centromeric chromatin, leading to a weak interaction between Mis18C and HJURP (Figure 6D), which could also explain the upper model. In addition to the co-IP data, we included the explanation why we prefer the upper model in the revised version.

Furthermore, we performed *in vitro* experiments. We prepared the Mis18C and its binding domain of HJURP as recombinant proteins and demonstrated that CENP-C does not compete with Mis18C-HJURP binding (see new Figure EV5). We note that we could not include Mis18 in Figure 3D, because we could not purify the full-length HJURP for technical reasons. Based on the combined results of co-IP, cell biology, and ChIP-seq experiments, we favor the upper model in Figure 6A. However, distinguishing complex formation by *in vitro* experiments alone is essentially difficult because our *in vitro* data support both the upper and bottom models. While we prefer the upper model when combining other data, we emphasize that we did not completely rule out the bottom model. It is also possible that two states exist in cells. We have carefully edited these statements in the revised version."

I think the authors agree that both models are consistent with the data. The fact that depletion of M18BP1 or CENP-C individually does not completely abolish CENP-A loading makes the authors favor the top model of an independent recruitment pathway. However, there is no direct evidence that a stoichiometric complex of all three components as described in the bottom model cannot form. The fact that the M18BP1 interaction with HJURP doesn't compete out CENP-C would be consistent with such a complex. It is possible that both these forms exist in cells, perhaps as a failsafe mechanism. That said, the author's justification based on localization of CENP-C and M18BP1 independent of each other does not lend support to either model in my understanding. CENP-C retention upon M18BP1 depletion could be due to its other contacts with CCAN or CENP-

A itself. M18BP1 retention upon CENP-C depletion could be due to a direct association with CENP-A itself. If the authors agree with this, these arguments to favor one model over the other should be removed.

Also, the last paragraph (page 13) in the section "Stable CENP-C-HJURP interaction.." is confusing. The authors describe their results in Figure 6C-D on the stability of the Mis18C-HJURP-CENP-C complex, which favors the bottom model (ternary complex) in Figure 6A. Instead, the authors state that these results favor the top model (independent recruitment).

Response to author response 4: "We appreciate this comment and have expanded our discussion in the revised version."

The argument that CENP-C may lose its Mis18C-recruiting function in *Xenopus* and chickens, while acquiring an HJURP-binding function instead, and that this change may explain why these organisms use dual pathways for CENP-A incorporation contradicts previous findings showing that *Xenopus* M18BP1 binds to CENP-C using a highly conserved SANTA domain, which is required for its metaphase centromere localization (PMID: 30606714). I do not have a better model explaining this discrepancy in HJURP recruitment between organisms (chicken/*Xenopus* vs humans) but perhaps the fact that CENP-B also defines centromeres in humans and interacts with several inner kinetochore components might alleviate the need of a dual HJURP recruitment pathway?

Referee #3:

The mechanism(s) of CENP-A deposition at centromeres remain an active area of research into chromosome segregation. This MS from Hori and co-workers reveals that in chicken DT40 cells there appear to be two parallel pathways that recruit the CENP-A chaperone HJURP to centromeres. One involves the Mis18 complex while the second involves the CCAN protein CENP-C. Previous work had suggested that the Mis18 complex was key for HJURP targeting, at least in human cells, although a role for CENP-C working together with Mis18C and HJURP in CENP-A recruitment had been reported in *Xenopus*. This work shows that the CENP-C and Mis18 pathways can work in parallel in chicken cells. The experiments reported here are of high quality and the discussion of the results is very thoughtful. On my initial reading of the MS, I was not convinced that the results reported here would be of sufficient novelty and general interest to merit a place in the EMBO J. However, upon further consideration and in respect to the reviews of the other two referees, I have revised that opinion. The authors have performed extensive re-writing and performed new experiments to address the referee comments and have made substantial efforts to show how their observations relate to the work of others. Although the re-writing could benefit from some editing for English usage, it is clear what the authors are meaning to say. I thus support acceptance of the revised MS.

Response to Reviewers Hori et al. (EMBOJ-2025-121470R)

Reviewer #1

Response to author response 1: "We appreciate the reviewer recognizing the high quality of our work. Based on the comments, we emphasized the significance of our work in the introduction of the revised text."

*While the authors now cite the previous results showing that in *Xenopus* CENP-C also assists in HJURP recruitment, they collapse these previous results into one very short sentence ("In fact, there is a CENP-C dependent CENP-A incorporation in *Xenopus* cells (...". To properly acknowledge these previous results, the authors should expand on this a bit more mentioning the existence of the dual pathway (Mis18c and CENP-C) for HJURP recruitment.*

*In addition, previous studies have investigated CENP-A incorporation in native centromeres of sperm nuclei when incubated with *Xenopus* egg extracts (PMID: 28743005 and 37141119). I think for the reader it is important that the authors describe how their study investigating native chicken centromeres is different from these previous studies.*

We appreciate this comment. As suggested, we expanded the introduction to previous *Xenopus* studies considerably.

Response to author response 2: "We appreciate this comment. Although we did not rule out the possibility of the bottom model in Figure 5A (Figure 6A in the revised version), we prefer the upper model, in which it is not necessary to form a stable stoichiometric ternary complex, because:1) HJURP localization remained on centromeres in Mis18C or CENP-C KO cells (Figure 2D and 5B), and 2) CENP-C localization did not change at interphase centromeres in Mis18C KO (Figure 2B). It would be difficult to explain such a phenotype if the bottom model were case. We agree with the reviewer that the weaker interaction between HJURP and Mis18C in the absence of CENP-C may support the bottom model (previous Figure 5D: now Figure 6D). However, CENP-C localization did not change in Mis18C KO, and efficiency of HJURP recovery by IP with CENP-C did not change in Mis18C KO (new Figure 6B). Knocking out CENP-C may alter the organization of centromeric chromatin, leading to a weak interaction between Mis18C and HJURP (Figure 6D), which could also explain the upper model. In addition to the co-IP data, we included the explanation why we prefer the upper model in the revised version. Furthermore, we performed in vitro experiments. We prepared the Mis18C and its binding domain of HJURP as recombinant proteins and demonstrated that CENP-C does not compete with Mis18C-HJURP binding (see new Figure EV5). We note that we could not include Mis18 in Figure 3D, because we could not purify the full-length HJURP for technical reasons. Based on the combined results of co-IP, cell biology, and ChIP-seq experiments, we favor the upper model in Figure 6A. However, distinguishing complex formation by in vitro experiments alone is essentially difficult because our in vitro data support both the upper and bottom models. While we prefer the upper model when combining other data, we emphasize that we did not completely rule out the bottom model. It is also possible that two states exist in cells. We have carefully edited these statements in the revised version."

I think the authors agree that both models are consistent with the data. The fact that depletion of M18BP1 or CENP-C individually does not completely abolish CENP-A loading makes the authors favor the top model of an independent recruitment pathway. However, there is no direct evidence that a stoichiometric complex

of all three components as described in the bottom model cannot form. The fact that the M18BP1 interaction with HJURP doesn't compete out CENP-C would be consistent with such a complex. It is possible that both these forms exist in cells, perhaps as a failsafe mechanism. That said, the author's justification based on localization of CENP-C and M18BP1 independent of each other does not lend support to either model in my understanding. CENP-C retention upon M18BP1 depletion could be due to its other contacts with CCAN or CENP-A itself. M18BP1 retention upon CENP-C depletion could be due to a direct association with CENP-A itself. If the authors agree with this, these arguments to favor one model over the other should be removed. Also, the last paragraph (page 13) in the section "Stable CENP-C-HJURP interaction.." is confusing. The authors describe their results in Figure 6C-D on the stability of the Mis18C-HJURP-CENP-C complex, which favors the bottom model (ternary complex) in Figure 6A. Instead, the authors state that these results favor the top model (independent recruitment).

We appreciate this comment. As the reviewer mentioned, we should remove the sentence "We prefer the top model" from the text. In addition to removing this sentence, we tried to make it clear throughout the text that both models are possible. Furthermore, we revised the last paragraph on page 13 to support both models.

Response to author response 4: "We appreciate this comment and have expanded our discussion in the revised version."

The argument that CENP-C may lose its Mis18C-recruiting function in Xenopus and chickens, while acquiring an HJURP-binding function instead, and that this change may explain why these organisms use dual pathways for CENP-A incorporation contradicts previous findings showing that Xenopus M18BP1 binds to CENP-C using a highly conserved SANTA domain, which is required for its metaphase centromere localization (PMID: 30606714). I do not have a better model explaining this discrepancy in HJURP recruitment between organisms (chicken/Xenopus vs humans) but perhaps the fact that CENP-B also defines centromeres in humans and interacts with several inner kinetochore components might alleviate the need of a dual HJURP recruitment pathway?

We appreciate this comment. As the reviewer noted, Xenopus M18BP1 binds to CENP-C. However, this interaction occurs during mitosis. Our hypothesis remains possible because CENP-A is incorporated during the G1 phase in Xenopus cells. Nevertheless, the contribution of CENP-B is an interesting idea. Therefore, we propose that the existence of CENP-B may reflect the use of the dual pathway in the revised text.

Reviewer #3

The mechanism(s) of CENP-A deposition at centromeres remain an active area of research into chromosome segregation. This MS from Hori and co-workers reveals that in chicken DT40 cells there appear to be two parallel pathways that recruit the CENP-A chaperone HJURP to centromeres. One involves the Mis18 complex while the second involves the CCAN protein CENP-C. Previous work had suggested that the Mis18 complex was key for HJURP targeting, at least in human cells, although a role for CENP-C working together with Mis18C and HJURP in CENP-A recruitment had been reported in Xenopus. This work shows that the CENP-C and Mis18 pathways can work in parallel in chicken cells. The experiments reported here are of high quality and the discussion of the results is very thoughtful. On my initial reading of the MS, I was not convinced that the results reported here would be of sufficient novelty and general

interest to merit a place in the EMBO J. However, upon further consideration and in respect to the reviews of the other two referees, I have revised that opinion. The authors have performed extensive re-writing and performed new experiments to address the referee comments and have made substantial efforts to show how their observations relate to the work of others. Although the re-writing could benefit from some editing for English usage, it is clear what the authors are meaning to say. I thus support acceptance of the revised MS.

We appreciate the positive comments. We also edited the manuscript for English usage to finalize it.

Dr. Tatsuo Fukagawa
Osaka University
Graduate School of Frontier Biosciences
1-3 Yamadaoka
Suita, Osaka 565-0871
Japan

8th Dec 2025

Re: EMBOJ-2025-121470R1
Dual pathways via CENP-C and Mis18C recruit HJURP for CENP-A deposition into vertebrate centromeres

Dear Tatsuo,

Thank you for submitting your final revised manuscript for our consideration. I am pleased to inform you that we have now accepted it for publication in The EMBO Journal.

You may qualify for financial assistance for your publication charges - either via a Springer Nature fully open access agreement or an EMBO initiative. Check your eligibility: <https://link.springer.com/journal/44318/how-to-publish-with-us>

With kind regards,

Hartmut

Please note that it is The EMBO Journal policy for the transcript of the editorial process (containing referee reports and your response letters) to be published as an online supplement to each paper. If you should prefer removal of any referee-only figures included in the point-by-point response(s), e.g. because they may still be used for future publication or because they have been reproduced from published work by others, please do let us know immediately via response email.

More information is available here: <https://link.springer.com/partners/embo-press/editorial-policies#Peer%20review>